# V-MAGE: A Game Evaluation Framework for Assessing Vision-Centric Capabilities in Multimodal Large Language Models

## Abstract

Recent advancements in Multimodal Large Language Models (MLLMs) have demonstrated impressive capabilities in visual-text processing. However, existing static image-text benchmarks are insufficient for evaluating their dynamic perception and interactive reasoning abilities. We introduce **V**ision-centric **M**ultiple **A**bilities **G**ame **E**valuation (**V-MAGE**), a novel game-based evaluation framework designed to systematically assess MLLMs' visual reasoning in interactive, continuous-space environments. V-MAGE features five distinct video games comprising over 30 carefully constructed evaluation scenarios. These scenarios are set in free-form, visually complex environments that require models to interpret dynamic game states and make decisions based solely on visual input, thereby closely reflecting the conditions encountered by human players. To ensure robust and interpretable comparisons across models, V-MAGE employs a dynamic ELO-based ranking system that accounts for varying difficulty levels and task diversity. Benchmarking state-of-the-art MLLMs against human baselines reveals that while leading models approach human-level performance in simple tasks, their performance drops significantly in complex scenarios requiring advanced reasoning and task orchestration. This persistent performance gap highlights fundamental limitations in current MLLMs' ability to perform real-time, vision-grounded interactions. Through extensive analyses, we demonstrate the utility of V-MAGE in uncovering these limitations and providing actionable insights for improving the visual and reasoning capabilities of MLLMs in dynamic, interactive settings.

## 1 Introduction

Building on the success of Large Language Models (LLMs) in text-based tasks(Bai et al. (2023); Cai et al. (2024); OpenAI (2023b)), researchers have extended their capabilities to visual-text multimodal tasks through Multimodal Large Language Models (MLLMs)(OpenAI (2023a); Liu et al. (2023a); Team (2023); Yang et al. (2023); Li et al. (2024); Wang et al. (2024); Bai et al. (2025a); Zhu et al. (2025)). Various multimodal evaluation benchmarks, such as MME(Fu et al. (2023)), MM-Bench(Liu et al. (2023b)), SEED-Bench(Li et al. (2023)) have driven improvements in MLLM performance. With improving model capabilities, researchers are shifting toward open-world, dynamic, multi-round tasks beyond static benchmarks with fixed image-text inputs, as these better reflect real-world interaction and reasoning challenges. Among the promising approaches for evaluating models in such dynamic settings, game-based evaluation has emerged as a promising alternative, offering a more natural and interactive assessment of a model's perception and reasoning abilities.

While progress has been made in game-based MLLM benchmarks, current approaches predominantly rely on text-based (Costarelli et al. (2024); Hu et al. (2024); Duan et al. (2024)) or grid-based(Zhang et al. (2024); Wang et al. (2025); Paglieri et al. (2024)) games. In such settings, limited visual reasoning demands and static, fully textually renderable content restrict evaluation of spatial, temporal, and dynamic complexities crucial for real-world problem-solving. In contrast, the rich visual information inherent in video games presents a valuable opportunity to assess MLLMs' genuine visual reasoning capabilities, potentially addressing the limitations of current methodologies.

Figure 1: The overview of the V-MAGE benchmark, designed to evaluate vision-centric capabilities and higher-level reasoning of MLLMs across 5 free-form games with 30+ levels. V-MAGE assesses critical abilities in visual reasoning, providing a comprehensive evaluation of model performance in complex, dynamic environments.

To address the lack of vision-centric video game benchmarks, we present **V**isual-centric **M**ultiple **A**bilities **G**ame **E**valuation (**V-MAGE**) , which allows for a thorough assessment of diverse model and agent abilities within dynamic, interactive game environments and addresses key limitations in current game-based evaluations of MLLM capabilities.

With V-MAGE, we evaluate leading MLLMs across five interactive games across 30+ levels. Results highlight significant challenges posed by the dynamic visual interaction environment for MLLMs. The results reveal that current MLLMs, despite excelling in static benchmarks, lack the perception, multi-step reasoning, and task orchestration required for human-level gameplay in dynamic settings.

Our contributions are summarized as follows:

- We established V-MAGE, an interactive and visually rich evaluation framework focused on dynamic interaction and vision-centric reasoning. It also serves as a sandbox environment conducive to vision agent development.
- We evaluated various publicly available MLLMs with V-MAGE, measuring model performance with ELO scores and highlighting the significant gap between model performance and human-level proficiency on complex tasks.
- Through the evaluation results of V-MAGE, we further analyzed the reasons for the suboptimal performance of current MLLMs on video game tasks, including deficiencies in several fundamental visual capabilities, challenges in reasoning during prolonged interactions, and issues such as anchoring bias, among others.

## 2 RELATED WORK

**MLLMs and Multimodal Agents.** As LLMs(Qwen et al. (2025); Cai et al. (2024); OpenAI (2023b)) advance, MLLMs have emerged to handle multimodal tasks by integrating text and visual inputs(Zhu et al. (2025); Bai et al. (2025b); Chen et al. (2024c); Wang et al. (2024); Liu et al. (2023a)). Open-source models like InternVL and QwenVL are narrowing the gap (Chen et al. (2024b)) with closed-source models such as GPT-4o(OpenAI (2024)), and Gemini(Team (2023)).

MLLMs are evolving into interactive multimodal agents, finding applications in areas such as robotics(Driess et al. (2023)), virtual assistants(Brohan et al. (2023; 2022)), GUI automation(Xu et al. (2024); Bonatti et al. (2024); Zhang et al. (2023)), and game agents(Tan et al. (2024); Chen et al. (2024a)). These domains necessitate capabilities like sequential reasoning, memory, and adaptability, which are not adequately captured by static benchmarks.

**MLLM Benchmarks.** Classic MLLM benchmarks have focused on tasks like Visual Question Answering (VQA)(Antol et al. (2015); Goyal et al. (2017); Li et al. (2018); Marino et al. (2019)) and image captioning(Chen et al. (2015); Agrawal et al. (2019); Sidorov et al. (2020)). More comprehensive benchmarks, such as MME(Fu et al. (2023)), MMBench(Liu et al. (2023b)), SEED-Bench(Li et al. (2023)), MMMU(Yue et al. (2024)), and MM-Vet(Yu et al. (2023; 2024)), introduce broader assessments across multiple domains.

Most of these evaluations rely on structured multiple-choice and VQA-style tasks, limiting their ability to measure real-world problem-solving and interactive reasoning. Recent multimodal agent benchmarks like OSWorld(Xie et al. (2024)), Windows Agent Arena(Bonatti et al. (2024)), and COMMA(Ossowski et al. (2024)) assess broader capabilities such as open-ended tasks in real environments, OS interaction, and multi-agent collaboration.

**Evaluating MLLMs in Games.** Recent work(Tan et al. (2024); Chen et al. (2024a); Ruoss et al. (2024)) has explored MLLMs in interactive gaming environments. Meanwhile, game-based evaluation has evolved from text-only benchmarks(Costarelli et al. (2024); Hu et al. (2024); Duan et al. (2024)) to vision-integrated tests(Zhang et al. (2024); Wang et al. (2025); Paglieri et al. (2024)). However, most existing benchmarks rely on grid-based games (e.g., Tic-Tac-Toe, Chess)(Zhang et al. (2024); Wang et al. (2025); Paglieri et al. (2024)), which can be fully represented in text. These evaluations primarily test game-state recognition, and in some cases, additional visual input even confuses models, reducing performance(Paglieri et al. (2024)). Consequently, by primarily focusing on or being reducible to text-based representations, these benchmarks offer limited insights into MLLMs' visual perception and reasoning abilities, providing little guidance for improving vision-centric skills.

## 3 V-MAGE BENCHMARK

We present V-MAGE, a benchmark built on video game environments designed to evaluate the comprehensive performance of MLLMs, with a focus on vision-centric capabilities. Its defining features are as follows:

- **Vision Centric Gameplay.** Models receive only visual input, requiring pixel-level scene understanding, object tracking, and spatial-temporal reasoning. V-MAGE features continuous-space environments, allowing models to explore the almost infinite state space. Each game is designed with different difficulty levels that target various skill dimensions.
- **Extensible Evaluation Framework.** V-MAGE extends beyond model evaluation to assess agentic skills that are out-of-scope for current MLLMs. Our game-agent-model three-module evaluation pipeline allows optimizations in both MLLMs and their agent strategies.
- **Adaptive ELO-based Ranking.** V-MAGE uses a dynamic ELO system to provide a unified and interpretable metric across diverse games and difficulty levels. Unlike raw scores, which vary in scale across tasks, the ELO rating captures relative skill levels by modeling win–loss dynamics between model performances on shared levels.

### 3.1 EVALUATION PIPELINE

V-MAGE separates the game environment from the MLLM, ensuring that all information is conveyed solely through visual input. The MLLM interacts with games in a human-like manner: it observes real-time screen states and generates actions based on continuous visual interpretation, mirroring human gameplay dynamics.

As depicted in Figure 2, the system operates through iterative action cycles, comprising three sequentially linked components. The Game Module serves as the environment interface, executing game logic, capturing real-time screenshots of the current game state, and transmitting these visual frames to subsequent modules. The Agent Module functions as the perceptual-cognitive processor, integrating three critical data streams: (1) raw visual inputs from the current frame, (2) temporal context from past observations, and (3) task-specific textual prompts such as game rules. This synthesized input is structured into a multimodal format compatible with the MLLM's processing requirements. The Model Execution Phase completes the cycle, wherein the MLLM generates an action command that undergoes semantic validation by the Agent Module before being relayed back to the Game Module for environmental state updates.

To prioritize unbiased evaluation of core MLLM capabilities, V-MAGE's architecture adopts a deliberately minimalistic design, avoiding auxiliary subsystems that might obscure model performance. The framework simultaneously retains modular extensibility, allowing researchers to modify agent strategies without altering core evaluation protocols. This dual emphasis on streamlined standardization and controlled customization ensures methodological rigor in benchmarking while maintaining compatibility with specialized investigative requirements.

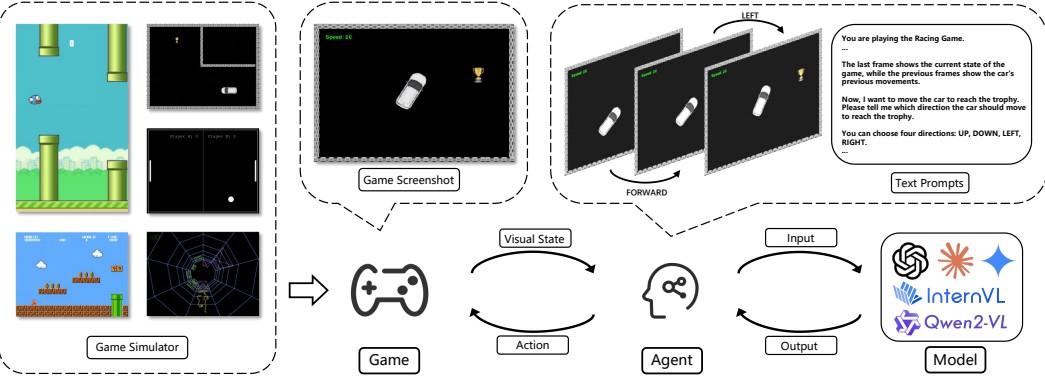

Figure 2: V-MAGE games and evaluation pipeline. V-MAGE employs five distinct games, each with several levels, to facilitate a decomposed evaluation of model performance. These games include **FlappyBird**, **Race**, **SuperMario**, **Pong** and **TempestRun**. During the evaluation process, the **Agent** module receives visual game state information directly from the **Game** module, primarily in the form of screenshots. The **Agent** module then structures these screenshots, combined with prompts containing the game rules, into the appropriate input format for MLLMs. Subsequently, the model's output is processed by the **Agent** module to generate executable actions, which are then transmitted back to the **Game** module to update the environment state.

## 3.2 GAMES AND LEVELS

V-MAGE incorporates five human-playable video games (Figure 2), each featuring 3 to 10 levels, culminating in over 30 distinct evaluation environments. In contrast to traditional grid-based evaluation setups, V-MAGE selects games based on specific principles. The games feature free-form or continuous-space visual environments, facilitating more nuanced and flexible model movement and interaction. Crucially, to effectively assess vision-centric capabilities, the game environments are designed to be **visually irreducible**. This characteristic ensures that the system state cannot be fully discretized or textually summarized without significant information loss, thereby necessitating continuous visual grounding throughout the reasoning process. Detailed discussions regarding the game selection criteria and sources are provided in Appendix C.

Existing game-based benchmarks indicate that MLLMs frequently struggle to achieve meaningful scores at standard human-level difficulties in conventional game-based benchmarks (Zhang et al. (2024); Wang et al. (2025)). This limits their discriminative power for fine-grained capability assessment and inter-model comparisons in complex tasks. To address this, V-MAGE introduces a multi-level assessment framework that evaluates models across various skill dimensions and provides granular performance diagnostics through difficulty-stratified tasks. Specifically, levels are designed for each game with gradually increasing complexity, varying control paradigms and perceptual challenges. For instance, Figure 3 illustrates the level design in Race. Detailed information on the level design for all games can be found in Appendix C.2.

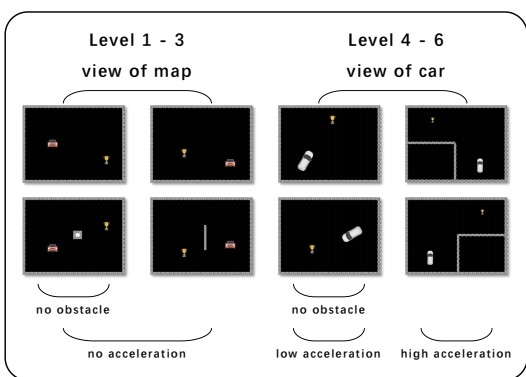

Figure 3: Race level design. Six levels progressively increase in difficulty while sharing the core objective: navigating a car to a trophy. Detailed Race level configurations are provided in Appendix Table19.

## 3.3 EVALUATION METRICS

V-MAGE employs a robust implementation of the ELO rating system to evaluate MLLMs, building on recent advancements in LLM benchmarking(Duan et al. (2024); Jiang et al. (2024)). This com-

petitive evaluation framework ensures reliable model rankings by leveraging competitive evaluation mechanisms, ensuring robustness through dynamically balanced interactions.

Games demonstrate diverse scales and difficulty thresholds. The ELO system mitigates this variability by providing a standardized metric for comparing model performance across heterogeneous environments. Moreover, ELO is inherently sensitive to performance consistency. A model that achieves a high average score through a mix of exceptional successes and frequent failures may still be ranked lower, as its instability would likely lead to more losses in direct, pairwise matchups. This allows the ELO system to reward reliable performance over erratic, high-variance gameplay, which simple score averaging might otherwise obscure.

Furthermore, it skillfully measures incremental advancement in games featuring non-linear scoring plateaus, where minor improvements can differ greatly depending on the performance range. For instance, it can differentiate between progress made from 80 to 85 and significant breakthroughs from 95 to 100.

We introduce an ELO-based ranking system to assess model performance by means of competitive pairwise comparisons. In each game level, models are randomly matched in pairs for up to 100 evaluation rounds. The outcomes are determined based on their gameplay scores and the rates of valid actions taken. Detailed mathematical formulations of the pairing mechanism, rating updates, and stabilization process are provided in Appendix D.

## 3.4 COMPARISON TO EXISTING WORKS

Humans play dynamic games using visual perception and intuitive reasoning, a process largely uncaptured by most existing MLLM game benchmarks. Many of these rely on grid-based games (Wang et al. (2025); Zhang et al. (2024)) where states are textually representable. While such benchmarks assess text-based reasoning, similar to traditional LLM tasks (Paglieri et al. (2024)), they offer limited insights into MLLMs' visual intuitive reasoning. Models often bypass genuine visual perception here, acting more like OCR converters, which hinders assessing and improving vision-centric abilities. Park et al. (2025) also employed video games as a testbed, wherein visual information remained auxiliary, and the game's state and information were accessed chiefly through text-based inputs.

In contrast to this paradigm, V-MAGE shifts the evaluation focus by embedding models in dynamic visual environments that fundamentally necessitate real-time perception and action grounded in visual input. V-MAGE deliberately adopts environments lacking rigid grid structures, where the states of characters and objects cannot be easily simplified into sparse, coordinate-based textual descriptions. This design compels models to continuously leverage the visual modality throughout the reasoning process, rather than discarding it after an initial conversion.

Furthermore, unlike benchmarks where decisions are made based on a single static frame, such as in many chess-like environments, V-MAGE requires sophisticated temporal reasoning across sequences of frames to make informed decisions, more closely mirroring human gameplay dynamics.

By shifting evaluation to more naturalistic and visually complex dynamic game environments, V-MAGE provides a more rigorous and representative test of MLLM capabilities, particularly in assessing their visual intuitive reasoning. A holistic comparison between V-MAGE and existing game benchmarks is presented in Table 1.

Table 1: The comparison of V-MAGE with existing game-based evaluation benchmarks. *Text in V-MAGE only represents the instructions for game rules and output format.

| Game Benchmarks | Game Type | Input | Reasoning Type | Level Design |
|---|---|---|---|---|
| GameBench(Costarelli et al. (2024)) | Word | Text-Only | Text Reasoning | ✗ |
| GameArena(Hu et al. (2024)) | Word | Text-Only | Text Reasoning | ✗ |
| GTBench(Duan et al. (2024)) | Word | Text-Only | Text Reasoning | ✗ |
| ING-VP(Zhang et al. (2024)) | Grid Based | Single-Image-Text | Visual Aid | ✓ |
| LVLM-Playground(Wang et al. (2025)) | Grid Based | Single-Image-Text | Visual Aid | ✓ |
| BALROG(Paglieri et al. (2024)) | Word / Grid Based | Single-Image-Text | Text / Visual Aid | ✗ |
| Orak(Park et al. (2025)) | Video | Single-Image-Text | Text / Visual Aid | ✗ |
| **V-MAGE** | **Video** | **Multi-Images-Text*** | **Vision-Centric Reasoning** | ✓ |

## 4 EXPERIMENTS

As the baseline settings for the V-MAGE benchmark, we evaluate state-of-the-art MLLMs using full-precision models under a minimal naive agent strategy(Appendix B.1.2) to ensure a fair comparison. The naive agent utilizes the most recent k frames (typically k=3) for reasoning, integrating them with reasoning history, action decisions, and game rules as input for the models. Detailed experiments settings and prompts can be found in Appendix B and C.4.

### 4.1 MAIN RESULT

Table 2: Performance comparison across different games based on the ELO ranking system. The Random baseline refers to randomly selecting actions from the predefined action space during decision-making phases. Average performance ratio, abbreviated as **Avg. Ratio**, refers to the average percentage of the model's score compared to the human baseline score.

| Model | Flappybird | Pong | Race | Supermario | Tempestrun | Avg. ELO Score | Avg. Ratio (%) |
|---|---|---|---|---|---|---|---|
| GPT-4o | **1618** | 1531 | **1716** | 1582 | 1548 | **1599** | **26.6** |
| Gemini-2.0-Flash-Thinking | 1579 | **1552** | 1648 | **1631** | 1525 | 1587 | 22.6 |
| Gemini-2.0-Flash | 1559 | 1541 | 1582 | 1561 | 1541 | 1557 | 16.7 |
| Qwen2.5-VL-72B-Instruct | 1563 | 1525 | 1624 | 1620 | **1559** | 1578 | 21.5 |
| InternVL2.5-78B | 1529 | 1539 | 1577 | 1614 | 1541 | 1560 | 19.2 |
| Qwen2-VL-72B-Instruct | 1490 | 1527 | 1587 | 1576 | 1561 | 1548 | 16.5 |
| InternVL2.5-8B | 1521 | 1530 | 1556 | 1438 | 1506 | 1510 | 12.9 |
| Qwen2.5-VL-7B-Instruct | 1469 | 1548 | 1548 | 1476 | 1494 | 1503 | 12.7 |
| Random | 1493 | 1516 | 1561 | 1490 | 1456 | 1503 | 11.0 |
| LLaVA-Onevision-Qwen2-7B | 1489 | 1495 | 1522 | 1448 | 1538 | 1498 | 13.0 |
| Keye-VL-8B-Preview | 1487 | 1518 | 1566 | 1401 | 1513 | 1497 | 13.1 |
| Qwen2-VL-7B-Instruct | 1484 | 1506 | 1529 | 1426 | 1518 | 1493 | 11.4 |
| LLaVA-v1.6-Mistral-7B | 1513 | 1512 | 1470 | 1396 | 1385 | 1455 | 9.2 |
| Phi-4-multimodal-instruct | 1441 | 1510 | 1388 | 1502 | 1389 | 1446 | 13.7 |
| LLaVA-1.5-7B | 1425 | 1304 | 1214 | 1473 | 1356 | 1354 | 14.1 |

**Scores and Rankings.** The evaluation results clearly demonstrate a performance gradient across models ranging from 7B to 70B+ parameters. This also highlights that the dynamic visual reasoning tasks we propose represent a universal challenge for current MLLMs. We note that rankings from ELO scores and the Average Ratio may occasionally differ. This discrepancy arises because the ELO system rewards performance consistency (penalizing unstable, high-variance results) and provides a more balanced, holistic assessment across games with varying score scales. In contrast, the Avg Ratio metric can be skewed when averaging across tasks with imbalanced performance levels. More detailed analysis are provided in Appendix B.2 and B.3.

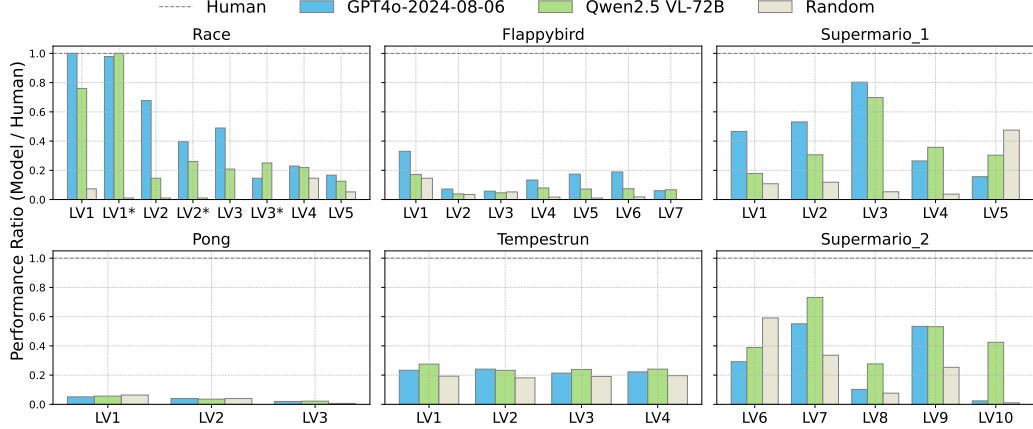

Figure 4: **The MLLM trails humans by a large margin in all six games.** The levels with an asterisk (*) represent 'no history'. Detailed performance metrics for each model across individual game levels are provided in Appendix B.2 (Tables 5-9).

**Significant Performance Gap Between MLLMs and Humans in Complex Scenes.** We invited 5 human participants to play the game in the same environment as the MLLMs and calculated their average score, which served as the baseline for human performance. Figure 4 compares the performance of leading MLLMs and human players across different game levels. The significant performance gap observed as task complexity increases underscores a critical limitation of current MLLMs in dynamic environments: they struggle to integrate real-time visual perception with the sophisticated reasoning and planning necessary for human-level gameplay, particularly in tasks demanding temporal understanding and flexible strategic adaptation.

## 4.2 FURTHER ANALYSIS

**Unit Tests for Core Visual Abilities.** We devised a unit test for vision-centric abilities by extracting foundational levels from V-MAGE. Figure 5 presents the capability profiles of various models across four core visual competencies. Scores near or below baseline suggest little effective relevant reasoning, while higher scores indicate a greater likelihood of correct reasoning. For each capability, effective reasoning was evaluated by calculating the percentage of model scores that exceeded a random baseline score on the corresponding unit test levels (as defined in Appendix F.1).

As depicted, most models substantially outperform the random baseline in **Positioning** and **Visual Grounding**, indicating a degree of proficiency in single-frame image comprehension and basic visual information perception. However, performance notably declines in **Tracking** and **Timing**, which require processing continuous frame information and executing precise spatiotemporal judgments. For the **Tracking** task, nearly all models fail to significantly surpass the random baseline.

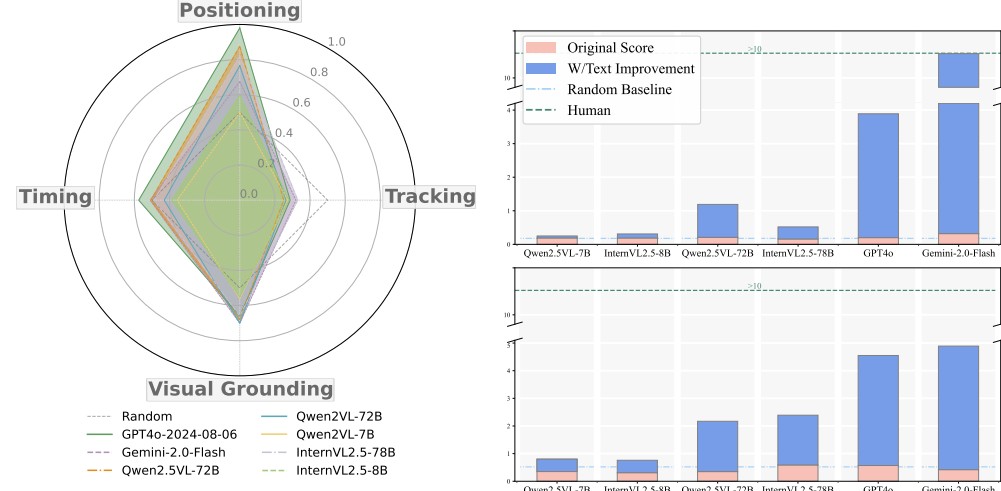

Figure 5: Capability maps of the underlying visual capabilities of each model.

Figure 6: Model performance with vs. without text information on Pong Level 2 (top) and Flappy Bird Level 3 (bottom).

**Limitations Beyond Visual Perception.** While visual perception constitutes a critical assessment dimension in V-MAGE's game tasks, our experiments revealed additional limitations and deficiencies in other aspects. To validate this, we conducted supplementary experiments in relatively simple levels providing textual descriptions of important game state information, thereby bypassing the perception process.

As shown in Figure 6, providing textual descriptions of the game state significantly improved the performance of most evaluated models, with this gain being particularly prominent in larger models such as Gemini and in games requiring precise state understanding like Pong. This notable performance increase when perception is bypassed strongly suggests that limitations in processing visual information are indeed a significant bottleneck for current MLLMs.

However, despite this substantial performance gain, the models' scores still remained considerably lower than the human baseline in most cases. This persistent gap indicates that while visual perception challenges are critical, the models' limitations extend beyond merely "seeing" the state

accurately. It highlights that significant bottlenecks also exist in the downstream processes responsible for robust interpretation of information (even when provided textually or perceived imperfectly), strategic planning, and effective action generation in complex and dynamic environments. Furthermore, the less pronounced performance improvement observed in smaller models(like Qwen2.5VL 7B) suggests that inherent limitations in their core reasoning capabilities may also act as a performance bottleneck. Check Appendix F.2 for more experimental details.

Table 3: Average number of rounds for each model to generate different responses.

| Model | Race | FlappyBird | Pong | TempestRun | Avg. |
|---|---|---|---|---|---|
| Qwen2VL 7B | 4.3 | 25.9 | 13.7 | 7.3 | 12.8 |
| Qwen2.5VL 72B | 2.3 | 19.3 | 2.6 | 5.3 | 7.4 |
| InternVL2.5 8B | 2.0 | 6.9 | 6.7 | 8.0 | 5.9 |
| InternVL2.5 78B | 6.8 | 16.0 | 2.0 | 3.0 | 7.0 |
| GPT4o | **1.0** | **1.6** | **1.0** | **1.0** | **1.1** |
| **PCC** $r$ (Avg. Rounds vs. ELO) | -0.57 | -0.71 | -0.87 | -0.72 | -0.72 |

**Anchoring Bias in Model Inference Processes.** When provided with historical information to aid reasoning in dynamic game progression, MLLMs often exhibit anchoring bias, particularly when processing similar consecutive frames. This bias manifests as an undue influence of prior inferences on current reasoning, hindering the accurate identification of subtle visual changes and unique frame details. Models tend to favor relying on historical textual descriptions over nuanced visual input, making them less sensitive to fine-grained visual updates, consequently leading to unchanged reasoning content over extended game sequences.

As shown in Table 3, models vary significantly in their responsiveness; for instance, in FlappyBird, Qwen2.5VL 72B altered its reasoning only once every 19.3 responses on average, significantly less frequently than GPT-4o (1.6 responses). The Pearson correlation coefficients (**PCC** $r$) reveal a consistent negative correlation between the average rounds to change response and ELO score across games, with an average $r$ of **-0.72**. This highlights a critical challenge in maintaining responsiveness to dynamic visual input and its direct impact on task success. To investigate the impact of settings within the pipeline (e.g., frame sampling and decision frequency) on anchoring bias, we conducted corresponding experiments, with results presented in Appendix F.3.

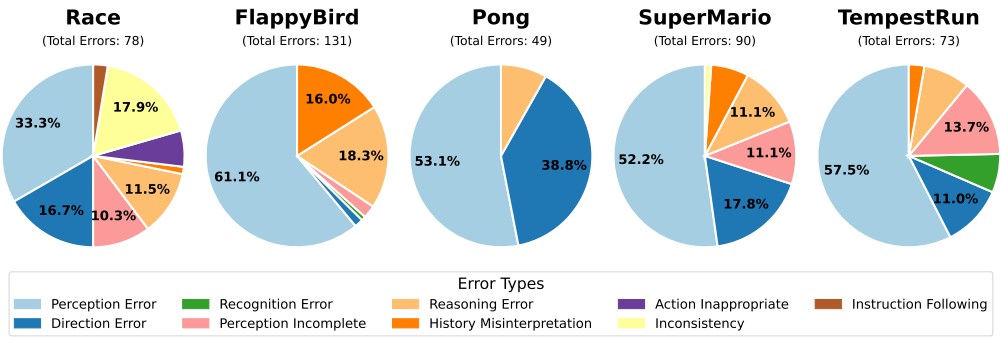

Figure 7: Error type probability distribution for GPT4o across 494 samples.

**Analysis of Model Errors in V-MAGE.** For GPT-4o's complete inputs and responses across all game levels after one to two rounds of gameplay, we uniformly sampled 494 interaction sets for manual annotation and categorized the primary error types. The visualization results depicting the distribution of these errors are presented in Figure 7. The annotation method for error distribution and the details of statistical data are provided in Appendix F.4.

The predominant error type was *perception error*, with *direction error* being particularly prevalent. In such cases, the model frequently misidentified directional cues in visual content, leading to incorrect inferences. Another major category was *reasoning error*, primarily involving logical flaws and decision-making failures, including misinterpretation of historical information (*history misinterpretation*) and selection of suboptimal actions (*action inappropriate*). Figure 8 presents case examples illustrating perception error and reasoning error.

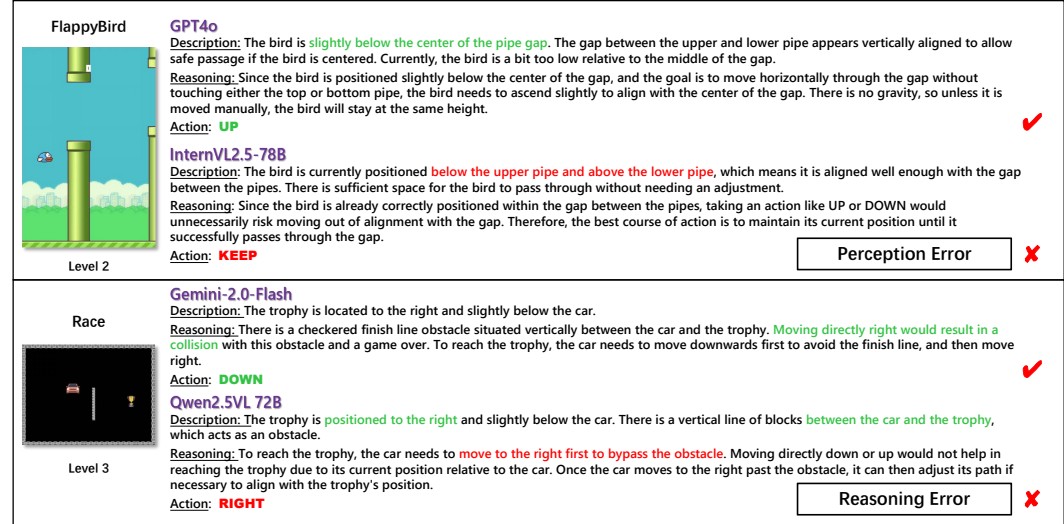

Figure 8: Case examples illustrating Perception Error and Reasoning Error in FlappyBird and Race. The FlappyBird example shows a *Perception Error* where the model misjudges the bird's vertical position relative to the pipe gap. The Race example illustrates a *Reasoning Error* where the model fails to plan a path around a obstacle between the car and the trophy, resulting in a suboptimal action.

Additionally, *perception incomplete* errors were commonly observed, where the model failed to fully extract useful information from visual inputs, resulting in partial information acquisition. *Inconsistency* errors mainly occurred in scenarios permitting multiple valid solutions, where the model exhibited unstable decision-making by frequently revising its choices, ultimately leading to timeout failures due to excessive deliberation. Notably, *instruction following* errors were virtually absent, as GPT-4o consistently adhered to the provided prompts. Additional case study analyses are documented in Appendix G.

**V-MAGE Poses Significant Challenges to MLLMs.** Unlike conventional static VQA or text-reducible grid-based benchmarks, V-MAGE necessitates real-time interaction within dynamic, vision-centric game environments, demanding human-like gameplay capabilities. The framework effectively exposes significant challenges and persistent limitations in current MLLMs. MLLMs demonstrate difficulty in processing and integrating information across sequences of dynamic frames, which impacts critical tasks like tracking, temporal reasoning, and trajectory understanding. This difficulty may contribute to anchoring bias, as models overly rely on prior inferences due to an insensitivity to subtle visual changes in consecutive frames, hindering their ability to adapt reasoning to dynamic game states. Furthermore, MLLMs demonstrate fundamental reasoning deficiencies that affect complex planning, strategic decision-making, and optimal action generation. These limitations persist even when initial visual processing challenges are mitigated, highlighting that deficiencies in the core reasoning process itself extend beyond perception.

## 5 CONCLUSION

This paper introduces V-MAGE, a pioneering game-based evaluation framework designed to assess the vision-centric capabilities of MLLMs in dynamic, interactive environments. Utilizing over 30 levels across 5 distinct games, our evaluation reveals significant limitations in current MLLMs. Specifically, models exhibit insufficient multi-image perception, leading to issues like anchoring bias, and demonstrate fundamental deficiencies in complex reasoning and strategic planning that persist even when perceptual challenges are mitigated. These findings highlight critical needs for future research, primarily in enhancing multi-frame visual processing and advancing higher-level reasoning capabilities. By systematically diagnosing these core deficiencies, V-MAGE sets a new and more demanding standard, challenging the field to develop MLLMs with robust, human-like visual intelligence for dynamic interactions.

## ETHICS STATEMENT

Research involving human subjects in this paper was limited to inviting a small number of partici-
pants (N=5) to perform tasks within the V-MAGE game environments for the purpose of establish-
ing a human performance baseline. The research has always been conducted under the guidance
and supervision of our institution's Institutional Review Board (IRB) and in full compliance with its
policies. To formally document this compliance for publication, our research protocol was reviewed
by the IRB committee. The committee confirmed the study's classification as 'minimal risk' and has
approved our research protocol.

## REPRODUCIBILITY STATEMENT

The experiments presented in this work are entirely reproducible. The code used to evaluate and
reproduce our findings is available in the supplementary materials. Appendices B and C detail
the necessary configuration environment, sample game scenes, and all input examples used in the
evaluation that could potentially influence the outcomes. We affirm that the provided resources are
sufficient to fully reproduce our experimental results.

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

APPENDIX OVERVIEW

In the appendix, we provide the following contents:

Sec A: Provides additional discussion of current limitations and directions for future work.

Sec B: Introduces experimental settings and provides detailed information on the experiments, models, and their performance.

Sec C: Delineates the game selection methodology and sources, including all level designs and prompts.

Sec D: Presents specific details of ELO-based ranking system in V-MAGE.

Sec E: Provides ablation studies on pipeline settings(e.g., frame sampling strategy, resolution, etc.)

Sec F: Provides details of supplemental experimental analyses.

Sec G: Presents some case studies.

Sec H: Provides details on miscellaneous material, including a **statement of LLM usage**(Sec. H.1) and a discussion about broader impacts.

## A    LIMITATIONS AND FUTURE WORK

**Limitations of Current Game Environments.** While V-MAGE represents a significant step towards evaluating MLLMs in interactive, dynamic, and visually complex environments that closer resemble real-world tasks, the current benchmark is still constrained by the inherent scale and complexity of the included games. Although these environments probe crucial vision-centric abilities and reasoning in continuous spaces, they cannot fully encompass the vast diversity and intricate challenges present in unrestricted real-world scenarios. This is a current boundary imposed by balancing complexity with controllability for systematic evaluation. As MLLM capabilities continue to advance and hardware performance improves, we anticipate being able to incorporate larger and more complex game environments in future iterations of V-MAGE. These future environments will be designed to offer a wider array of challenges, further pushing the limits of MLLM evaluation and narrowing the gap between simulated and real-world performance assessment.

**Future Directions in Agent Evaluation.** The primary focus of this work is the evaluation of MLLM capabilities under a predefined baseline agent strategy, which utilizes a simple historical information processing approach. However, we strongly believe that advancements in agent design and sophisticated strategies can yield substantial performance improvements within the V-MAGE evaluation environment. While our current findings highlight the limitations of the MLLM models themselves, the overall performance in complex game tasks is a product of both the MLLM's core abilities and the agent's effectiveness in perception processing, information utilization, and action generation. Therefore, a crucial direction for future work involves conducting further testing and evaluation specifically on the impact of novel and more advanced agentic approaches within V-MAGE. We look forward to exploring how enhanced agents can leverage MLLM capabilities more effectively, thereby driving further development in the field of multimodal agents.

## B    EXPERIMENTS DETAILS

### B.1    EVALUATION PIPELINE DETAILS

V-MAGE employs a three-module architecture, as illustrated in Figure 2. The specific configurations and details within each module are as follows:

#### B.1.1    GAME MODULE

The Game module encompasses the game simulators and their operational parameters. In V-MAGE, game simulators, representing environments developed with Pygame, are configured to present tasks that test specific visual and reasoning capabilities. To address potential confounding factors such as API latency and computational constraints, V-MAGE employs a frame-pausing mechanism during

model inference. This ensures that the game environment remains static while the model processes visual inputs and generates actions, effectively decoupling timing evaluation from raw inference speed.

Regarding real-time execution and inference delays, the frame-pausing mechanism ensures fairness across models. While current models may not consistently achieve real-time inference due to API or GPU limitations, V-MAGE automatically pauses the game environment during model processing. This approach isolates the model's temporal reasoning ability (strategic "when to act") from infrastructure-related delays, enabling a focused assessment of capabilities such as Timing. In V-MAGE, Timing is explicitly designed to evaluate the model's ability to choose the optimal timing of actions, not the system's response speed.

The frames per second (FPS) for all our games is 30. In our standard benchmark setting, we use a sample rate of 3. This corresponds to the model making a decision approximately every 100 milliseconds (30 FPS / 3) in the game.

### B.1.2 AGENT MODULE

Researchers can modify the agent's operational mode by adjusting the configuration file. This includes altering historical strategies, such as employing a longer history of steps or sparsely sampling more distant historical records to provide different contextual information to the models. However, for fairness in evaluation and due to resource constraints, we present results obtained using only the naive configuration as a baseline in our main findings. This allows for standardized comparisons across different MLLMs.

In the baseline agent strategy, we utilize historical information comprising three preceding screenshots and the model's reasoning outputs associated with these screenshots, combined with the current game screenshot, to form the `<history>` and `<image_current_step>` components (as shown in Sec. C.4) that are incorporated into the model's input framework.

The `<history>` component is structured as follows:

> This screenshot is `<step>` steps before the current step of the game. After this frame, your reasoning message was `<history_reasoning>`. After the action was executed, the game info was `<history_action_info>`.

Upon obtaining the model's output, the system parses the structured response format specified in the prompt to systematically extract both the cognitive reasoning log and action parameters. These extracted elements are subsequently stored and fed back to the game module through a standardized data pipeline. This process ensures consistent interpretation of the model's decision-making trajectory while maintaining synchronization with the game state.

Taking an input from the actual evaluation process in the **Tempest Run** as an example:

> Assume you are playing a PC game called 'Tempest Run'.
> You need to control a character who moves through a three-dimensional space inside a futuristic tunnel filled with various obstacles and enemies. Your goal is to navigate through the tunnel, avoid or overcome obstacles, and run as far as possible. Avoid colliding with red spikes, purple walls, or failing to deal with green enemies.
> Use the optimal combination of movements to progress through the tunnel smoothly and efficiently. Monitor the character's position relative to obstacles and react appropriately to avoid losing progress. Now, I will give you some history screenshots in the game for decision making.
> `<image>`
> This screenshot is three steps before the current step of the game. After this frame, your reasoning message was "'observation': 'The character is running forward in a narrow tunnel with blue lines forming the walls and ceiling. Red spikes are visible on the ground ahead, and green enemies are further down the tunnel. The path appears clear with no immediate obstacles or enemies to avoid.', 'reasoning': 'Maintaining the current trajectory with no action is optimal to continue moving straight towards the green enemy and prepare for a potential slide to eliminate it.', 'action': 'NONE'". After the action was excuted, the game info was "Game is running."
> ...

```
<image>
This screenshot represents the current step of the game.
The last frame shows the current state of the game, while the previous frames show the character's
previous movements.
Important notes: 1. Use JUMP to jump over red spikes on the ground. 2. Use SLIDE to duck and
kick green enemies to eliminate them.
...
You should think step by step and respond with the following format, remember to respond with
plain text without any special characters or symbols, DO NOT respond in markdown or Latex or
any other format.
Response:
Observation: ... (Describe the character's current position and nearby obstacles or enemies.)
Reasoning: ... (Think step by step and explain how you choose the action.)
Action: ... (Choose ONE of the six actions to control the character. Do NOT add any other words.)
```

### B.1.3 MODEL MODULE

The Model module is primarily responsible for model deployment and parameter control. In addition to closed-source models accessed via APIs, we deployed open-source models on an **8×V100 GPU Azure cluster** for our experiments, utilizing the **vLLM** library for efficient serving. For text output generation across all models, we standardized the decoding parameters by setting **top_p**=0.9 and **temperature**=0.8.

The following models are involved in the V-MAGE evaluation:

Table 4: Models involved in V-MAGE.

| Model | Organization | Website | Open Source |
|---|---|---|---|
| GPT-4o-2024-08-06 | OpenAI | https://openai.com/index/hello-gpt-4o/ | No |
| Gemini-2.0-Flash | Google DeepMind | https://deepmind.google/technologies/gemini/flash/ | |
| InternVL2.5-78B | Shanghai AI Lab | https://huggingface.co/OpenGVLab/InternVL2_5-78B | Yes |
| InternVL2.5-8B | | https://huggingface.co/OpenGVLab/InternVL2_5-8B | |
| Qwen2.5VL-72B-Instruct | Alibaba Cloud | https://huggingface.co/Qwen/Qwen2.5-VL-72B-Instruct | |
| Qwen2VL-72B-Instruct | | https://huggingface.co/Qwen/Qwen2-VL-72B-Instruct | |
| Qwen2VL-7B-Instruct | | https://huggingface.co/Qwen/Qwen2-VL-7B-Instruct | |

## B.2 DETAILED STATISTICS

### B.2.1 SCORE-BASED PERFORMANCE

Cross-task result analysis reveals the limitations of parameter scaling: In RaceGame Level 1 (with historical frame input), Qwen2VL showed a 429% improvement in score when scaling from 7B to 72B (from 10.43 to 55.19), reaching about 55% of the human baseline score. However, in more complex tasks such as Tempestrun Level 4, InternVL2.5-78B (199.78 points) only improved by 14.4% compared to its 8B version (174.58 points), still achieving only 25% of the human score (800 points). This suggests that parameter scaling cannot compensate for the inherent ability gap in complex dynamic tasks. The detailed scores are presented below:

### B.2.2 ADDITIONAL INDICATORS

Due to the dynamic game environments inherent in the V-MAGE evaluation process, certain levels may necessitate a considerable number of tokens during assessment. In this section, using the **Qwen2.5VL-72B** model deployed with **vLLM** as an illustrative example, we provide the statistically averaged frame counts (equal to the number of frames between two neighboring interactions multiplied by the number of interactions) and the average input and output token consumption per game round, serving as a reference.

Table 5: Performance analysis based on average scores in Race

| Level | GPT-4o | Gemini 2.0-flash | Qwen2.5VL 72B | Qwen2VL 7B | Qwen2VL 72B | InternVL2.5 8B | InternVL2.5 78B | Random | Human |
|---|---|---|---|---|---|---|---|---|---|
| Level1 | **99.99** | 35.41 | 76.01 | 10.43 | 55.19 | 28.12 | 64.56 | 7.30 | 100.00 |
| Level1 No History | 97.87 | 98.91 | 99.95 | 87.46 | 97.87 | 89.54 | **99.99** | 1.06 | 100.00 |
| Level2 | **67.68** | 7.30 | 14.59 | 0.00 | 1.06 | 3.14 | 15.63 | 1.06 | 100.00 |
| Level2 No History | **39.57** | 22.92 | 26.04 | 1.06 | 23.96 | 5.22 | 26.04 | 1.06 | 100.00 |
| Level3 | **48.94** | 5.22 | 20.84 | 4.18 | 7.30 | 6.26 | 11.47 | 0.02 | 100.00 |
| Level3 No History | 14.59 | 4.18 | **25.00** | 4.18 | 16.67 | 11.47 | 22.92 | 0.00 | 100.00 |
| Level4 | **22.92** | 6.26 | 21.88 | 0.02 | 2.10 | 0.00 | 2.10 | 14.59 | 100.00 |
| Level5 | 16.67 | 8.34 | 12.51 | 13.55 | 4.18 | **20.84** | 7.30 | 5.22 | 100.00 |

Table 6: Performance analysis based on average scores in Pong

| Level | GPT-4o | Gemini 2.0-flash | Qwen2.5VL 72B | Qwen2VL 7B | Qwen2VL 72B | InternVL2.5 8B | InternVL2.5 78B | Random | Human |
|---|---|---|---|---|---|---|---|---|---|
| Level1 | 0.51 | 0.54 | 0.56 | 0.59 | 0.54 | 0.68 | **0.77** | 0.63 | 10.00 |
| Level2 | 0.39 | **0.41** | 0.35 | 0.31 | 0.33 | 0.31 | 0.38 | 0.39 | 10.00 |
| Level3 | 0.19 | **0.32** | 0.21 | 0.18 | 0.20 | 0.18 | 0.15 | 0.06 | 10.00 |

Table 7: Performance analysis based on average scores in Supermario

| Level | GPT-4o | Gemini 2.0-flash | Qwen2.5VL 72B | Qwen2VL 7B | Qwen2VL 72B | InternVL2.5 8B | InternVL2.5 78B | Random | Human |
|---|---|---|---|---|---|---|---|---|---|
| Level10 | 18.77 | 108.30 | **339.57** | 12.51 | 29.16 | 14.61 | 80.19 | 8.36 | 800.00 |
| Level1 | **372.85** | 109.41 | 142.76 | 33.41 | 216.67 | 69.83 | 203.12 | 86.50 | 800.00 |
| Level2 | **424.92** | 127.17 | 244.78 | 102.12 | 338.47 | 102.12 | 186.48 | 94.83 | 800.00 |
| Level3 | **802.99** | 429.10 | 697.91 | 188.54 | 565.46 | 286.44 | 610.26 | 53.19 | 1000.00 |
| Level4 | 369.76 | 251.07 | **499.89** | 112.53 | 346.84 | 151.09 | 447.84 | 52.15 | 1400.00 |
| Level5 | 125.08 | 258.33 | 242.72 | 232.29 | 192.75 | 209.41 | **433.23** | 380.13 | 800.00 |
| Level6 | 233.36 | 325.96 | 311.36 | 324.96 | 296.79 | 267.70 | 344.74 | **472.78** | 800.00 |
| Level7 | 440.66 | 527.96 | **585.21** | 161.48 | 490.48 | 220.86 | 491.52 | 268.74 | 800.00 |
| Level8 | 91.75 | 211.43 | **248.96** | 51.13 | 179.20 | 76.09 | 168.74 | 68.79 | 900.00 |
| Level9 | 693.56 | 594.67 | 690.46 | 162.62 | 508.24 | 201.12 | **756.02** | 329.19 | 1300.00 |

Table 8: Performance analysis based on average scores in Flappybird

| Level | GPT-4o | Gemini 2.0-flash | Qwen2.5VL 72B | Qwen2VL 7B | Qwen2VL 72B | InternVL2.5 8B | InternVL2.5 78B | Random | Human |
|---|---|---|---|---|---|---|---|---|---|
| Level1 | **3.30** | 2.38 | 1.70 | 0.76 | 0.47 | 1.20 | 1.54 | 1.45 | 10.00 |
| Level2 | **0.71** | 0.47 | 0.38 | 0.20 | 0.12 | 0.36 | 0.39 | 0.34 | 10.00 |
| Level3 | **0.57** | 0.41 | 0.45 | 0.20 | 0.35 | 0.33 | 0.43 | 0.52 | 10.00 |
| Level4 | 1.33 | 1.50 | 0.79 | **1.52** | 0.38 | 1.43 | 0.64 | 0.16 | 10.00 |
| Level5 | **1.74** | 1.38 | 0.71 | 1.44 | 0.51 | 1.20 | 0.49 | 0.10 | 10.00 |
| Level6 | **1.88** | 1.05 | 0.73 | 1.62 | 0.56 | 1.14 | 0.66 | 0.17 | 10.00 |
| Level7 | 0.60 | 0.07 | **0.66** | 0.03 | 0.14 | 0.00 | 0.13 | 0.00 | 10.00 |

Table 9: Performance analysis based on average scores in Tempestrun

| Level | GPT-4o | Gemini 2.0-flash | Qwen2.5VL 72B | Qwen2VL 7B | Qwen2VL 72B | InternVL2.5 8B | InternVL2.5 78B | Random | Human |
|-------|--------|------------------|---------------|------------|-------------|----------------|-----------------|--------|-------|
| Level1 | 466.25 | 478.35 | **549.98** | 446.92 | 519.22 | 444.71 | 475.22 | 385.72 | 2000.00 |
| Level2 | 361.44 | 356.05 | 349.06 | 352.76 | **370.13** | 327.38 | 333.37 | 271.65 | 1500.00 |
| Level3 | 213.73 | 197.91 | **238.74** | 208.75 | 220.21 | 197.71 | 216.64 | 190.71 | 1000.00 |
| Level4 | 177.60 | **201.67** | 192.79 | 182.91 | 195.19 | 174.58 | 199.78 | 157.17 | 800.00 |

Depending on the differences in the models and the randomness of the games and reasoning, as well as other further experiments, the full research project may require **more** compute than the experiments reported here. The time of execution of the experiment depends on the network environment and computational power.

Table 10: **SuperMario** Average Frames and Tokens Consumed

| Metric | Level 1 | Level 2 | Level 3 | Level 4 | Level 5 | Level 6 | Level 7 | Level 8 | Level 9 | Level 10 | All |
|--------|---------|---------|---------|---------|---------|---------|---------|---------|---------|----------|-----|
| Average Frames | 400 | 655.56 | 1000 | 641.2 | 234.00 | 300 | 300 | 148.06 | 504.35 | 950.45 | 5133.62 |
| Average Prompt Tokens | 150004.78 | 253457.38 | 379649.92 | 266125.09 | 100595.30 | 112241.57 | 111288.00 | 54231.22 | 192642.31 | 361208.32 | 1981443.89 |
| Average Completion Tokens | 10054.86 | 18437.53 | 25428.29 | 18242.65 | 7075.53 | 7639.30 | 7314.46 | 3710.36 | 13492.79 | 24197.61 | 135593.38 |

Table 11: **Race** Average Frames and Tokens Consumed

| Metric | Level 1 No History | Level 2 No History | Level 3 No History | Level 1 | Level 2 | Level 3 | Level 4 | Level 5 | Level 6 | All |
|--------|--------------------|--------------------|--------------------|---------|---------|---------|---------|---------|---------|-----|
| Average Frames | 12.66 | 15.39 | 16.66 | 29.20 | 30.69 | 32.14 | 58.07 | 98.06 | 32.46 | 325.33 |
| Average Prompt Tokens | 1738.83 | 2309.85 | 2562.15 | 12317.22 | 14044.87 | 14934.32 | 31164.03 | 54346.81 | 17399.59 | 255136.23 |
| Average Completion Tokens | 275.35 | 531.41 | 595.91 | 693.11 | 937.66 | 1060.77 | 2243.55 | 3898.42 | 1517.55 | 20798.72 |

Table 12: **FlappyBird** Average Frames and Tokens Consumed

| Metric | Level 1 | Level 2 | Level 3 | Level 4 | Level 5 | Level 6 | Level 7 | All |
|--------|---------|---------|---------|---------|---------|---------|---------|-----|
| Average Frames | 224.73 | 133.34 | 76.49 | 153.11 | 153.87 | 152 | 143.28 | 1036.82 |
| Average Prompt Tokens | 98273.78 | 57332.59 | 32326.91 | 65853.39 | 66500.27 | 65322.27 | 56528.63 | 442137.84 |
| Average Completion Tokens | 9979.17 | 5772.39 | 3319.76 | 7142.32 | 7309.31 | 7082.73 | 6259.62 | 46865.30 |

Table 13: **TempestRun** and **PongGame** Average Frames and Tokens Consumed

| Metric | Level 1 | Level 2 | Level 3 | Level 4 | All | Level 1 | Level 2 | Level 3 | All |
|--------|---------|---------|---------|---------|-----|---------|---------|---------|-----|
| Average Frames | 173.58 | 92.70 | 38.98 | 28.72 | 333.98 | 221.79 | 83.98 | 47.00 | 352.77 |
| Average Prompt Tokens | 108291.56 | 57096.18 | 33218.84 | 22874.80 | 237820.07 | 136254.76 | 50056.53 | 26981.30 | 213292.59 |
| Average Completion Tokens | 7000.84 | 3799.28 | 2316.48 | 1619.33 | 15942.53 | 10998.40 | 4064.67 | 2208.39 | 17271.46 |

### B.2.3 MORE MODELS

In addition to the models mentioned in Table B.1.3, we also evaluated the more recent **Claude-3.7-sonnet**[1]. Due to budgetary constraints, we were only able to conduct approximately 5 to 10 rounds of testing. When we included Claude-3.7-sonnet in the ELO calculation, the results are shown in Table 14.

Table 14: Performance comparison across different games based on the elo ranking system.

| Model | Pong | Race | Flappybird | Tempestrun | SuperMario | Average |
|---|---|---|---|---|---|---|
| *Closed-Source Models* | | | | | | |
| *Claude-3.7-sonnet* | **1607** | **1626** | **1578** | 1513 | **1601** | **1591** |
| *GPT-4o* | 1487 | 1582 | 1573 | 1514 | 1512 | 1526 |
| *Gemini-2.0-Flash* (Thinking) | 1518 | 1550 | 1533 | 1498 | 1588 | 1553 |
| *Gemini-2.0-Flash* | 1502 | 1498 | 1513 | 1515 | 1512 | 1510 |
| *Open-Source Models* | | | | | | |
| *Qwen2VL-7B* | 1464 | 1417 | 1438 | 1488 | 1361 | 1412 |
| *Qwen2VL-72B* | 1479 | 1527 | 1521 | 1530 | 1580 | 1543 |
| *Qwen2.5VL-72B* | 1485 | 1489 | 1440 | **1531** | 1509 | 1494 |
| *InternVL2.5-8B* | 1489 | 1442 | 1481 | 1471 | 1372 | 1428 |
| *InternVL2.5-78B* | 1492 | 1447 | 1481 | 1514 | 1546 | 1510 |
| *Baseline* | | | | | | |
| Random | 1477 | 1424 | 1440 | 1424 | 1419 | 1431 |

It is important to note that these results may be biased because the number of evaluation rounds is incomplete compared to other models, which is why the **Claude-3.7-sonnet** model was not included in the main results discussed.

Nevertheless, based on the current findings, it is one of the best-performing models on V-MAGE to date.

### B.3 INCONSISTENCY BETWEEN ELO AND PERFORMANCE RATIO RANKINGS

As shown in Table 2 in the main text, ELO and Performance Ratio sometimes do not align in rankings.

We examine **Keye-VL-8B-Preview** and **Qwen2.5-VL-7B-Instruct**, with **LLaVA-v1.6-Mistral-7B** as a control.

Table 15: Elo Scores and Average Performance Ratios (E/R) Across Games.

| | Race(E/R) | SuperMario(E/R) | Pong(E/R) | FlappyBird(E/R) | TempestRun(E/R) |
|---|---|---|---|---|---|
| Qwen2.5-VL-7B-Instruct | 1487/0.120 | 1459/0.239 | **1503/0.035** | 1431/0.030 | 1485/0.210 |
| Keye-VL-8B-Preview | 1487/0.118 | 1430/0.217 | **1495/0.039** | 1450/0.044 | 1513/0.239 |
| LLaVA-v1.6-Mistral-7B | 1462/0.051 | 1374/0.127 | **1494/0.035** | 1489/0.077 | 1379/0.169 |

In Pong, Qwen shows higher ELO but lower average ratio. We analyzed level-wise scores and variances to explore this. The variance is calculated as: variance $= \frac{\sum_{i=1}^{n}(\text{score}_i - \bar{\text{score}})^2}{n}$.

---

[1]Anthropic, https://www.anthropic.com/claude/sonnet

Table 16: Pong Scores by Level (Avg: average score, Var: variance).

|  | L1 Avg | L1 Var | L2 Avg | L2 Var | L3 Avg | L3 Var |
|---|---|---|---|---|---|---|
| Qwen2.5-VL-7B-Instruct | 0.48 | 0.50 | 0.37 | 0.25 | 0.20 | 0.18 |
| Keye-VL-8B-Preview | 0.68 | 0.67 | 0.26 | 0.33 | 0.23 | 0.36 |
| LLaVA-v1.6-Mistral-7B | 0.48 | 0.58 | 0.29 | 0.26 | 0.29 | 0.34 |

Keye's higher variance across all Pong levels indicates unstable performance, where high-scoring outliers mask frequent weak results. In the ELO system, this instability leads to more losses against a consistent opponent, resulting in a lower rating despite a competitive average score.

Additionally, current models perform poorly on Pong, with ratios tightly clustered in the 0–10% range. When calculating the performance ratio by averaging across games, minor differences in Pong (3.5% vs. 3.9%) are overshadowed by larger gaps in other games(21% vs. 24%). The ELO system, in contrast, is based on the aggregate outcomes of all pairwise matchups. The ELO rating boost from a consistent pattern of wins in Pong is just as significant as from wins in any other game. This demonstrates that ELO is more robust in **fairly** assessing a model's holistic capabilities across tasks with imbalanced performance levels.

We also observed that in terms of Response Format Accuracy, GPT-4o is slightly lower than Gemini model (by 0.04%), and InternVL2.5-78B is slightly lower than Qwen2-VL-72B (by 0.25%). This may also be an influencing factor.

## C  GAMES IN V-MAGE

### C.1  PRINCIPLES AND STANDARDS FOR GAME SELECTION

**Simplified and unrealistic considerations.**  While the simplified visuals in these games differ from real-world scenes, empirical evidence demonstrate that MLLMs comprehend core game semantics (objectives, rules, entities) despite stylistic simplifications. Performance limitations primarily emerge from perceptual inaccuracies (e.g., dynamic object tracking) and multi-step reasoning deficiencies rather than misinterpretation. V-MAGE therefore focuses more on **precise evaluation** than **visual realism** to drive targeted improvements in visual reasoning.

**Selection criteria.**  The five games in V-MAGE share critical characteristics (e.g., non-textual state representation, free-form gameplay, and continuous-space environments) while offering diverse challenges.

Our current minimal set covers four **2D** game types through this matrix:

Table 17: 2D Game Taxonomy in V-MAGE

|                    | XY-axis   | XZ-axis    |
| ------------------ | --------- | ---------- |
| **Linear Process** | PongGame  | FlappyBird |
| **Open Planning**  | RaceGame  | SuperMario |

The Linear Process implies that the game's progression is, to some extent, enforced. In PongGame, the ball's movement direction is determined by the game environment, requiring the model to move paddles on both sides to catch the ball, while in FlappyBird, the forward movement of the bird is compulsory, with the model controlling the height to navigate through pipes. OpenPlanning, in contrast, is relatively more open-ended. In RaceGame, the model can freely control the car's movement and direction on a plane to reach a trophy. In SuperMario, the model can move and jump in a relatively open environment to collect rewards and earn points.

For **3D** environments, we selected **Tempest Run** for its streamlined visual elements.

V-MAGE's flexible framework allows seamless integration of new PyGame-based environments. For instance, Tempest Run (one of our five games) was sourced from PyWeek[3], a community-driven game jam with thousands of open-source entries. This demonstrates our framework's capacity to incorporate externally developed, human-designed games. We provide APIs to wrap new games into V-MAGE's evaluation pipeline. This allows researchers to easily integrate additional games.

We will continue expanding the benchmark with more diverse titles that meet our selection criteria (e.g., Player vs Player (PVP) games) and will open-source both the codebase and detailed documentation to facilitate community contributions.

### C.2  DESIGN AND IMPLEMENTATION

As previously mentioned, V-MAGE enhances the diversity of the evaluation environment by expanding it through level design. Tables in this section detail the settings, rewards, and design objectives for each game's levels. For more comprehensive visual comparisons and prompt information, please refer to Appendix C.4.

**Race Game** is a skill-based driving game where the objective is to control a car through a maze-like track to reach the trophy while avoiding obstacles. The car is represented by a red or white vehicle with a visible front and back, while the trophy is shown as a golden cup icon. The surrounding white-lined boundaries represent walls, which the car must avoid. For the overall observation and action spaces of the game, including the task and reward definitions, please refer to Table 18.

Each level in Race has a different set of rules and challenges. As presented in Table 19, we manually designed six levels. Levels 1–3 use a *map-view perspective*("map" view), where models adjust absolute coordinates. The four types of movement operations directly translate the vehicle on the map

Table 18: Race Environment Details (* means potentially requires observation).

| Observation space | Action Space | Task | Reward |
|---|---|---|---|
| Car Position Trophy Position Obstacle Position* Speed* Acceleration* Facing Angle* | UP, DOWN, LEFT, RIGHT | Move the car to reach the trophy | +100 Success +0 Timeout +0 Destroyed |

according to the direction of action. Conversely, Levels 4–6 shift to a *first-person perspective*("car" view), the observation is centered on the vehicle, and movements are performed based on the vehicle's perspective, Requiring real-time interpretation of velocity vectors and acceleration constraints. Furthermore, acceleration is introduced in the high-difficulty levels, which further expands the observation space. This requires the model to extract more information from the visual input, including current speed and acceleration, in order to perform rational reasoning.

Table 19: Race Level Configurations

| Level | View | bstacle | Initial Direction | Acceleration | Max Rounds | Sample Frames |
|---|---|---|---|---|---|---|
| 1 | Map | No | - | No | 100 | 1 |
| 2 | Map | Yes | - | No | 150 | 1 |
| 3 | Map | Yes | - | No | 150 | 1 |
| 4 | Car | No | Vertical(up) | Low | 150 | 3 |
| 5 | Car | No | Horizontal(random) | Mid | 150 | 3 |
| 6 | Car | Yes | Vertical(up) | Mid | 150 | 1 |

**SuperMario** is a two-dimensional side-scrolling platformer where the player controls the character Mario navigating through environments populated with various platforms, enemies, and obstacles. The goal is to traverse the level, collect coins, evade or defeat enemies, and reach the flagpole at the stage's conclusion. Players must avoid falling off platforms, colliding with enemies, or being struck by obstacles. Successful gameplay involves employing optimal movement combinations for smooth and efficient progression, alongside monitoring Mario's position relative to environmental elements. Task and reward defination is shown in Table 20.

Table 20: SuperMario Environment Details.

| Observation space | Action Space | Task | Reward |
|---|---|---|---|
| Mario Position Platforms Position Enemies Position Obstacles Position | UP, UP+LEFT, UP+RIGHT, LEFT, RIGHT, NONE | Collect coins and evade or defeat enemies | +100 for collecting a coin +100 for defeating a Goomba Penalties for falling or collisions |

Table 21: SuperMario Level Configurations

| Level | Enemy count | Coin Count | CoinBox Count | Max Rounds | Gameplay |
|---|---|---|---|---|---|
| 1 | 0 | 6 | 2 | 400 | Common |
| 2 | 2 | 6 | 2 | 1000 | Common |
| 3 | 0 | 17 | 4 | 1000 | Long History (Two ways) |
| 4 | 2 | 17 | 4 | 1000 | Long History (Two ways) |
| 5 | 3 | 8 | 0 | 300 | Left or Right |
| 6 | 0 | 13 | 0 | 300 | Left or Right |
| 7 | 0 | 8 | 0 | 300 | Left or Right |
| 8 | 0 | 12 | 0 | 1000 | Jump Only |
| 9 | 5 | 8 | 0 | 1000 | Jump and Enemy |
| 10 | 12 | 0 | 9 | 5000 | Classic W1-1 |

To provide a comprehensive evaluation of MLLMs' visual reasoning and planning capabilities, SuperMario features ten levels with configurations detailed in Table 21. These levels vary in enemy count, coin and coinbox quantities, maximum allowed rounds, and specific gameplay mechanics or focuses. Of these, level 10 serves as a standard human-difficulty benchmark, providing a 1:1 replica of the original Super Mario game's World 1-1 stage.

**Flappy Bird** is a widely recognized side-scrolling mobile game serving as a common benchmark in reinforcement learning. The objective is to control a bird's vertical movement to navigate through a continuous series of horizontal gaps within vertically oriented pipes. Successful traversal of a pipe pair increments the player's score, while collision with any pipe or the ground constitutes a terminal state, ending the game. The game mechanic involves a constant downward gravitational pull, counteracted by discrete upward 'flaps' initiated by the player.

Table 22: Flappy Bird Environment Details (* means only available at certain levels).

| Observation space | Action Space | Task | Reward |
|---|---|---|---|
| Bird Position | UP | Maneuver the bird to avoid | +1 per pipe pair passed |
| Bird Velocity | NONE | hitting the pipes | +0 Collision |
| Next Pipe Distance | DOWN* | | |
| Gap VerticalPosition | KEEP* | | |

Given the high difficulty of human-standard levels for MLLMs, we designed seven levels with progressive difficulty. Specifically, as presented in Table 23, levels 1-3 constitute a simplified game environment where the gravity factor is removed, and height is controlled via UP and DOWN actions to navigate through the pipes. Levels 4-6 are based on the standard difficulty but incorporate a 'KEEP' option, enabling the model to maintain the bird's altitude through this action. Within the same difficulty tier, levels are differentiated by varying the bird's forward speed and the pipe gap width. Level 7 represents the standard human game difficulty, retaining the original game settings.

Table 23: FlappyBird Level Configurations

| Level | Gravity | Availability of "DOWN" | Availability of "KEEP" | Others |
|---|---|---|---|---|
| 1 - 3 | No | Yes | Yes | Distinguished by gap |
| 4 - 6 | Yes | No | Yes | clearance and speed |
| 7 | Yes | No | No | Human Standard |

**Pong** Game is a classic two-player adversarial game. The objective is to control the paddles on the left and right sides of the screen to return the ball, preventing it from passing one's own paddle while simultaneously attempting to make the ball pass the opponent's paddle. One point is awarded to the player for each successful return of the ball. The final score is the sum of both players' scores. Task and reward defination is shown in Table 24.

Table 24: Pong Game Environment Details.

| Observation space | Action Space | Task | Reward |
|---|---|---|---|
| Left Paddle Position | LEFTUP | Track the ball's trajectory | +1 per successful hit |
| Right Paddle Position | LEFTDOWN | and maneuver the left and | +0 if ball passes paddle |
| Ball Position | RIGHTUP | right paddles to intercept | |
| Ball Trajectory | RIGHTDOWN | and return the ball. | |
| | NONE | | |

Table 25: Pong Game Level Configurations

| Level | Paddle Width | Ball Speed | Ball Size | Others |
|---|---|---|---|---|
| 1 | Big | Slow | Big | Ball initial position |
| 2 | Mid | Mid | Mid | randomly changes. |
| 3 | Small | Fast | Small | |

Considering the challenges MLLMs face in tracking and temporal tasks, we designed levels with varying difficulty. As shown in Table 25, difficulty for Levels 1-3 is differentiated by adjusting the

paddle width and the speed of the ball. Within the same level, the initial position of the ball is randomized, but the relative difficulty remains consistent.

Table 26: Tempest Run Environment Details.

| Observation space | Action Space | Task | Reward |
|---|---|---|---|
| Current Character State Nearby Obstacles Position Nearby Obstacles Type Visual Information Quantity | JUMP, LEFT, RIGHT, SLIDE, RISE, NONE | Perform corresponding actions to avoid or destroy obstacles. | Score increases with distance run. |

**Tempest Run** is a third-person perspective 3D runner game where the player controls a character moving within a futuristic tunnel filled with various obstacles and enemies. The objective is to navigate through the tunnel, avoiding or overcoming impediments, and to run as far as possible. Players must specifically avoid colliding with red spikes, purple walls, or failing to manage green enemies. Successful gameplay requires employing optimal combinations of movements for smooth and efficient tunnel traversal, alongside monitoring the character's position relative to obstacles and reacting appropriately. Task and reward defination is shown in Table 26.

To evaluate MLLMs' visual comprehension and reactive capabilities within a dynamic 3D environment, Tempest Run includes four levels of varying difficulty. As outlined in Table 27, Levels 1-4 are primarily differentiated by parameters including role speed, cell length (denoting the distance between environmental segments), and random rate (controlling obstacle spawning frequency). These parameters collectively influence the pace of barrier generation and the overall visual complexity of the tunnel environment, thereby varying the level of challenge. Within the same level, the positioning of environmental elements is randomized, while maintaining consistent relative difficulty.

Table 27: Tempest Run Level Configurations

| Level | Role Speed | Cell Length | Random Rate | Others |
|---|---|---|---|---|
| 1 | Slow | Large | Low | Environmental elements initial |
| 2 | Medium | Medium | Medium-Low | positions randomly change. |
| 3 | Fast | Small | Medium-High | |
| 4 | Very Fast | Small | High | |

### C.3 ORIGINAL SOURCES

Thanks to the open-source community, we are able to leverage existing game codebases to build our benchmark. Here are the codebases we used:

Table 28: Game Codebase Sources

| Game | Codebase |
|---|---|
| **Race** | `https://github.com/tdostilio/Race_Game` |
| **FlappyBird** | `https://github.com/agneay/pygame-projects/tree/master/Flappy%20Bird` |
| **Pong** | `https://github.com/pyGuru123/Python-Games/tree/master/Pong` |
| **SuperMario** | `https://github.com/mx0c/super-mario-python` |
| **Tempest Run** | `https://github.com/davidpendergast/pygame-summer-team-jam` |

In most cases, the original codebases lacked comprehensive difficulty settings and level designs suitable for systematic evaluation. We therefore modified the default human-oriented game configurations to adapt them for benchmarking purposes, while meticulously designing a diverse set of challenging levels to ensure rigorous assessment.

## C.4 GAMES AND PROMPTS

All the games have been modified based on publicly available code. The detailed design is provided below:

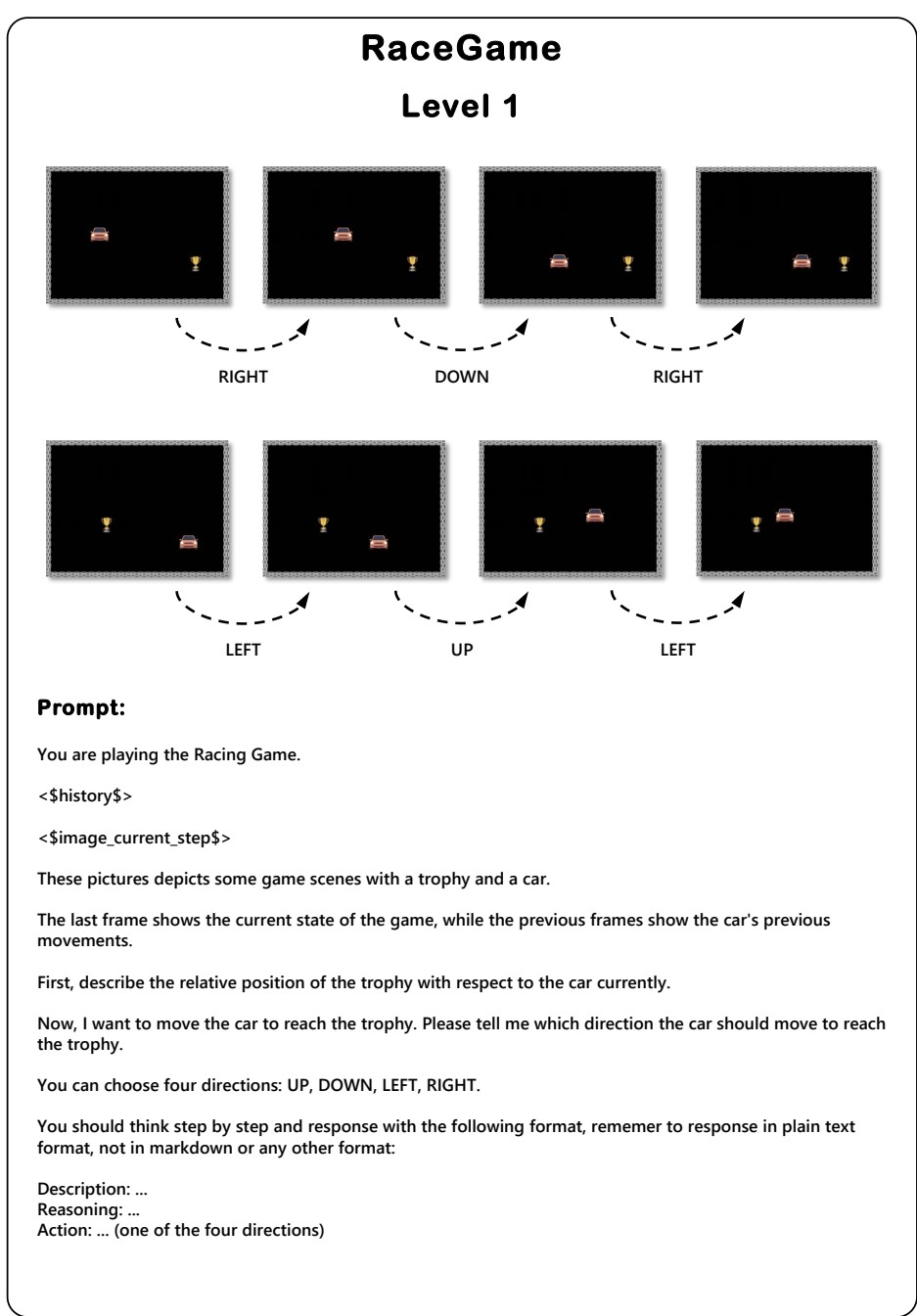

Figure 9: **RaceGame Level 1: Level Design and Prompt Overview.** The images showcase the scene from Level 1, illustrating the level design and corresponding prompt. Elements in the same level will randomly change their initial positions while maintaining consistent relative difficulty.

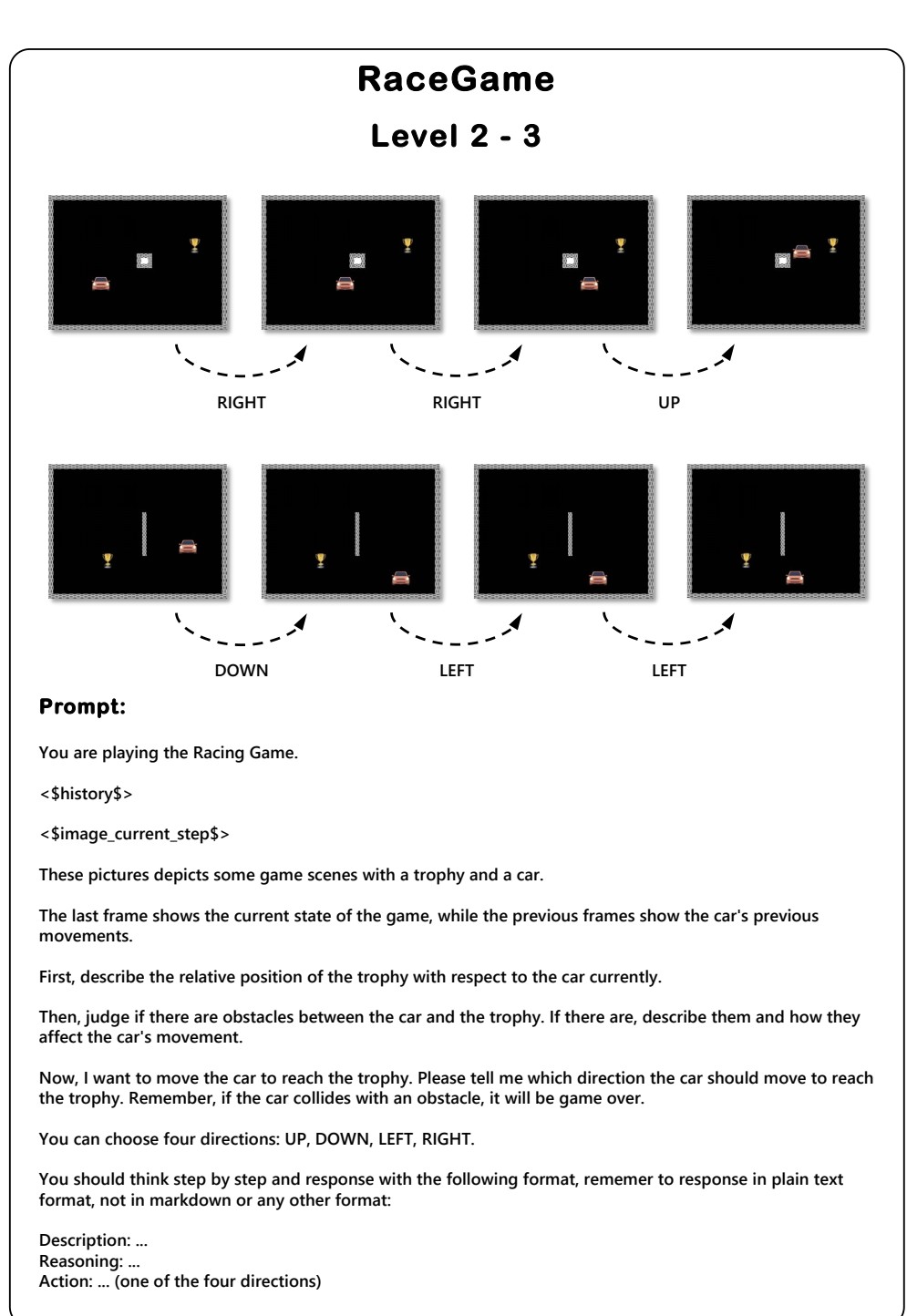

Figure 10: **RaceGame Level 2-3: Level Design and Prompt Overview.** The images showcase the scene from Level 2-3, illustrating the level design and corresponding prompt. Elements in the scene will randomly change their initial positions while maintaining consistent relative difficulty.

**RaceGame**

**Level 1 - 3 No History**

**Prompt:**

You are playing the Racing Game.

<$image_current_step$>

These pictures depicts some game scenes with a trophy and a car.

The last frame shows the current state of the game, while the previous frames show the car's previous movements.

First, describe the relative position of the trophy with respect to the car currently.

(Only for Level 2/3)
Then, judge if there are obstacles between the car and the trophy. If there are, describe them and how they affect the car's movement.

Now, I want to move the car to reach the trophy. Please tell me which direction the car should move to reach the trophy. Remember, if the car collides with an obstacle, it will be game over.

You can choose four directions: UP, DOWN, LEFT, RIGHT.

You should think step by step and response with the following format, rememer to response in plain text format, not in markdown or any other format:

Description: ...
Reasoning: ...
Action: ... (one of the four directions)

Figure 11: **RaceGame Level 1-3 No History: Level Design and Prompt Overview.** The images showcase the scene from Level 1-3 No History, illustrating the level design and corresponding prompt. Elements in the scene will randomly change their initial positions while maintaining consistent relative difficulty. Same as the original levels except the input sequence has been changed to the single image.

Figure 12: **RaceGame Level 4: Level Design and Prompt Overview.** The images showcase the scene from Level 4, illustrating the level design and corresponding prompt. Elements in the same level will randomly change their initial positions while maintaining consistent relative difficulty.

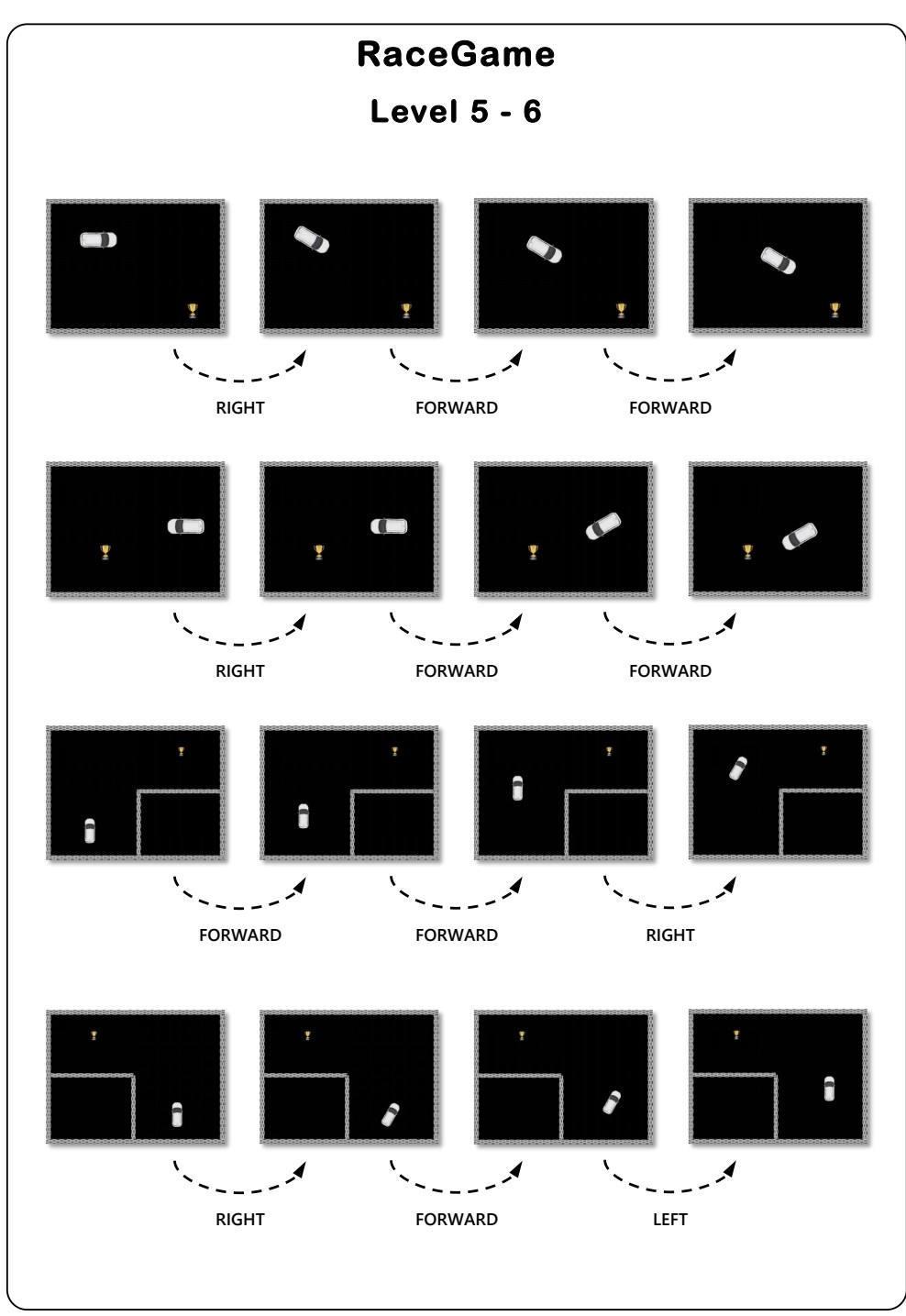

Figure 13: **RaceGame Level 5-6: Level Design and Prompt Overview.** The images showcase the scene from Level 5-6, illustrating the level design and corresponding prompt. Elements in the same level will randomly change their initial positions while maintaining consistent relative difficulty. **The prompt is the same as in Level 4.**

**SuperMario**

**Level 1**

UP+RIGHT          RIGHT          UP+RIGHT

**Prompt:**

Assume you are playing a PC game called 'Super Mario'.

You need to control Mario, who can move in a two-dimensional space consisting of various platforms, enemies, and obstacles. Your goal is to navigate through the level, collect coins, avoid or defeat enemies, and reach the flagpole at the end of the stage. Avoid falling off the platforms, colliding with enemies, or being hit by obstacles.
Use the optimal combination of movements to progress through the level smoothly and efficiently. Monitor Mario's position relative to obstacles, platforms, and enemies.

<$history$>

<$image_current_step$>
This screenshot represents the current step of the game.

The last frame shows the current state of the game, while the previous frames show Mario's previous movements.

**Important notes:**
1. Mario can jump (actions involving UP) only if he is on the ground or on a solid surface like a platform or pipe.
2. If Mario is in mid-air, he can only use LEFT or RIGHT to adjust his position, or NONE to continue falling or moving with momentum.

You can make six types of actions to control Mario:
1. UP: Makes Mario jump upward (only available when Mario is on the ground or solid platforms).
2. LEFT: Moves Mario left.
3. RIGHT: Moves Mario right.
4. UP+LEFT: Makes Mario jump upward and left simultaneously (only available when on the ground or solid platforms).
5. UP+RIGHT: Makes Mario jump upward and right simultaneously (only available when on the ground or solid platforms).
6. NONE: No new action is performed; Mario continues to be affected by gravity (if airborne) or momentum from previous movements.

Note that DOWN has no effect and cannot be used, so you should never attempt to use it.

You should think step by step and respond with the following format, remember to respond with plain text without any special characters or symbols, DO NOT respond in markdown or Latex format.

Response:

Observation: ... (Describe Mario's current position, nearby platforms, enemies, and obstacles.)
Reasoning: ... (Think step by step and explain how you choose the action.)
Action: ... (Choose one of the six actions to control Mario. Do NOT add any other words.)

Figure 14: **SuperMario Level 1: Level Design and Prompt Overview.** The images showcase the scene from Level 1, illustrating the level design and corresponding prompt.

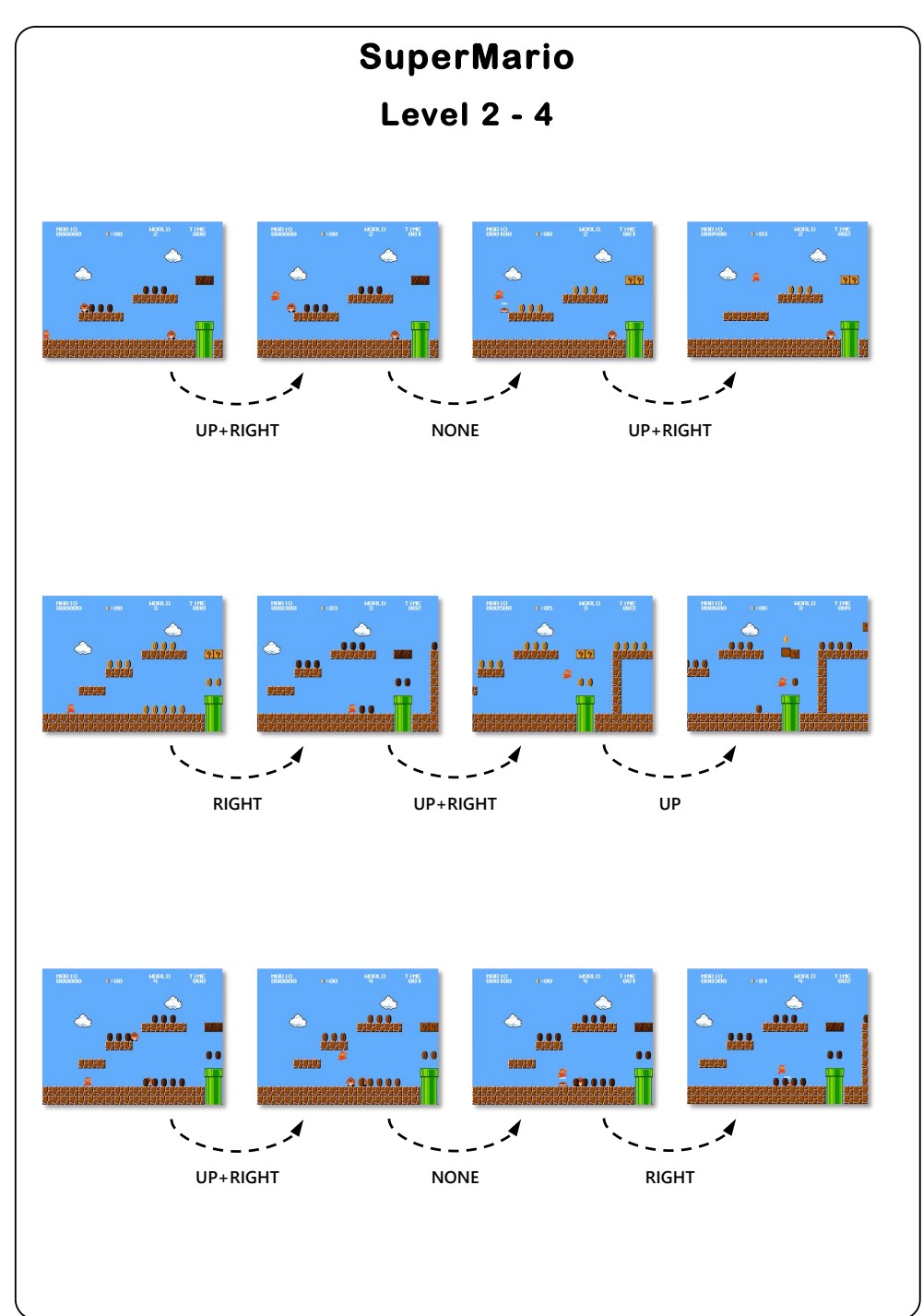

Figure 15: **SuperMario Level 2-4: Level Design and Prompt Overview.** The images showcase the scene from Level 2-4, illustrating the level design and corresponding prompt.**The prompt is the same as in Level 4.**

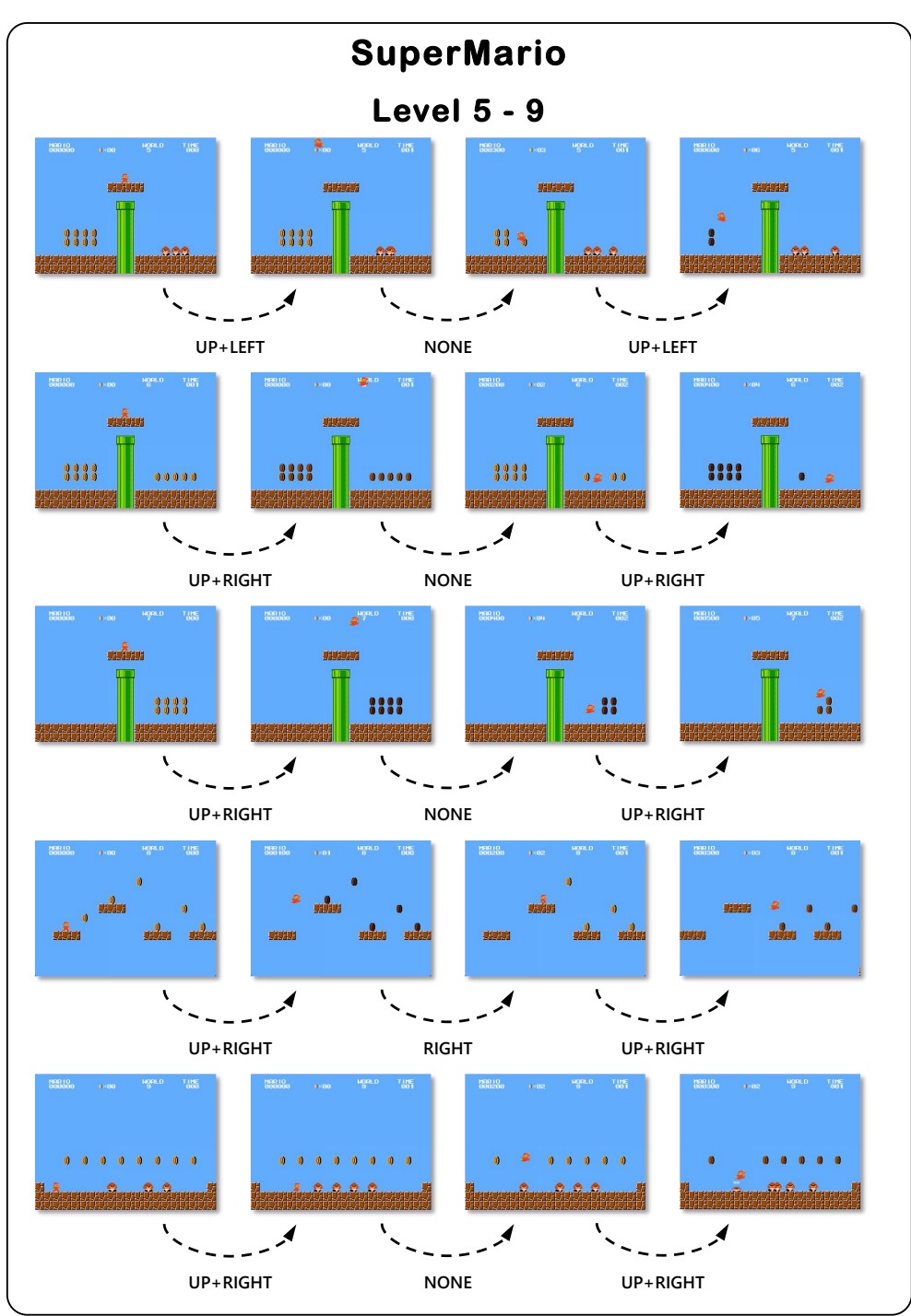

Figure 16: **SuperMario Level 5-9: Level Design and Prompt Overview.** The images showcase the scene from Level 5-9, illustrating the level design and corresponding prompt.**The prompt is the same as in Level 4.**

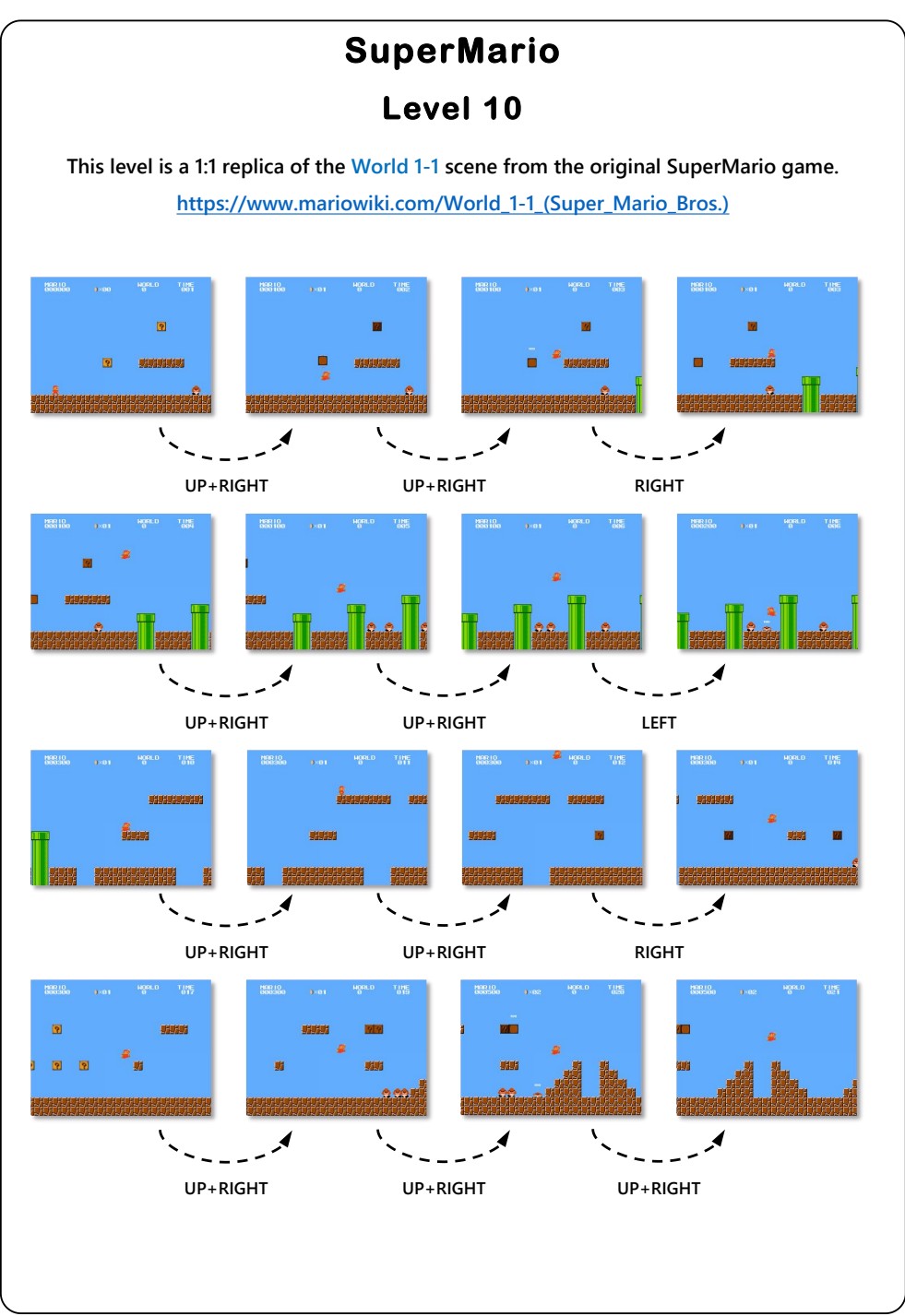

Figure 17: **SuperMario Level 10 (Standard Level): Level Design and Prompt Overview.** The images showcase the scene from Level 10, illustrating the level design and corresponding prompt. This is The standard level that matches the difficulty of the human game. **The prompt is the same as in Level 4.**

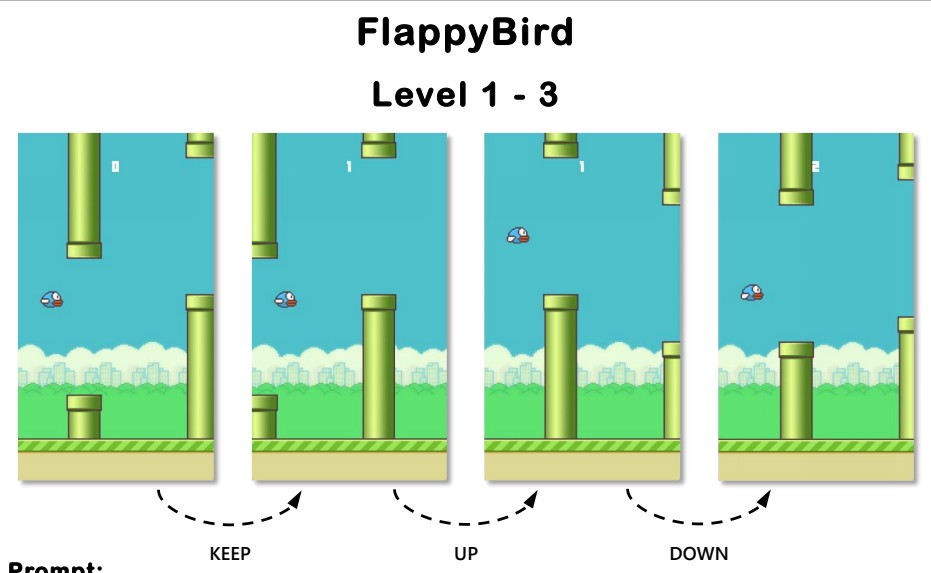

Figure 18: **FlappyBird Level 1-3: Level Design and Prompt Overview.** The images showcase the scene from Level 1, illustrating the level design and corresponding prompt. Levels are differentiated by the pipe gap width and the bird's forward speed. Elements in the same level will randomly change their initial positions while maintaining consistent relative difficulty.

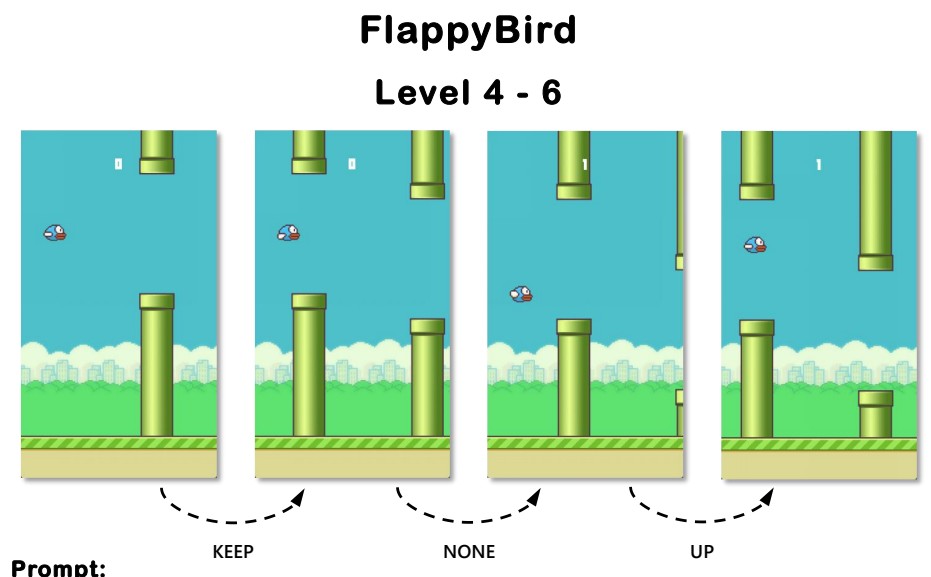

Figure 19: **FlappyBird Level 4-6: Level Design and Prompt Overview.** The images showcase the scene from Level 4, illustrating the level design and corresponding prompt. Levels are differentiated by the pipe gap width and the bird's forward speed. Elements in the same level will randomly change their initial positions while maintaining consistent relative difficulty.

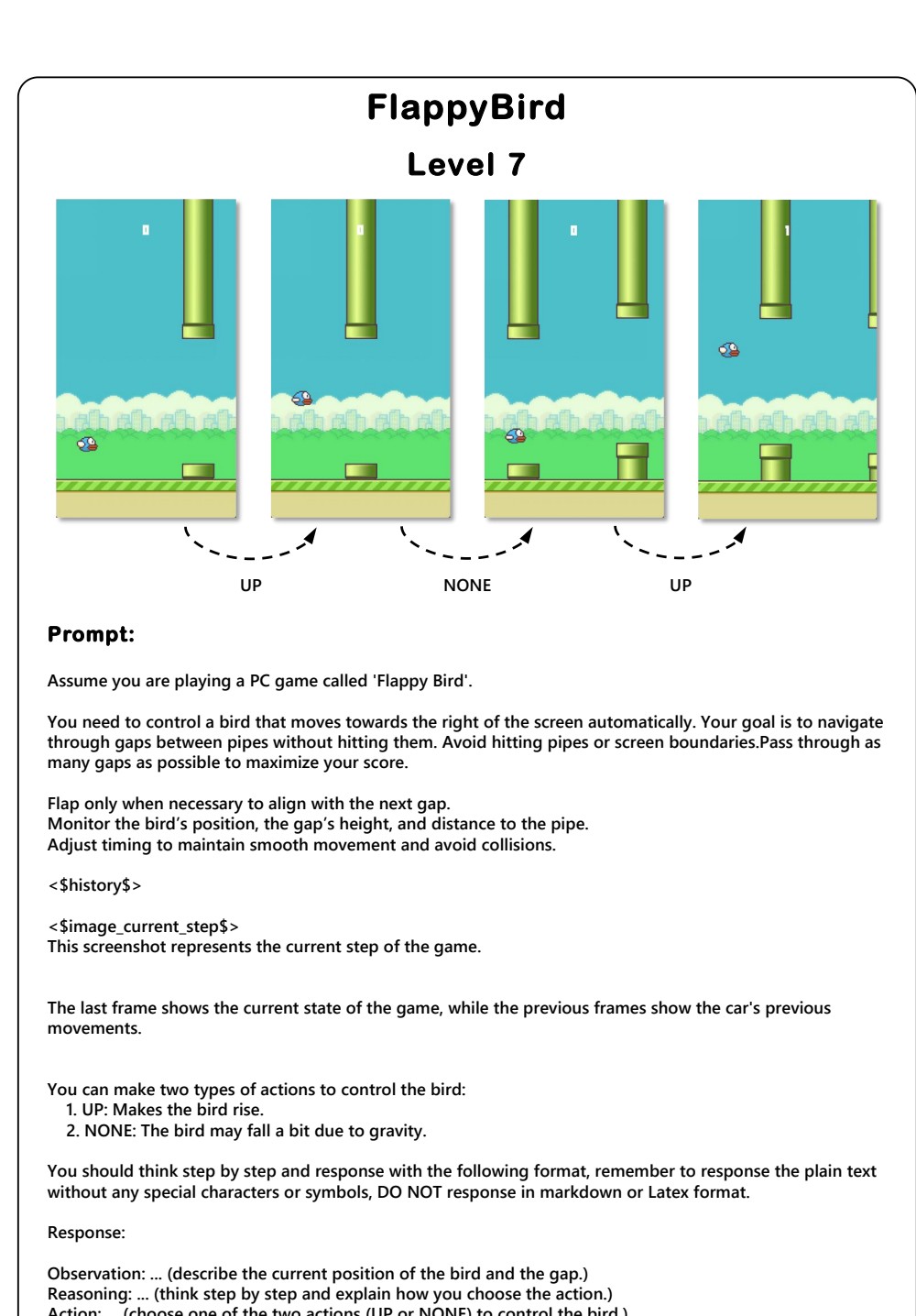

Figure 20: **FlappyBird Level 7 (Standard Level): Level Design and Prompt Overview.** The images showcase the scene from Level 7, illustrating the level design and corresponding prompt. Elements in the same level will randomly change their initial positions while maintaining consistent relative difficulty. This is The standard level that matches the difficulty of the human game.

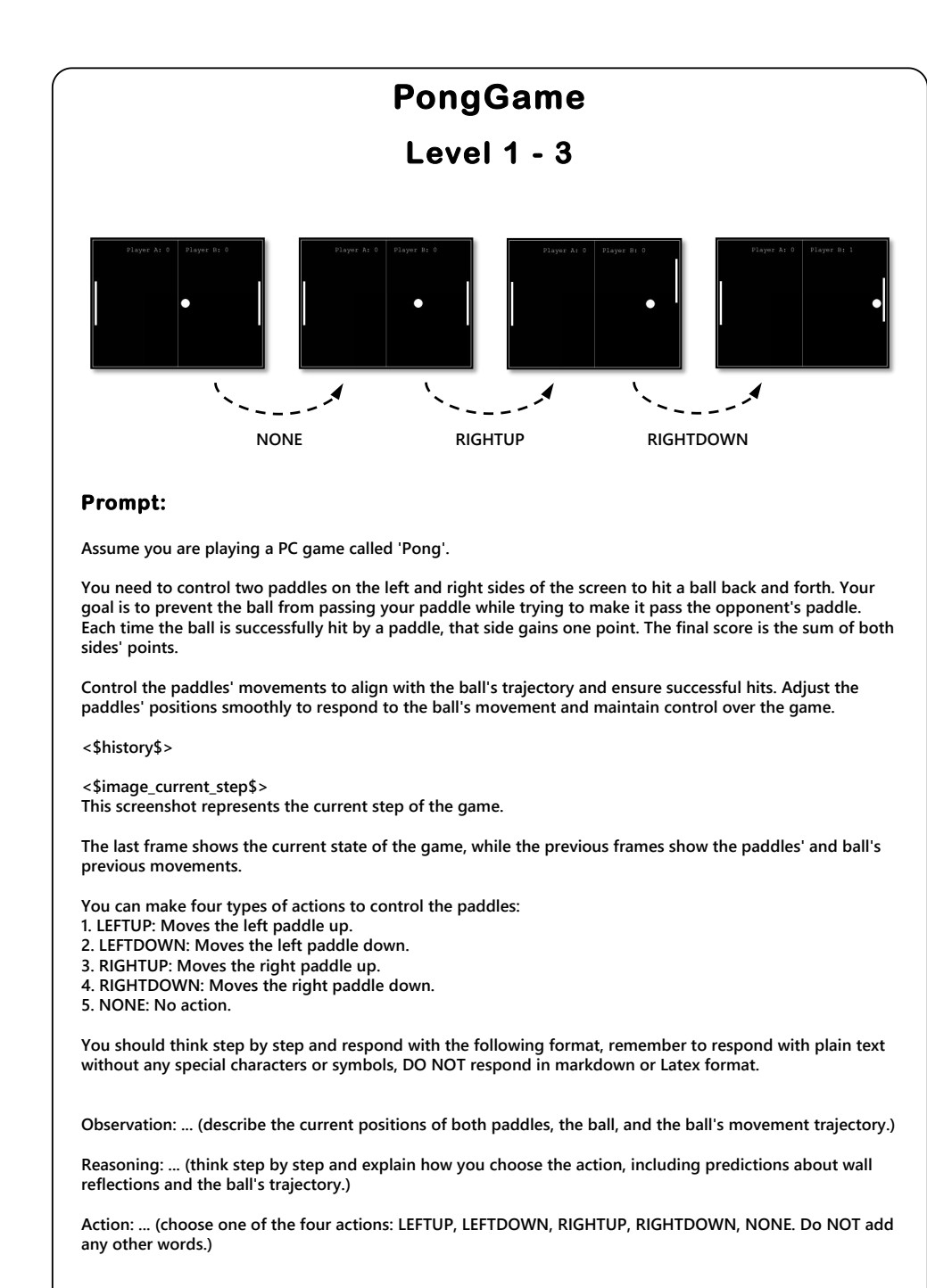

Figure 21: **PongGame Level 1-3: Level Design and Prompt Overview.** The images showcase the scene from Level 1, illustrating the level design and corresponding prompt. Levels are differentiated by the paddle width and the ping pong ball's speed. The ping pong ball in the same level will randomly change its initial position while maintaining consistent relative difficulty.

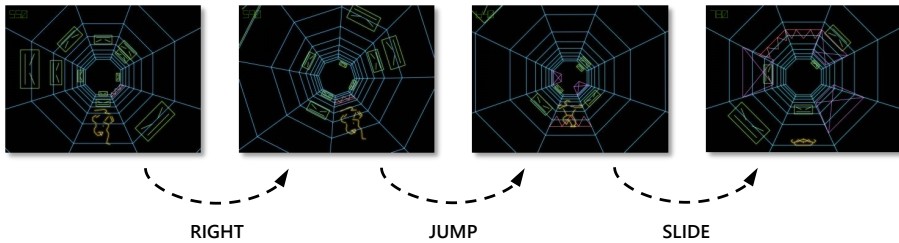

# Tempest Run

## Level 1 - 4

RIGHT          JUMP          SLIDE

**Prompt:**

Assume you are playing a PC game called 'Tempest Run'.

You need to control a character who moves through a three-dimensional space inside a futuristic tunnel filled with various obstacles and enemies. Your goal is to navigate through the tunnel, avoid or overcome obstacles, and run as far as possible. Avoid colliding with red spikes, purple walls, or failing to deal with green enemies.

Use the optimal combination of movements to progress through the tunnel smoothly and efficiently. Monitor the character's position relative to obstacles and react appropriately to avoid losing progress.

<$history$>

<$image_current_step$>
This screenshot represents the current step of the game.

The last frame shows the current state of the game, while the previous frames show the character's previous movements.

Important notes:
1. Use JUMP to jump over red spikes on the ground.
2. Use SLIDE to duck and kick green enemies to eliminate them.
3. Use LEFT or RIGHT to move around obstacles, such as purple walls or spikes.
4. Use RISE to return to a normal running position after a SLIDE.
5. NONE is a valid action to maintain the current state if no immediate action is needed.

You can make six types of actions to control the character:
1. JUMP: Makes the character jump upward, useful for avoiding ground obstacles like red spikes.
2. LEFT: Moves the character to the left.
3. RIGHT: Moves the character to the right.
4. SLIDE: Makes the character duck and slide forward, useful for dealing with green enemies or passing under certain obstacles.
5. RISE: Returns the character to a normal running position after sliding.
6. NONE: No new action is performed; the character maintains their current trajectory.

You should think step by step and respond with the following format, remember to respond with plain text without any special characters or symbols, DO NOT respond in markdown or Latex or any other format.

Response:

Observation: ... (Describe the character's current position and nearby obstacles or enemies.)
Reasoning: ... (Think step by step and explain how you choose the action.)
Action: ... (Choose ONE of the six actions to control the character. Do NOT add any other words.)

Figure 22: **Tempest Run Level 1-4: Level Design and Prompt Overview.** The images showcase the scene from Level 1, illustrating the level design and corresponding prompt. Levels are differentiated by the speed of barrier generation and the amount of visual information available. Elements in the same level will randomly change their initial positions while maintaining consistent relative difficulty.

## D  ELO PERFORMANCE COMPARISON PROTOCOL DETAILS

The core methodology for metrics evaluation in V-MAGE comprises two interconnected components: *performance comparison* and *statistical stabilization*.

**Performance Comparison Protocol.**

Each model begins with an initial Elo rating $R_m^{(0)} = 1500$, where $m \in \mathcal{M}$ represents the set of candidate models. We conducted 100 evaluation rounds for each game level $\ell$ where models were randomly paired in each round through a shuffle operation:

$$\mathcal{P}_t = \{(A_t, B_t) \mid A_t, B_t \overset{\text{rand}}{\sim} \mathcal{M}^\ell, A_t \neq B_t\} \tag{1}$$

where $A_t$ and $B_t$ denote the paired models in round $t$.

For paired models $(A, B)$, their game scores $\text{score}_A$ and $\text{score}_B$ are compared next. We first construct

$$f(m) = (\text{score}_m, \text{valid\_rate}_m) \tag{2}$$

where $\text{valid\_rate}_m$ represents the proportion of actions output by model $m$ in this game round that conform to the formatting requirements. The outcome $(S_A, S_B) \in \{(1, 0), (0, 1), (0.5, 0.5)\}$ is determined by:

$$(S_A, S_B) = \begin{cases} (1, 0) & \text{if } f(A) \succ f(B) \\ (0, 1) & \text{if } f(A) \prec f(B) \\ (0.5, 0.5) & \text{otherwise} \end{cases} \tag{3}$$

The rating update mechanism follows the classical Elo formulation with enhanced stability controls. For models $A$ and $B$ with pre-match ratings $R_A$ and $R_B$, their expected performance is calculated as:

$$E_A = \frac{1}{1 + 10^{(R_B - R_A)/400}}, \quad E_B = \frac{1}{1 + 10^{(R_A - R_B)/400}} \tag{4}$$

where the denominator base 10 and scaling factor 400 establish a logarithmic relationship between rating differences and expected outcomes. The post-match ratings become:

$$\begin{aligned} R'_A &= R_A + K(S_A - E_A) \\ R'_B &= R_B + K(S_B - E_B) \end{aligned} \tag{5}$$

where $K$ is a constant determining the sensitivity of the rating system (typically set to 32),

**Stabilization through Randomized Iteration.**

To ensure the robustness of rating updates, V-MAGE implements multi-pass stabilization protocol. All historical comparisons are aggregated into a win/loss pool:

$$\mathcal{W} = \bigcup_{g \in \mathcal{G}} \bigcup_{\ell \in \mathcal{L}_g} \bigcup_{t=1}^{N} (A_{g\ell t}, B_{g\ell t}, S_A^{g\ell t}, S_B^{g\ell t}) \tag{6}$$

which undergoes $T = 10^4$ independent shuffles. For each permutation $\pi_i(\mathcal{W})$, complete rating recalculation yields $R_m^{(i)}$. The final stabilized rating combines these trials:

$$\bar{R}_m = \frac{1}{T} \sum_{i=1}^{T} R_m^{(i)} \tag{7}$$

# E ABLATION STUDY ON PIPELINE SETTINGS

## E.1 IMPACT OF HISTORY SAMPLING CONFIGURATION

we conducted supplementary experiments on the **Qwen2.5-VL-7B** and **Qwen2.5-VL-72B** models using various history strategies (including increasing the number of history steps and altering sampling methods). The results are presented in the tables below. These scores were calculated as a percentage of model scores versus human performance in a manner similar to Figure 4.

In the default setting of our main experiments, the history sampling configuration is one where decisions are made using information from the most recent **3steps**, and the game screen is sampled every 3frames.

Table 29: Performance comparison of different history strategies for Qwen2.5-VL 7B and 72B models.

| Game | Qwen2.5-VL-7B | | | | | Qwen2.5-VL-72B | | | | |
|---|---|---|---|---|---|---|---|---|---|---|
| | 3steps_2sample | 3steps_5sample | 3steps | 5steps | 8steps | 3steps_2sample | 3steps_5sample | 3steps | 5steps | 8steps |
| **race** | 11.20 | 11.20 | 12.60 | 11.20 | 12.40 | 30.00 | 29.00 | 29.60 | 32.60 | 33.60 |
| **supermario** | 20.10 | 22.10 | 22.60 | 22.80 | 21.20 | 34.50 | 33.90 | 42.10 | 36.40 | 39.80 |
| **pong** | 3.30 | 4.30 | 3.70 | 4.00 | 4.30 | 4.50 | 3.90 | 4.10 | 4.80 | 5.00 |
| **flappybird** | 6.70 | 11.20 | 3.40 | 2.10 | 5.00 | 17.70 | 13.10 | 8.10 | 13.30 | 13.60 |
| **tempestrun** | 18.80 | 17.80 | 21.10 | 18.80 | 17.80 | 22.00 | 21.10 | 24.80 | 22.70 | 23.70 |
| **average** | **12.02** | **13.32** | **12.68** | **11.78** | **12.14** | **21.74** | **20.20** | **21.74** | **21.96** | **23.14** |

Table 30: Performance comparison of different frame sampling strategies for Qwen2.5-VL 7B and 72B models.

| Game | Qwen2.5-VL-7B | | | | Qwen2.5-VL-72B | | | |
|---|---|---|---|---|---|---|---|---|
| | 8frames | 5frames | 3frames | 1frames | 8frames | 5frames | 3frames | 1frames |
| **race** | 13.80 | 11.60 | 12.60 | 10.40 | 19.60 | 27.20 | 29.60 | 26.00 |
| **pong** | 4.00 | 3.90 | 3.70 | 4.50 | 4.80 | 5.90 | 4.10 | 7.60 |
| **flappybird** | 3.40 | 5.70 | 3.40 | 7.60 | 10.80 | 14.00 | 8.10 | 14.10 |
| **tempestrun** | 19.00 | 19.30 | 21.10 | 16.50 | 23.40 | 18.50 | 24.80 | 26.60 |
| **average** | **10.05** | **10.13** | **10.20** | **9.75** | **14.65** | **16.40** | **16.65** | **18.58** |

The experimental results show that simply increasing the length of the history window (e.g., from 3 to 8 steps) does not yield significant performance gains. This finding supports our core argument: the bottleneck for current MLLMs lies **not in the quantity** of historical information they receive, but in their ability to **understand and utilize** this dynamic visual information.

Therefore, we chose a 3-frame history as our baseline configuration. This provides the necessary temporal context while establishing a fair, simple, and effective standard for exposing the models' core deficiencies, without confounding the evaluation with complex agent strategies.

As mentioned in the main text, to investigate the impact of settings within the sampling strategies on anchoring bias, we also conducted relevant experiments, with the results presented in Appendix F.3.2.

## E.2 IMPACT OF INPUT RESOLUTION

To systematically investigate the impact of input resolution on model performance, we conducted a new set of experiments, testing the **Qwen2.5-VL 7B** and **72B** models on four different resolutions.

The resolutions from 120 to 480 refer to images with heights of 120 to 480 pixels, respectively, with the width scaled according to the original aspect ratio. We selected these four resolutions to cover different levels of visual detail, from low to high. In the default setting of our main experiments, the model's input resolution was 360 pixels height. The scores were calculated as a percentage of model scores versus human performance in a manner similar to Figure 4. The results are presented in Table 31.

Table 31: Performance comparison of Qwen2.5-VL 7B and 72B models across different input resolutions (height in pixels). Scores are percentages relative to human performance.

| Game | Qwen2.5-VL 7B | | | | Qwen2.5-VL 72B | | | |
|---|---|---|---|---|---|---|---|---|
| | 120 (7B) | 240 (7B) | 360 (7B) | 480 (7B) | 120 (72B) | 240 (72B) | 360 (72B) | 480 (72B) |
| race | 9.80 | 10.60 | 12.60 | 11.00 | 15.80 | 23.60 | 29.60 | 28.20 |
| supermario | 17.50 | 21.70 | 22.60 | 17.20 | 38.90 | 44.90 | 42.10 | 47.90 |
| pong | 4.00 | 3.90 | 3.70 | 3.60 | 3.50 | 3.50 | 4.10 | 3.60 |
| flappybird | 4.40 | 5.40 | 3.40 | 9.80 | 7.70 | 12.70 | 8.10 | 12.00 |
| tempestrun | 19.60 | 19.10 | 21.10 | 18.50 | 19.60 | 24.30 | 24.80 | 22.50 |
| average | *11.06* | *12.14* | **12.68** | *12.02* | *17.10* | *21.80* | *21.74* | **22.84** |

This data reveals a nuanced relationship: for the more capable 72B model, the overall performance trend improves with higher resolution, peaking at 480px. This suggests it can benefit from the finer details in higher-resolution images. However, for the smaller 7B model, performance peaks at our default setting of 360px and declines at the higher 480px resolution.

This indicates that the relationship between model performance and input resolution is **not simply linear**. For less capable models, excessive resolution might introduce 'noise' that they struggle to filter effectively, thereby interfering with their decision-making process.

# F  ADDITIONAL EXPERIMENTAL DETAILS

## F.1  UNIT TESTS FOR CORE VISUAL ABILITIES EXPERIMENT

Table 32: Basic visual capabilities and their corresponding simple game levels.

| Visual Abilities | Game | Levels |
|---|---|---|
| Tracking | Pong | 1, 2, 3 |
| Positioning | Race | 1, 1_no_history |
| Visual Grounding | TempestRun | 1 |
| Timing | FlappyBird | 1, 2, 3 |

The unit testing framework conducts a systematic assessment of fundamental visual capabilities by drawing from the comprehensive V-MAGE benchmark. In each carefully designed level of a game, a random baseline score is first determined by averaging scores from random actions. Following this, the performance of each evaluated model on the said level is quantified by calculating the percentage of rounds where the model's score outperforms this established random baseline. The specific game levels used for assessing each ability are listed in Table 32.

As illustrated in Figure 5, model performances across representative levels for four fundamental visual competencies reveal critical insights: In tracking tasks requiring cross-frame analysis, nearly all models underperform random baselines. This indicates that while current models achieve reasoning through caption-based approaches in single-frame tasks, they struggle to extract discriminative features in multi-frame scenarios requiring fine-grained spatiotemporal comparisons. The quantitative results for each model across the four core visual abilities are presented in Table 33.

Table 33: Performance of MLLMs on Core Visual Ability Unit Tests (% Exceeding Random Baseline)

| Model | Positioning | Tracking | Visual Grounding | Timing |
|---|---|---|---|---|
| Qwen2VL 7B | 0.50 | 0.27 | 0.56 | 0.36 |
| Qwen2VL 72B | 0.76 | 0.26 | 0.70 | 0.43 |
| Qwen2.5VL 72B | 0.88 | 0.25 | 0.68 | **0.51** |
| InternVL2.5 78B | 0.82 | **0.33** | 0.66 | 0.49 |
| InternVL2.5 8B | 0.60 | 0.28 | 0.55 | 0.39 |
| Gemini-2.0-Flash | 0.68 | 0.32 | **0.70** | **0.51** |
| GPT4o | **0.98** | 0.29 | 0.66 | 0.58 |

It is important to interpret the results of these unit tests within their intended scope. Designed to assess fundamental visual competencies, these tests utilize a random baseline score as the primary reference point. While a model significantly outperforming this random baseline indicates a degree of relevant reasoning ability in that specific task dimension, it does not necessarily imply a high level of overall competence. The random baseline represents minimal performance, and even achieving scores far exceeding it on these foundational tests serves primarily to diagnose basic capabilities rather than validate advanced mastery required for complex gameplay.

## F.2  PERCEPTUAL SKIPPING EXPERIMENT

To further investigate the interplay between visual perception and reasoning, we conducted supplementary experiments where textual descriptions of the game state were provided, effectively bypassing the visual perception module (see Table 34 for detailed results on Flappy Bird Level 3 and Pong Level 2).

Table 34: Model performance on simple levels with and without textual state information.

| Model | Flappy Bird | | Pong | |
|---|---|---|---|---|
| | w/o Text | w/ Text | w/o Text | w/ Text |
| Qwen2.5VL 7B | 0.8 | 0.35 | 0.19 | 0.25 |
| InternVL2.5 8B | 0.31 | 0.76 | 0.19 | 0.31 |
| Qwen2.5VL 72B | 0.35 | 2.17 | 0.21 | 1.19 |
| InternVL2.5 78B | 0.59 | 2.39 | 0.16 | 0.52 |
| GPT4o | 0.57 | 4.55 | 0.20 | 3.89 |
| Gemini-2.0-Flash | 0.42 | 4.89 | 0.32 | ¿10 |
| random | 0.52 | | 0.18 | |
| human | > 10 | | > 10 | |

The results indicate that alleviating the perceptual challenge generally improves performance, particularly for larger models like GPT-4o and the 72B/78B parameter models, supporting the hypothesis that visual perception is a significant bottleneck. However, even with this intervention, model scores remained substantially lower than the human baseline (¿10), underscoring the presence of critical reasoning and planning deficiencies beyond visual perception, as discussed earlier.

Notably, the performance gains from text input were more pronounced for larger models, suggesting their enhanced capacity to leverage structured textual information for reasoning, whereas smaller models exhibited less consistent benefits or even performance degradation in some cases. This finding further highlights that while perception is a challenge, fundamental reasoning limitations persist across models and are not fully overcome even when provided with simplified, textual state representations.

## F.3 ANCHORING BIAS EXPERIMENTS

### F.3.1 DETAILS AND EXAMPLES

Table 35: Average number of rounds for each model to generate different responses.

| Model | Race | FlappyBird | Pong | TempestRun | Avg. |
|---|---|---|---|---|---|
| Qwen2VL 7B | 4.3 | 25.9 | 13.7 | 7.3 | 12.8 |
| Qwen2.5VL 72B | 2.3 | 19.3 | 2.6 | 5.3 | 7.4 |
| InternVL2.5 8B | 2.0 | 6.9 | 6.7 | 8.0 | 5.9 |
| InternVL2.5 78B | 6.8 | 16.0 | 2.0 | 3.0 | 7.0 |
| GPT4o | **1.0** | **1.6** | **1.0** | **1.0** | **1.1** |
| **PCC** $r$ (Avg. Rounds vs. ELO) | -0.63 | -0.86 | -0.88 | -0.64 | -0.75 |

Due to the possibility of models receiving identical visual inputs over multiple rounds in Super Mario (e.g., being stuck in a corner), we conducted a statistical analysis using lots of rounds of responses from each model across the other four games. This was done by iterating through the recorded interactions for each level and measuring the number of sequential rounds where the model's output response remained unchanged. The average of these durations across interactions within a game provides the metric presented in Table 35.

The results indicate that GPT-4o updates its responses more actively and frequently when the visuals change, while other models do so less frequently. This may suggest that GPT-4o is more sensitive to subtle visual updates, enabling it to make timely inferences and more accurately track game progress.

In the example of Figure 23, GPT-4o's response during the final interaction is as follows:

Figure 23: In the prior reasoning processes, GPT-4o accurately identified the car's orientation relative to the trophy's position. However, in the final reasoning instance, due to anchoring bias, the model misinterpreted previous historical information and incorrectly assumed that the car's orientation was already directed toward the trophy. Consequently, despite correctly determining the direction, the model executed an erroneous action.

---

Description: The car is currently facing diagonally towards the **bottom-right** of the screen. The trophy is positioned near the car, slightly to the **right**. There are no visible obstacles between the car and the trophy, indicating a clear path. The car has been moving consistently forward in the previous steps, maintaining its alignment towards the trophy.

Reasoning: Given the car's orientation towards the trophy and the absence of obstacles, the car's current position and direction remain favorable for reaching the trophy. Continuing to move forward is the most efficient strategy to maintain the direct path to the trophy. The car's current trajectory should allow it to reach the trophy without any adjustments.

Action: FORWARD

---

### F.3.2 IMPACT OF FRAME SAMPLING AND DECISION FREQUENCY

To further quantify the relationship between perceptual sensitivity and anchoring bias, we have conducted a deeper quantitative exploration of the relationship between anchoring bias and model performance. To more objectively measure a model's reaction to dynamic changes in the game world, we introduced a new metric: **Average Response Game Frames** (abbreviated as **'avg frames'** in the results). This metric is calculated by:

$$\text{(sampling interval)} \times \text{(average rounds to generate different responses)}$$

and represents **how many game frames, on average, have elapsed before a model makes a substantive change in its reasoning**.

We performed a series of experiments with different sampling strategies, first testing the Qwen2.5VL-72B model. The results are as follows:

Table 36: Performance of Qwen2.5VL-72B under different frame sampling strategies. The top section shows response frequency metrics, while the bottom shows game scores.

| Game | 8frames | 5frames | 3frames(default) | 1frames |
|---|---|---|---|---|
| *Average Response Game Frames* | | | | |
| **race** | 5 | 1.8 | 2.3 | 10.8 |
| **pong** | 1.7 | 1.7 | 2.6 | 19.7 |
| **flappybird** | 1.5 | 2.3 | 19.3 | 64.2 |
| **tempestrun** | 1.8 | 14.2 | 5.3 | 23.2 |
| **avg request** | 2.5 | 5 | 7.4 | 29.5 |
| **avg frames** | **20** | **25** | **22.2** | **29.5** |
| *Game Score* | | | | |
| **race** | 19.60 | 27.20 | 29.60 | 26.00 |
| **pong** | 4.80 | 5.90 | 4.10 | 7.60 |
| **flappybird** | 10.80 | 14.00 | 8.10 | 11.90 |
| **tempestrun** | 23.40 | 18.50 | 24.80 | 26.60 |
| **avg score** | **14.65 ↓2.00** | **16.40 ↓0.25** | **16.65 ↑0.00** | **18.03 ↑1.38** |

**Stable 'visual reaction threshold' in strong models:** From the avg frames metric, the Qwen2.5VL-72B model demonstrates remarkable consistency across different sampling strategies, with its av-

erage response time stabilizing within a narrow range of **20-30** game frames. This suggests that the model possesses a relatively constant intrinsic reaction threshold, where a certain amount of accumulated visual change triggers a shift in its reasoning.

**Regarding task score (avg score):** The 72B model's performance clearly improves as the sampling interval decreases, with the highest score achieved at the highest decision frequency (1-frame interval). Under such high-frequency decision-making, the model can capture crucial task timings with the highest precision. As the decision frequency decreases, the opportunities for the model to take appropriate action at the right moment are reduced, thus may leading to a drop in performance.

Next, we compared the **Qwen2.5VL-7B** and **72B** models under the same sampling strategies:

Table 37: Comparison of Qwen2.5VL-7B and 72B models across sampling strategies. The 72B model shows a consistently lower reaction threshold (avg frames) and higher scores.

| Game | 8frames | 5frames | 3frames | 1frames |
|---|---|---|---|---|
| **Qwen2.5VL-7B** | | | | |
| avg request | 13.5 | 12.5 | 34.7 | 97.8 |
| avg frames | **108.0** | **62.5** | **104.1** | **97.8** |
| avg score | **10.1** | **10.1** | **10.2** | **9.8** |
| **Qwen2.5VL-72B** | | | | |
| avg request | 2.5 | 5 | 7.4 | 29.5 |
| avg frames | **20.0 ↓88.0** | **25.0 ↓37.5** | **22.2 ↓81.9** | **29.5 ↓68.3** |
| avg score | **14.7 ↑4.6** | **16.4 ↑6.3** | **16.7 ↑6.5** | **18.6 ↑8.8** |

The correlation between the "visual reaction threshold" and "task score" remains clear when comparing across models. The 72B model exhibits a **lower** Average Response Game Frames (indicating higher perceptual sensitivity) and a **higher** task score, while the 7B model shows the opposite. This is consistent with the conclusions about anchoring bias: a more powerful model possesses greater sensitivity to dynamic visual perception, which forms the basis for more accurate decision-making in interactive tasks.

### F.4    ANALYSIS OF GPT4O ERRORS IN V-MAGE

We have collected **2,351** prompt-response pairs generated by GPT-4o while completing all levels for 1 to 5 rounds. From these, **494** examples were randomly and uniformly sampled for manual error annotation. The frequency of occurrence for various error types is presented in Table 38.

Table 38: Error count by error type and game environment

| Error Type | FlappyBird | Pong | Race | SuperMario | TempestRun |
|---|---|---|---|---|---|
| no error | 30 | 18 | 54 | 88 | 21 |
| perception error | 80 | 26 | 26 | 47 | 42 |
| direction error | 2 | 19 | 13 | 16 | 8 |
| recognition error | 1 | 0 | 0 | 0 | 5 |
| perception incomplete | 3 | 0 | 8 | 10 | 10 |
| reasoning error | 24 | 4 | 9 | 10 | 6 |
| history misinterpretation | 21 | 0 | 1 | 6 | 2 |
| action inappropriate | 0 | 0 | 5 | 0 | 0 |
| inconsistency | 0 | 0 | 14 | 1 | 0 |
| instruction following | 0 | 0 | 2 | 0 | 0 |

The definitions of each error type are presented as follows:

- **no error**: There is no error in the response.

- **perception error**: Description misinterpreted elements.

- **direction error**: A type of perception error. Confused directions (e.g., LEFT/RIGHT)
- **recognition error**: A type of perception error. Failed to identify key objects/elements.
- **perception incomplete**: Description missed important elements in the scene.
- **reasoning error**: Flawed logic in the reasoning section for the chosen action.
- **history misinterpretation**: A type of reasoning error. Misunderstood the game history.
- **action inappropriate**: A type of reasoning error. The chosen Action is clearly wrong given the Observation/Reasoning.
- **inconsistency**: Inconsistent action plans in multiple response processes.
- **instruction following**: Failed to follow instructions in the prompt.

# G   CASE STUDY

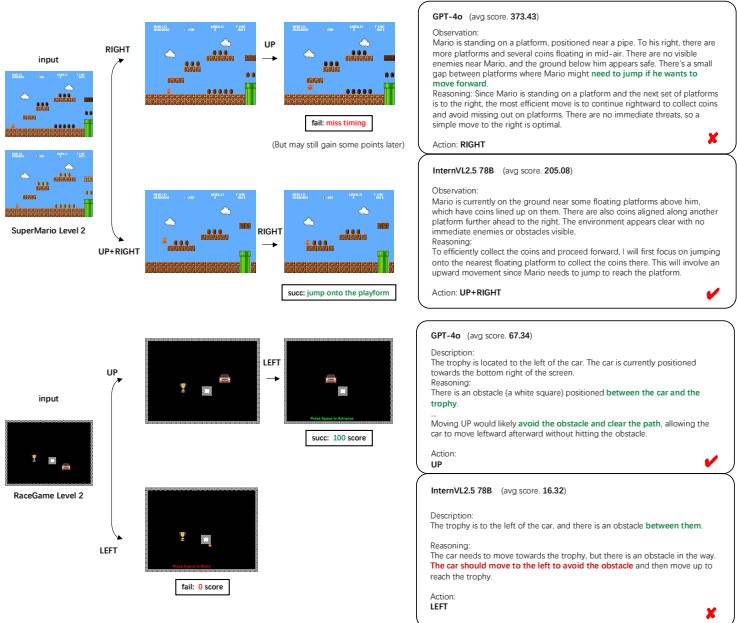

Figure 24: case studies group 1.

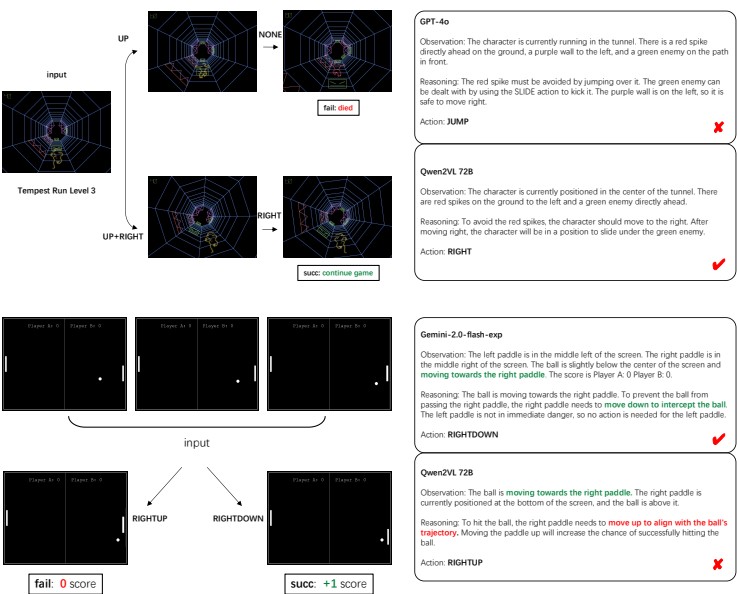

Figure 25: case studies group 2.

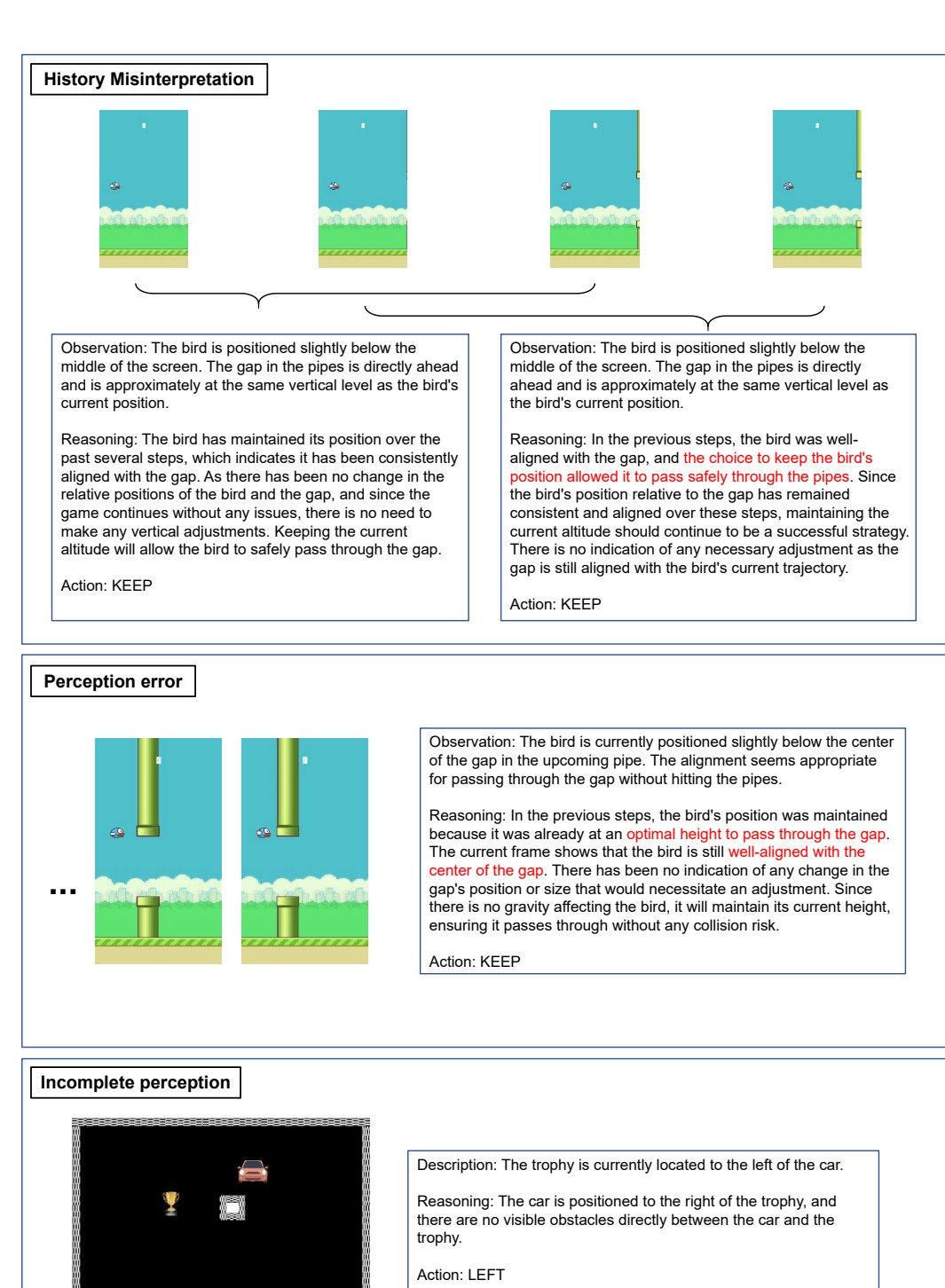

Figure 26: Error analysis in GPT4o cases.

**Inconsistency**

Description: The trophy is currently positioned to the right of the car.

Reasoning: There is an obstacle, which appears to be a wall, located between the car and the trophy. The obstacle is directly in the path if the car attempts to move RIGHT towards the trophy. Therefore, the car should avoid moving directly towards the obstacle and instead find an alternative path.

Action: UP

Description: The trophy is located to the right and slightly below the car.

Reasoning: There is an obstacle (a square block) directly between the car and the trophy. Moving directly to the right would result in a collision with the obstacle. Therefore, the car must first navigate around the obstacle by moving downward, then move to the right to approach the trophy safely.

Action: DOWN

**Action Inappropriate**

Description: The trophy is to the right of the car.

Reasoning: The trophy is directly to the right of the car. There is a vertical obstacle (a wall) between the car and the trophy. The car needs to **avoid colliding with the obstacle** to reach the trophy.

Action: **RIGHT**

**Perception error**

Description: The car is currently facing diagonally towards the bottom-right of the screen. The trophy is positioned in front of the car, slightly to the right. There are no visible obstacles between the car and the trophy, indicating a clear path. The car has been moving consistently forward in the previous steps, maintaining its alignment towards the trophy.

Reasoning: Given the car's orientation towards the trophy and the absence of obstacles, the car's current position and direction remain favorable for reaching the trophy. Continuing to move forward is the most efficient strategy to maintain the direct path to the trophy. The car's current trajectory should allow it to reach the trophy without any adjustments.

Action: FORWARD

Figure 27: Error analysis in GPT4o cases.

**Direction Error**

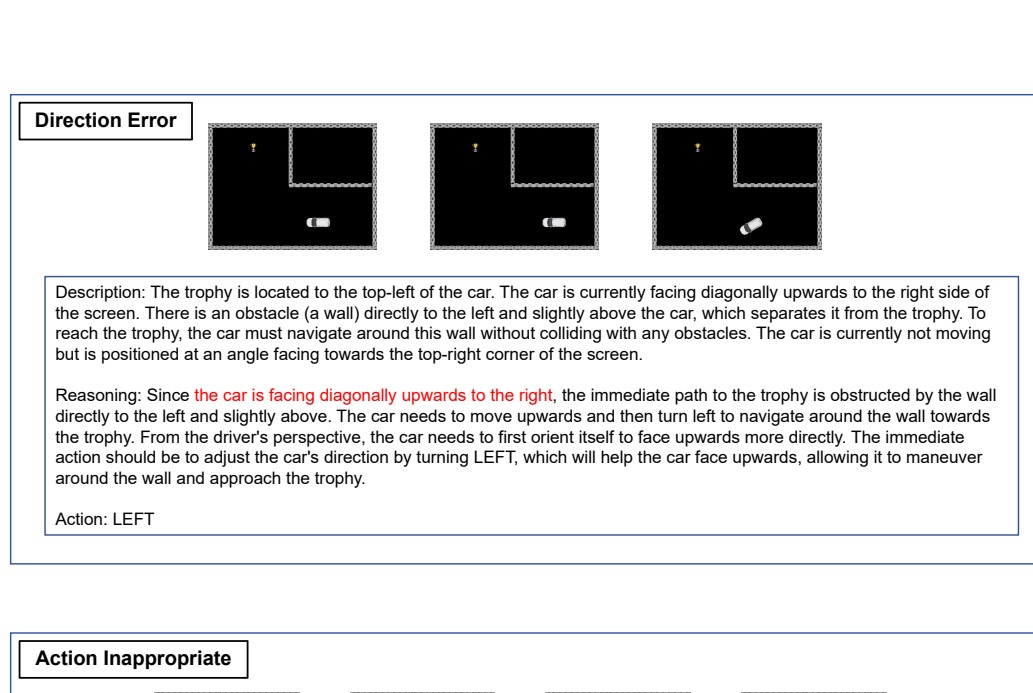

Description: The trophy is located to the top-left of the car. The car is currently facing diagonally upwards to the right side of the screen. There is an obstacle (a wall) directly to the left and slightly above the car, which separates it from the trophy. To reach the trophy, the car must navigate around this wall without colliding with any obstacles. The car is currently not moving but is positioned at an angle facing towards the top-right corner of the screen.

Reasoning: Since the car is facing diagonally upwards to the right, the immediate path to the trophy is obstructed by the wall directly to the left and slightly above. The car needs to move upwards and then turn left to navigate around the wall towards the trophy. From the driver's perspective, the car needs to first orient itself to face upwards more directly. The immediate action should be to adjust the car's direction by turning LEFT, which will help the car face upwards, allowing it to maneuver around the wall and approach the trophy.

Action: LEFT

**Action Inappropriate**

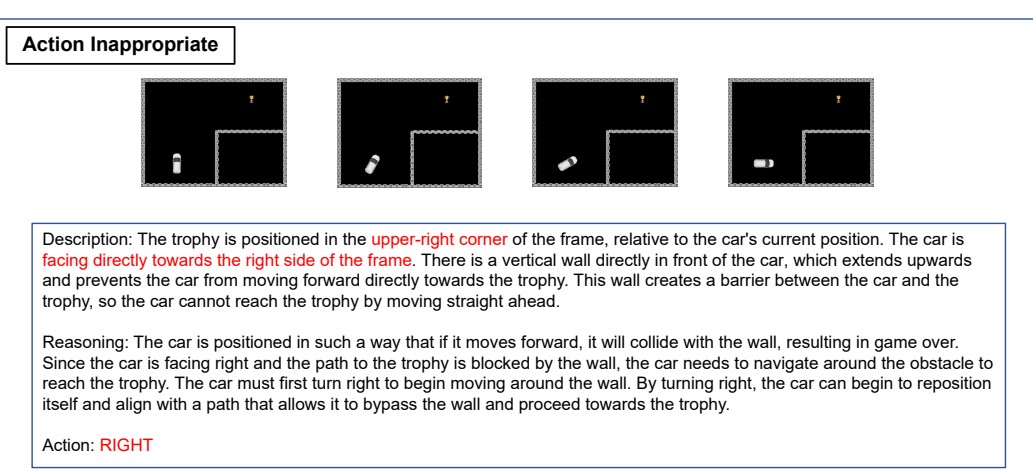

Description: The trophy is positioned in the upper-right corner of the frame, relative to the car's current position. The car is facing directly towards the right side of the frame. There is a vertical wall directly in front of the car, which extends upwards and prevents the car from moving forward directly towards the trophy. This wall creates a barrier between the car and the trophy, so the car cannot reach the trophy by moving straight ahead.

Reasoning: The car is positioned in such a way that if it moves forward, it will collide with the wall, resulting in game over. Since the car is facing right and the path to the trophy is blocked by the wall, the car needs to navigate around the obstacle to reach the trophy. The car must first turn right to begin moving around the wall. By turning right, the car can begin to reposition itself and align with a path that allows it to bypass the wall and proceed towards the trophy.

Action: RIGHT

**Instruction Following**

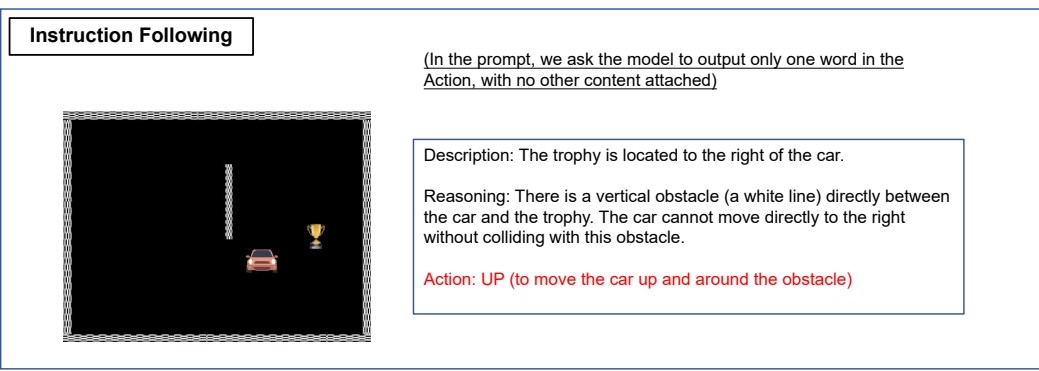

(In the prompt, we ask the model to output only one word in the Action, with no other content attached)

Description: The trophy is located to the right of the car.

Reasoning: There is a vertical obstacle (a white line) directly between the car and the trophy. The car cannot move directly to the right without colliding with this obstacle.

Action: UP (to move the car up and around the obstacle)

Figure 28: Error analysis in GPT4o cases.

**Direction Error**

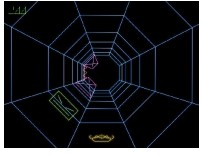 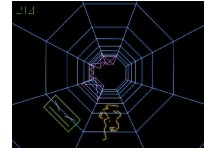 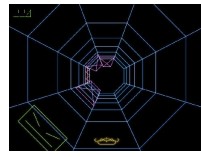 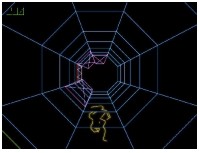

Observation: The left paddle is positioned in the middle of the left side. The right paddle is slightly above the center on the right side. The ball is moving towards the right side of the screen and is slightly above the center line, continuing its trajectory towards the right paddle.

Reasoning: The ball is heading towards the right paddle and is slightly above the center line. Given the ball's trajectory, it appears to be on a direct path towards the upper part of the right paddle. The right paddle is already positioned slightly above the center, which aligns well with the ball's current path. To ensure the ball is intercepted effectively, the right paddle should be moved up slightly to account for any upward drift in the ball's trajectory. The left paddle does not need any adjustment since the ball is moving away from it.

Action: RIGHTUP

**Direction Error**

Observation: The character is currently in a normal running position at the bottom center of the tunnel. There is a purple wall approaching in the center of the tunnel.

Reasoning: The character is in a normal running position, and there is a purple wall directly ahead. To avoid the purple wall, the character should move around it by using the LEFT action to shift to the left. This will help the character to safely navigate through the tunnel without colliding with the wall.

Action: LEFT

**Action Inappropriate**

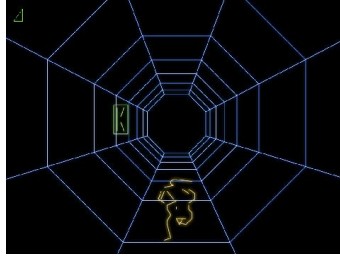

Observation: The character is currently running in the center of the tunnel, and there is a green enemy approaching in the left section of the tunnel ahead.

Reasoning: The green enemy is approaching on the left side, and to eliminate it, the character needs to perform a slide. Sliding will allow the character to duck and kick the green enemy. Since the character is currently running in the center, sliding will position them correctly to deal with the green enemy.

Action: SLIDE

Figure 29: Error analysis in GPT4o cases.

# H   MISCELLANEOUS MATERIAL

## H.1   LLM USAGE STATEMENT

Our research methodology centered on the evaluation of various MLLMs. Models such as GPT-4o served as the subjects within our V-MAGE pipeline, generating the outputs that form the basis of our analysis and conclusions on MLLM performance. The role of these MLLMs was strictly limited to this evaluation phase. The conceptualization and implementation of the V-MAGE framework and its software were carried out entirely by the authors.

For the manuscript preparation, we employed LLMs for the sole purpose of improving grammar and polishing the language. All scientific contributions, including the research ideas, experimental design, and results interpretation, originate exclusively from the authors.

## H.2   IMPACTS STATEMENT

This research contributes to the field of multimodal models by providing a novel and challenging benchmark for evaluating vision-centric capabilities in dynamic environments. The primary positive impact is facilitating the diagnosis of limitations in current MLLMs and guiding future research towards developing more capable, robust, and potentially safer AI systems for real-world interaction. As our work focuses on foundational evaluation in simulated environments and does not involve the deployment of high-risk models or the collection of sensitive personal data, the potential for negative societal impacts is considered minimal and indirect at this stage. We believe that developing better evaluation tools is a crucial step towards building more reliable and trustworthy AI.

