# OpenReview forum: "V-MAGE: A Game Evaluation Framework for Assessing Vision-Centric Capabilities in Multimodal Large Language Models"
_ICLR.cc/2026/Conference — ICLR 2026 Conference Withdrawn Submission_

### Official Review · Reviewer_4tME · 2025-10-21

**Soundness:** 3
**Presentation:** 3
**Contribution:** 2
**Rating:** 4
**Confidence:** 3

**Summary:**

The paper introduces V-MAGE, a game-based evaluation framework for MLLMs that targets vision-centric abilities under dynamic, interactive conditions. V-MAGE comprises five video games with 30+ levels, provides visual-only inputs (continuous state spaces, difficulty tiers), and evaluates models through a modular game–agent–model pipeline. To compare heterogeneous tasks, it adopts an adaptive ELO-based ranking for interpretable cross-game scoring. Experiments on state-of-the-art MLLMs (with human references) show that models can handle simpler scenarios but degrade markedly on complex tasks requiring perception, temporal reasoning, memory, and action orchestration.

**Strengths:**

1. The paper is well-structured, with clear motivation against text/grid baselines and a readable exposition of the ELO-based scoring. Figures and sectioning make the contributions and gaps intuitive.

2. The framework is practical: lightweight game environments, standardized interfaces, and an extensible level design lower the barrier to adoption and invite community contributions (new games, agents, or metrics) while supporting reproducibility.

3. The core idea—using controllable, visual-only video games with continuous states to evaluate vision-centric, interactive reasoning—is timely. The modular game–agent–model design plus an adaptive ELO scheme pushes beyond text/grid reductions and enables comparable, cross-level scoring.

**Weaknesses:**

1. The paper does not convincingly explain why these five games, beyond high-level descriptions. It remains unclear how each game uniquely probes a specific vision-centric or interactive capability, or whether the set is minimally sufficient/covering.

2. Several state-of-the-art closed-source models (e.g., GPT-o1/o3, Gemini 2.5, Claude family) are absent, leaving open whether the reported gaps persist under the strongest available systems. This weakens external validity and limits the benchmark’s positioning.

3. Most importantly, becuase these games are widely available online, strong performance may partly reflect prior exposure rather than genuine vision-centric capability. The paper does not present leakage audits or controls to rule out memorization effects.

**Questions:**

1. What a priori criteria led to these five games (vs. others), and how does each map to a concrete capability taxonomy (e.g., perceptual binding, spatiotemporal prediction, visual working memory, causal control)?

2. Are there counterexamples you considered but excluded—why?

3. Could you include some current top-tier closed model (e.g., GPT-o1/o3, Gemini 2.5, Claude) or a carefully controlled proxy comparison?

4. Please report performance under standardized prompting regimes: zero-shot, k-shot ICL (vary k), and CoT.

---

> ### Author Response · Authors · 2025-11-21
>
> We appreciate the reviewer’s recognition of V-MAGE’s timeliness and practical value. We address your concerns regarding game selection criteria, model coverage, and leakage/memorization below with updated experiments and clarifications.
>
> > **Q1. Principles for game selection and capability taxonomy**
>
> Our primary selection criterion is **“visually irreducible”**: we specifically excluded games where the state can be easily converted into text/grid representations via OCR or a small set of discrete symbols.
>
> **Exclusions.** We excluded grid-based games (e.g., *2048*, *Go*, chess) and high-complexity black-box games (e.g., large-scale RTS titles) to focus on reproducible, fine-grained “unit testing” of cognitive skills such as pixel-level positioning, continuous tracking, and simple physics intuition, rather than complex domain knowledge or long-horizon narratives.
>
> **Capability taxonomy.** Rather than simply scaling the number of games, we emphasized **level design within games** to probe distinct vision-centric abilities under controlled conditions. We have reorganized the design intent for each game and level as follows, and will integrate a clearer version of this table into the final manuscript:
>
> | Game | Level | Target Ability |
> | :--- | :--- | :--- |
> | **Race** | 1 - 3 | Pixel Positioning, Planning |
> | | 4 - 6 | Multi-frame Reasoning, Acceleration Understanding |
> | **FlappyBird** | 1 - 3 | Pixel Positioning, Timing |
> | | 4 - 6 | Gravity Understanding, Planning |
> | | 7 | Visual Memory |
> | **SuperMario** | 1 - 2 | Rule Understanding, Timing |
> | | 3 - 4 | Long Term Memory, Reflection |
> | | 5 - 7 | Visual Counting |
> | | 8 - 9 | Timing |
> | | 10 | All-Rounding |
> | **Pong** | 1 - 3 | Tracking, Timing, Pixel Positioning |
> | **TempestRun** | 1 - 2 | Grounding, Rule Understanding, Timing |
> | | 3 - 4 | Planning, Timing |
>
> These games collectively cover: single-frame perception, spatiotemporal tracking, physical intuition (gravity, acceleration), visual working memory, and multi-step planning. We agree there are many other possible choices, but we deliberately opted for a **minimal yet diverse set** that (1) remains lightweight and easy to reproduce, and (2) can be systematically extended by the community via new levels or games within the same framework.
>
> > **Q2. Evaluation on stronger SOTA models**
>
> To strengthen external validity, we expanded our benchmark to include currently available top-tier models, including **GPT-5**, **GPT-5.1**, **Gemini 2.5 Pro**, **Gemini 2.5 Flash**, and **Qwen3-VL-235B-A22B**. The updated ELO ratings and average performance ratios (model score / human score, in %) are:
>
> | Model                              | Avg ELO Rating | Performance Ratio (%) |
> | :--------------------------------- | :------------: | :-------------------: |
> | **GPT-5-2025-08-07**               | **1709.6**     | **43.4**              |
> | **Gemini-2.5-Pro**                 | **1604.0**     | **36.3**              |
> | **Claude-3.7-Sonnet**              | **1542.9**     | **30.8**              |
> | **Gemini-2.5-Flash**               | **1528.3**     | **23.8**              |
> | GPT-4o                             | 1526.4         | 26.6                  |
> | Gemini-2.0-Flash-Thinking          | 1515.5         | 22.6                  |
> | **Qwen3-VL-235B-A22B-Instruct**    | **1515.4**     | **24.3**              |
> | Qwen2.5-VL-72B-Instruct            | 1515.0         | 22.8                  |
> | InternVL2.5-78B                    | 1492.2         | 19.2                  |
> | **GPT-5.1-2025-11-13**             | **1486.3**     | **20.1**              |
> | Gemini-2.0-Flash                   | 1484.2         | 16.7                  |
> | Qwen2-VL-72B-Instruct              | 1473.2         | 16.5                  |
> | InternVL2.5-8B                     | 1441.2         | 12.9                  |
> | Qwen2.5-VL-7B-Instruct             | 1439.3         | 12.1                  |
> | Qwen2-VL-7B-Instruct               | 1437.5         | 11.4                  |
> | Keye-VL-8B-Preview                 | 1434.2         | 13.1                  |
> | Phi-4-multimodal-instruct          | 1429.2         | 13.7                  |
> | Random                             | 1425.6         | 11.0                  |
>
> **GPT-5 (default thinking)** sets a new record ELO of **1709.6**. Despite this progress, it still reaches only **43.4%** of the human baseline on average. The fact that even such frontier models fall well short of human-level performance confirms that the **dynamic reasoning gap** captured by V-MAGE persists under the strongest systems currently available, underscoring the benchmark’s continued relevance.

---

> ### Author Response · Authors · 2025-11-21
>
> > **Q3. Leakage audits and memorization controls**
>
> We agree that, because some of the games (e.g., Super Mario 1-1) are widely available online, it is important to rule out the possibility that strong performance is driven primarily by memorization of training data rather than genuine dynamic ability.
>
> **Dynamic initialization via seeding.**  Unlike static datasets or fixed trajectories, V-MAGE introduces controlled variability across evaluation rounds. For each episode, we generate a **seed-dependent random initialization** of the game state (e.g., different starting coordinates in *Race*, varying ball trajectories in *Pong*, and procedurally sampled obstacle layouts in *TempestRun*).
>
> Seeds are chosen as a function of the round index and level, so that: 1) the evaluation is **reproducible**, and 2) models must handle a **distribution over initial conditions**, not a single fixed trajectory.
>
> This design does not make memorization theoretically impossible, but it does render naïve “trajectory replay” strategies ineffective, as there is no single optimal path that can be applied to every run.
>
> **The "Mario 1-1" Performance:** If models were primarily relying on training-set leakage (e.g., imitating popular YouTube playthroughs), we would expect near-perfect performance on the iconic *Super Mario Bros. World 1-1* (Level 10), which is heavily represented in web-scale corpora. However, our experiments show that models perform **substantially worse** than humans on this level, and in many runs fail in basic timing and obstacle-avoidance decisions. This suggests that even on a “familiar” map, models struggle with real-time control and continuous perception, indicating that memorized trajectories alone are insufficient.
>
> **Visual Irreducibility:** As noted in our selection criteria, success in V-MAGE requires perceiving continuous, analog values (e.g., precise jump height in *FlappyBird*) that are rarely available as explicit captions in training data, and cannot be trivially reduced to short textual state descriptions. This significantly limits the usefulness of text-based memorization or knowledge recall shortcuts.
>
> Overall, while we cannot fully exclude all forms of distributional overlap, the combination of seeded variability, performance on “seen” maps like Mario 1-1, and reliance on fine-grained visual signals strongly suggests that V-MAGE scores reflect genuine dynamic visual competence rather than simple memorization.

---

> ### Author Response · Authors · 2025-11-21
>
> > **Q4. Standardized Prompting Regimes**
>
> We conducted additional ablations on prompting strategies to assess how **in-context learning** and **chain-of-thought** reasoning affect performance.
>
> ### A. Zero-shot / K-shot ICL
>
> In the ICL setting, we augment the prompt with human demonstration snippets (screenshots plus actions and brief reasoning). “$k$-shot” denotes the number of such demonstrations embedded in the prompt.
>
> | Model | 0-shot (Default) | 1-shot | 3-shot |
> | :--- | :---: | :---: | :---: |
> | **Qwen3-VL-235B-A22B** | 24.3% | 24.1% ($\\color{red}{\\downarrow 0.2\\%}$) | 22.6%  ($\\color{red}{\\downarrow 1.7\\%}$) |
> | **Qwen2.5-VL-72B** | 22.8% | 23.7% ($\\color{green}{\\uparrow 0.9\\%}$) | 21.4% ($\\color{red}{\\downarrow 1.4\\%}$) |
> | **Qwen2.5-VL-7B** | 12.1% | 13.5% ($\\color{green}{\\uparrow 1.4\\%}$) | 12.2% ($\\color{green}{\\uparrow 0.1\\%}$) |
>
> Overall, performance remains **largely stable** across different ICL settings. 1-shot examples sometimes provide a small benefit (typically < 1.5%), but moving from 1-shot to 3-shot yields no consistent improvement and can even cause a small degradation. This suggests that, in our setting, adding more static text demonstrations in the context window has **limited impact** on models’ abilities to handle real-time visual dynamics.
>
> ### B. CoT / no-CoT
>
> We also compared models with and without explicitly encouraged reasoning steps. The impact depends strongly on both **task type** and **model scale**.
>
> *(Cells show “No-CoT % $\\to$ CoT % (Change)”. Green ($\\color{green}{\\uparrow}$) = CoT helps; Red ($\\color{red}{\\downarrow}$) = CoT hurts.)*
>
> | Model                      | Overall Average                         | Race (tracking-heavy)                      | SuperMario (planning-heavy)                      |
> | :------------------------- | :--------------------------------------:| :------------------------------------------:| :-----------------------------------------------:|
> | **Qwen3-VL-235B-A22B**    | 21.6% $\\to$ 24.3% ($\\color{green}{\uparrow 2.7\\%}$) | 36.2% $\\to$ 34.9% ($\\color{red}{\\downarrow 1.3\\%}$) | 32.3% $\\to$ 44.5% ($\\color{green}{\\uparrow 12.2\\%}$) |
> | **Qwen2.5-VL-72B**        | 24.1% $\\to$ 22.8% ($\\color{red}{\\downarrow 1.3\\%}$) | 42.0% $\\to$ 31.6% ($\\color{red}{\\downarrow 10.4\\%}$) | 39.0% $\\to$ 40.4% ($\\color{green}{\\uparrow 1.4\\%}$)   |
> | **Qwen2.5-VL-7B**         | 15.2% $\\to$ 12.1% ($\\color{red}{\\downarrow 3.1\\%}$) | 15.6% $\\to$ 10.4% ($\\color{red}{\\downarrow 5.2\\%}$)  | 20.4% $\\to$ 17.2% ($\\color{red}{\\downarrow 3.2\\%}$)   |
>
> **Scale-dependent benefits in planning tasks.**
> For planning-heavy tasks such as SuperMario, CoT brings substantial gains for the largest model (Qwen3-VL-235B), with an improvement of **$\\color{green}{\\uparrow 12.3\\%}$**. The 72B model sees only a marginal gain, while the 7B model’s performance decreases, likely due to weaker instruction adherence and increased hallucination in long generations. This indicates that only sufficiently strong models can effectively leverage explicit reasoning chains for long-horizon visual planning.
>
> **“More thinking, less seeing.”**
> In reaction-heavy tasks like Race, CoT consistently degrades performance across all model sizes (e.g., a **$\\color{red}{\\downarrow 10.4\\%}$** drop for the 72B model). This aligns with recent findings [1] suggesting that extensive verbal reasoning can sometimes pull attention away from immediate visual cues, causing models to “overthink” low-level perceptual decisions instead of reacting quickly to the current frame.
>
> Taken together, these prompting ablations support our central claim: **the main bottlenecks in V-MAGE lie in dynamic visual processing and temporal reasoning**, which cannot be overcome simply by adding more in-context examples or generic CoT prompts.
>
> ---
>
> **References:**
> [1] "More Thinking, Less Seeing? Assessing Amplified Hallucination in Multimodal Reasoning Models"

---

> > ### Comment · Reviewer_4tME · 2025-11-25
> >
> > Thank you for the detailed rebuttal and the updates to the paper. That said, I am keeping my overall score at 4, mainly because I still see the benchmark’s value as limited by (1) relatively incremental insights compared to existing work, (2) a somewhat under-theorized connection to “diagnostic generalization,” and (3) unresolved concerns about data leakage and cross-model comparability.
> >
> > 1. Incremental contribution relative to existing benchmarks
> >
> > The main qualitative conclusion of V-MAGE is that state-of-the-art MLLMs can handle relatively simple visual tasks but degrade sharply on dynamic, interactive games. This is an important message, but very similar limitations have already been highlighted in recent game-based and video/physics benchmarks. The related-work section already cites several of these or closely related works, and positions V-MAGE among game-based evaluations. Where V-MAGE is more distinctive is in the specific design choices: visually irreducible, continuous-state games; a focus on visual-only inputs; an ELO-style cross-game rating; and a compact suite of popular arcade titles. These are useful design choices, but for me they amount to a solid incremental benchmark rather than a clearly game-changing one.
> >
> > 2. “Diagnostic benchmark for generalization” feels under-theorized
> >
> > In the rebuttal to another reviewer, the authors emphasize that their primary goal is to establish V-MAGE as a diagnostic benchmark for generalization. I like this ambition, but I do not think the current design is fully aligned with that claim.
> >
> > The paper introduces four “core visual competencies”: Positioning, Tracking, Visual Grounding, and Timing. And uses them as unit-test dimensions for the five games. This decomposition is reasonable at an intuitive level, but the manuscript does not really argue why these four axes, applied to these particular games, should be especially diagnostic of “generalization”. There is no theoretical link to existing taxonomies of generalization or to cognitive-science views of visual generalization.
> >
> > More broadly, the game selection still feels somewhat opportunistic. Appendix C discusses selection principles such as visually irreducible state, continuous dynamics, lightweight environments, and coverage of XY/XZ/3D and linear vs. open layouts. These are good engineering principles, and the new game–level table in the rebuttal clarifies which levels target which micro-abilities (pixel positioning, gravity intuition, visual counting, etc.). But from the lens of “diagnostic generalization,” the choices (Flappy Bird, Mario, Pong, etc.) still read more like convenient/popular minigames than the outcome of a principled search over the space of generalization challenges.
> >
> > The notion of “visually irreducible” is a nice criterion for ensuring that models must truly use pixel-level visual information .
> > However, this is orthogonal to generalization: a game can be visually irreducible yet still be highly in-distribution (because it appears often in web-scale training data), or it can probe narrow skills that do not necessarily generalize beyond its own genre. In other words, “visually irreducible” ensures dependence on vision, but does not by itself make the benchmark a diagnostic test of generalization.
> >
> > 3. Data leakage and cross-model comparability
> >
> > The rebuttal gives a more detailed discussion of possible memorization and distributional overlap. These points make V-MAGE less likely to be trivially solved by memorizing a single fixed trajectory, and I appreciate the clarification. However, my main concern is slightly different and still remains:
> >
> > Different open and closed models have unknown and non-identical pretraining distributions. Some are almost certainly trained on large volumes of gameplay videos, YouTube playthroughs, and tutorial content for these exact games; others may have much less exposure. Even with seeded variability, a model that has seen many Flappy Bird or Mario trajectories can plausibly benefit from that in-distribution exposure in ways we cannot quantify.
> >
> > As a result, differences in ELO ratings across models may partially reflect different degrees of “in-domain familiarity” with the games, not just differences in underlying generalization or visual reasoning ability. This is not unique to V-MAGE—most benchmarks suffer from unknown data contamination—but it is especially salient when the tasks are built around a handful of very famous public games. The current paper does not attempt any systematic contamination audit. Nor does it strongly caveat the cross-model ranking in terms of this limitation.
> >
> > I therefore see V-MAGE as a useful stress test that reveals a persistent dynamic-visual gap even in frontier models (as your extended results with GPT-5 and Gemini 2.5 show), but I would be cautious about drawing strong conclusions from the ELO ordering about which model is “better at generalization” in any deeper sense.

---

> > > ### Author Response · Authors · 2025-12-03
> > >
> > > We appreciate the reviewer’s thoughtful follow-up and the clarity of the concerns raised. We briefly restate how we see the contribution and where our perspective differs.
> > >
> > > First, on the “incremental” nature of V‑MAGE: we agree that several recent works have already shown that current MLLMs struggle on dynamic or interactive tasks. Our goal is not to claim a completely new phenomenon, but to **pull several design choices into a single, reusable benchmark**: (i) strictly visual‑only state (no text/grid reduction), (ii) lightweight but continuous‑state games with controllable difficulty, (iii) a modular game–agent–model pipeline that makes swapping agents and models easy, and (iv) a shared set of capability axes (positioning, tracking, timing, etc.) with corresponding unit tests, error analysis, and ablations on ICL/CoT and anchoring bias. From our vantage point, this combination turns V‑MAGE into more than “yet another game leaderboard”: it is meant to be a **practical diagnostic platform** that many groups can extend and plug into their own agent/model designs.
> > >
> > > Second, regarding the “diagnostic benchmark for generalization” phrasing: we will soften that language. Our intent is to diagnose how well models **generalize within a family of dynamic visual control tasks**, across seeds, layouts, and difficulty settings, rather than address all forms of generalization. The four “core visual competencies” we introduce are meant as **operational dimensions** that make failure modes easier to localize (e.g., “mostly tracking failures vs. mostly timing failures”).
> > >
> > > Finally, on data leakage and cross‑model comparability: we believe V‑MAGE still exposes a **real algorithmic gap**: even with the strongest models we tested, performance remains far below human, and errors on levels like Mario 1‑1 show that prior exposure alone does not translate into robust real‑time control. Our view is that ELO in V‑MAGE should be read as “**relative ability on this specific family of dynamic visual tasks**”, not as a universal ordering of “who generalizes better” in all senses.
> > >
> > > We respect the reviewer’s assessment, but we continue to see V‑MAGE as a useful and needed step: a compact, extensible benchmark that stresses vision‑only, continuous, multi‑frame interaction in a way that is easy to adopt and that surfaces concrete, reproducible failure patterns in current MLLMs.

---

### Official Review · Reviewer_Xs7D · 2025-10-28

**Soundness:** 2
**Presentation:** 3
**Contribution:** 2
**Rating:** 4
**Confidence:** 3

**Summary:**

This paper introduces V-MAGE, an interactive and visually rich evaluation framework focused on dynamic interaction and vision-centric reasoning (also serving as a sandbox for vision agent development). Using V-MAGE, the authors evaluated various public MLLMs with ELO scores, highlighting a significant gap between model performance and human proficiency in complex tasks. They further analyzed why current MLLMs perform poorly in video game tasks, including deficiencies in fundamental visual capabilities, reasoning challenges during prolonged interactions, and issues like anchoring bias.

**Strengths:**

1. Addresses the inability of existing static image-text benchmarks to evaluate MLLMs’ dynamic perception and interactive reasoning; it is both an evaluation framework focusing on "dynamic interaction + vision-centric reasoning" and a sandbox for vision agent development.
2. Adopts a dynamic ELO ranking system to resolve score scale differences across games, enabling standardized cross-model comparison and accurate measurement of incremental progress in non-linear scoring tasks.
3. Quantifies MLLMs’ performance via ELO scores (e.g., near human level in simple tasks, large gaps in complex tasks) and analyzes poor performance causes (visual capability deficiencies, reasoning bottlenecks, etc.), providing clear directions for MLLMs optimization.

**Weaknesses:**

1. Only 5 human participants were invited to establish the performance baseline; the small sample size may affect the representativeness and robustness of the human benchmark.
2. In-depth analysis of model errors (e.g., manual annotation of error types) focuses mainly on GPT-4o, with insufficient coverage of error characteristics of other models, limiting the generalizability of conclusions.

**Questions:**

1. What specific mechanisms does V-MAGE’s ELO rating system use to address the issue of score scale differences across different games?
2. Model error analysis focuses mainly on GPT-4o, and no equally in-depth error annotation is conducted for other models (e.g., Qwen2.5VL-72B, InternVL2.5-78B); are there technical or resource constraints behind this choice?
3. The paper mentions that V-MAGE’s game levels have the "visually irreducible" feature; how is this feature reflected in specific levels (e.g., Race or SuperMario)?

---

> ### Author Response · Authors · 2025-11-21
>
> We sincerely thank the reviewer for the constructive feedback and for explicitly recognizing V-MAGE's dual value as both a rigorous evaluation framework and a **sandbox for vision agent development**. We appreciate your guidance on improving the robustness of our baselines and analysis. Below, we address your questions with expanded experiments and clarifications.
>
>
> > **Q1. Robustness of human baselines**
>
> In the original submission, our human baselines were obtained from 5 participants. Following the reviewer’s advice, we expanded the human test group to **10 participants** under the same protocol. The aggregate statistics and qualitative trends remained highly consistent with our initial findings.
>
> Across the levels included in V-MAGE (excluding the more open-ended TempestRun), we observed a clear **ceiling effect**: for bounded tasks such as Race, Pong, and the simplified FlappyBird/SuperMario levels, almost all participants reached the maximum or near-maximum achievable scores. In the additional runs collected during rebuttal, we recorded only a very small number of failures (e.g., a single missed trial in FlappyBird where the participant did not reach the capped score).
>
> This indicates that, for the current set of levels, the human baseline effectively represents a stable “near-perfect” reference. In contrast, state-of-the-art MLLMs remain far below this ceiling across all games. The substantial model-human gap we report is therefore a robust phenomenon, and not an artifact of limited or noisy human data.
>
> > **Q2. Mechanism of the ELO rating system**
>
> V-MAGE uses an ELO-based system specifically to address the problem of comparing heterogeneous game metrics. Directly averaging raw scores across games is problematic because scales differ (e.g., typical Pong scores are on the order of $0$-$20$, TempestRun scores can reach $0$-$2000$) and difficulty is non-linear.
>
> Our system converts raw performance metrics (whether time, distance, or points) into a unified **probability of winning**. We simulate pairwise matches between Model A and Model B on the same level. If Model A's performance metric > Model B's, Model A "wins". The ELO algorithm aggregates the win/loss/draw outcomes from these pairwise comparisons across all models and levels.
>
> This converts diverse task-specific metrics into a single **relative skill scale**: for example, a 100-point ELO advantage corresponds to a similar increase in expected win probability regardless of whether the underlying game is Race or SuperMario. This is what enables cross-task interpretability that simple averaging of raw scores cannot provide.

---

> ### Author Response · Authors · 2025-11-21
>
> > **Q3. Expanded error analysis on Qwen3-VL-235B-A22B**
>
> Our original error analysis for V-MAGE was based on manual annotation of approximately 500 model-environment interaction snippets (inputs and outputs), focusing on GPT-4o. We relied on human annotators rather than automated evaluators (e.g., GPT-based judges) to avoid introducing additional uncertainty into the labeling process. While this ensures accuracy, it also makes large-scale analysis labor-intensive.
>
> To assess whether our conclusions generalize beyond GPT-4o, we conducted a parallel manual annotation study (approximately 600 samples) on the newly released **Qwen3-VL-235B-A22B-Instruct**. In our updated leaderboard, this model achieves an ELO of **1515.4**, performing on par with **Gemini-2.0-Flash-Thinking (1515.5)** and **Qwen2.5-VL-72B (1515.0)**, i.e., among the strongest open-weight models we evaluated.
>
> The distribution of primary error types for Qwen3-VL-235B-A22B is summarized below:
>
> | Error Type               | FlappyBird | Pong | Race | SuperMario | TempestRun |
> | :----------------------- | :--------: | :--: | :--: | :--------: | :--------: |
> | **no error**             | 44         | 20   | 52   | 90         | 22         |
> | **perception error**     | 35         | 15   | 14   | 25         | 57         |
> | **direction error**      | 3          | 18   | 12   | 17         | 9          |
> | **perception incomplete**| 10         | 0    | 6    | 8          | 12         |
> | **reasoning error**      | 3          | 5    | 10   | 0          | 5          |
> | **history misinterpretation** | **41** | 1    | 11   | 15         | 3          |
> | **action inappropriate** | 0          | 0    | 6    | 0          | 0          |
> | **inconsistency**        | 10         | 5    | 15   | 0          | 0          |
> | **instruction following**| 0          | 0    | 0    | 0          | 0          |
>
> *(A visualization figure analogous to Fig. 7 will be included in the final version.)*
>
> In **TempestRun**, Qwen3-VL-235B-A22B shows a disproportionately high number of **perception errors** (57) compared to the 2D games. This contrasts with GPT-4o’s error profile and suggests that even very capable open models still struggle with depth and 3D spatial interpretation.
>
> The large number of **history misinterpretation** errors in **FlappyBird** (41) is consistent with what we observed for other models and aligns with our analysis of **anchoring bias**: models tend to over-rely on past textual descriptions and underweight subtle visual changes in consecutive frames.
>
> These findings indicate that the limitations we highlight (especially in multi-frame perception and history-sensitive reasoning) are not specific to a single model family, but appear across different architectures and training regimes.
>
>
> > **Q4. "Visually Irreducible" Features**
>
> We define a game as **visually irreducible** if its underlying state and optimal decisions depend on **continuous spatial/temporal nuances** that cannot be faithfully reduced to a low-dimensional grid or a small set of discrete symbols without losing crucial information.
>
> Games such as chess or 2048 are “reducible” in this sense: the entire state can be encoded as a discrete matrix (e.g., piece types on a board), and a text-only description suffices. The visual channel adds little beyond what can be obtained via OCR.
>
> In contrast, V-MAGE games are designed so that solving them requires leveraging fine-grained visual information:
>
> **SuperMario.** Success often depends on **pixel-level distances**: whether Mario clears a gap or clips a platform edge hinges on his exact horizontal and vertical offsets and momentum. A coarse text description like “Mario is near the pipe” does not capture whether a given jump is feasible.
>
> **FlappyBird:** The decision relies on an analog judgment of the bird's parabolic trajectory and velocity relative to the moving gap. This requires a physical intuition from the visual stream that is lost when converted to static text logs.
>
> **Race:** Control is governed by **continuous dynamics** (speed, acceleration, turning radius), not discrete grid moves. The model must infer acceleration and inertia from multi-frame visual differences and time its steering/braking accordingly. A static coordinate log of car and trophy positions cannot fully encode these momentum-related aspects.
>
> We selected these environments specifically to **require models to perform genuine visual-spatial reasoning** over sequences of frames, rather than treating the image as a mere carrier for text-like symbolic information. This is what makes V-MAGE complementary to existing grid-based or text-reducible game benchmarks.
>
> We hope these clarifications address the reviewer’s concerns about human baselines, the ELO mechanism, extended error analysis, and the notion of “visually irreducible” tasks in V-MAGE.

---

### Official Review · Reviewer_aqeE · 2025-11-04

**Soundness:** 2
**Presentation:** 2
**Contribution:** 2
**Rating:** 4
**Confidence:** 4

**Summary:**

This paper introduces a new benchmark V-MAGE to measure how well MLLMs like GPT4o or Gemini can perform real-time visual reasoning in fixed-screen and platformer video games - simulated environments that are reminiscent of interactive and continuous spaces.

**Strengths:**

I personally find this work possesses the following strong aspects:

1. A refreshing angle to tackling a widely recognized issue.
2. Good awareness on shortcut prevention as in Chap 3.2 - It’s quite commendable to see the authors make sure the visual inputs cannot be easily converted to simple texts.

**Weaknesses:**

I find the work at its current stage suffers from the following major weak points.

1. **(Soundness) Ambiguity in the ELO-Based Evaluation Metric.** Although I personally have a brief knowledge of ELO ranking in chess and go, the description of the ELO ranking mechanism used in this work is not fully intuitive for readers unfamiliar with it. In fact, the current metric is inconsistent on what constitutes a “win” or “loss” in each game - whether success corresponds to surviving longer (e.g. Flappy Bird), completing levels without failure (e.g. Super Mario), or achieving higher raw scores (e.g. Pong). The criteria for scoring the same ELO score are different across the games included in this study. We would need a clearer definition of **unified evaluation standards** for better transparency and interpretability.

2. **(Presentation) Limited Robustness of Human Baselines.** The reported human baselines may lack robustness, as individual skill levels and familiarity with specific games can strongly influence performance. This unpredictable variability in skill weakens the reliability of the human-based reference baselines shown in Figure 4 and 6. To counter this, I encourage the authors might consider incorporating **Tool-Assisted Speedruns (TAS)** or other controlled, reproducible playthroughs to establish a more stable *theoretical upper bound* on human-level performance. For example, many TAS runs for Super Mario Bros in video format can be found on: https://tasvideos.org/1G .

3. **(Contribution) Unclear Downstream Usage of V-MAGE.** While V-MAGE is proposed as an evaluation benchmark, its broader practical utility is not well demonstrated. To be specific, *it is unclear to me how improved performance on V-MAGE translates to advancements in real-world or practical multimodal reasoning tasks.* Can the samples in V-MAGE be integrated into **post-training or reinforcement learning pipelines** (e.g., SFT, DPO, or GRPO) to enhance visual reasoning? Unfortunately, this work has not yet demonstrated much concrete proof for its relevant downstream usage.

**Questions:**

Please find my major concerns in the Weakness section. Out of the three concerns, I suggest the authors prioritize the 1st and the 3rd, given the time limit for rebuttal.

---

> ### Author Response · Authors · 2025-11-21
>
> We thank the reviewer for the thoughtful feedback and for highlighting both our **shortcut-prevention design** (Sec. 3.2) and the potential of V-MAGE. We also appreciate the concrete suggestions regarding ELO clarity, human baselines, and downstream usage. We address these points below.
>
> > **Q1. Clarifying the ELO metric and “win” conditions**
>
> We agree that the ELO mechanism can be non-intuitive for readers who are less familiar with rating systems, and we will further clarify it in the final version with **per-game definitions of the performance function $f(m)$**.
>
> In V-MAGE, $f(m)$ is the **raw game score**. For the **per-game score component** we use:
>
> **Binary scoring (Race).**  In Race, the environment score reflects **task completion**: +100 for successfully reaching the trophy, 0 otherwise. Thus the scalar score is effectively binary, and $f(m)$ distinguishes models based on success rate together with valid-action rate.
>
> **Cumulative scoring (Pong, FlappyBird, SuperMario, TempestRun).**  In these games, the score is cumulative and reflects **continued progress or successful interactions**: number of ball returns in Pong, pipes passed in FlappyBird, coins and enemies in SuperMario, and distance in TempestRun.
>
> As detailed in **Appendix D (Eq. 1-3)**, the ELO calculation relies on the outcome of **pairwise comparisons** based on these scores:
> $$
> S_A = \\begin{cases}
> 1 \\text{ (Win)} & \\text{if } f(A) > f(B) \\\\
> 0 \\text{ (Loss)} & \\text{if } f(A) < f(B)
> \\end{cases}
> $$
>
> A key advantage of this ELO formulation is that it **normalizes heterogeneous raw metrics**. Different games use different native scales (e.g., “seconds survived” in FlappyBird vs. “coins collected” in SuperMario), but the ELO rating reflects a model’s **probability of outperforming other models** in pairwise comparisons. Thus, an ELO of 1600 in Pong and in SuperMario represents a similar relative capability level within the evaluated model population, which is not achievable by comparing raw scores directly.
>
> We will add a short table in the camera-ready version that explicitly lists the score definition for each game (Race, Pong, FlappyBird, SuperMario, TempestRun), to make the mapping from environment score to $f(m)$ completely transparent.

---

> ### Author Response · Authors · 2025-11-21
>
> > **Q2. Robustness of human baselines and TAS feasibility**
>
> We share the reviewer’s concern that human skill can vary across individuals and games. In the original submission we reported results from **5 participants**. During the rebuttal period, we additionally collected data from **5 more participants** under the same protocol (for a total of $N=10$), and the aggregate statistics and qualitative trends remained consistent with the originally reported baselines.
>
> Our intention is to use human results as a **soft reference point** rather than a mathematically precise upper bound. We would like to clarify why, despite variance, they are still informative:
>
> **Ceiling effects in bounded games.** For some levels (e.g., simplified FlappyBird variants), we cap the effective human score at a moderate value (e.g., 10). Most human players reach or approach this cap quickly, so inter-person variance is small relative to the model-human gap.
>
> **Order-of-magnitude gaps in unbounded games.**  In unbounded settings such as TempestRun, human scores do show variability (e.g., in the range of roughly 800-2000 in our study). However, current state-of-the-art MLLMs typically remain far below this range (often below 500 and frequently under 200). This **order-of-magnitude difference** suggests that, at the current stage, the exact human upper bound is less critical for the main qualitative conclusion: models are still far from robust human-level performance in dynamic visual tasks.
>
>
> Regarding the suggestion of **Tool-Assisted Speedruns (TAS)**: we agree that TAS runs can, in principle, provide a very tight upper bound in classic deterministic emulators (e.g., NES Super Mario Bros.). In V-MAGE, each episode of a given level is initialized with a **seed-controlled random configuration** (e.g., different starting positions in Race, different obstacle patterns in TempestRun). This means that:
>
> - for a fixed seed, a run is reproducible;
> - across episodes, models must handle a **distribution over initial conditions**, rather than a single fixed trajectory.
>
> Adapting TAS-style scripts to this setting would therefore require not only replaying a single optimal trajectory, but designing and maintaining **scripted control policies** that are robust across many seeded variants. Implementing and validating such a TAS pipeline on top of our current PyGame-based environments would require substantial additional engineering and is not yet supported in our codebase.
>
> Given the large model-human performance gap we already observe, we believe that our current human baseline, while imperfect, is sufficient as a practical reference point at this stage. Incorporating TAS-like controlled upper bounds is an interesting direction we would like to explore in future extensions of V-MAGE.
>
> > **Q3. Downstream usage**
>
> We agree with the reviewer that V-MAGE has significant potential beyond evaluation. Based on our framework design, V-MAGE can serve as a **high-quality data generator**:
>
> Because V-MAGE’s game states are fully accessible via code, we can programmatically convert internal states (e.g., exact obstacle coordinates, relative velocities, collision events) into rule-based **captions** or **chain-of-thought-style reasoning traces**. This enables batch generation of **multimodal instruction data** whose labels are derived directly from ground-truth simulator states, leading to much lower annotation noise than is typical for real-world video datasets. Such data could be used for supervised fine-tuning (e.g., instruction tuning) or preference-based methods (e.g., DPO) to strengthen models’ temporally grounded visual reasoning.
>
> Furthermore, V-MAGE naturally functions as an **RL / agent-training environment**. Its scoring systems (e.g., +1 for passing a pipe, binary success in Race, game-over signals) provide well-defined reward signals and termination conditions, offering the kind of “outcome supervision” required by reinforcement learning algorithms (including GRPO-style methods).
>
> However, our primary goal in this work is to establish V-MAGE as a **diagnostic benchmark for generalization**. If models were heavily trained on these specific games, they might overfit to particular physics and layouts, becoming V-MAGE “specialists” rather than revealing the **emergent visual reasoning** capabilities of general-purpose MLLMs. Therefore, in this paper we intentionally evaluate largely **pre-trained models** without V-MAGE-specific fine-tuning, and we position V-MAGE primarily as a rigorous *test set* for tracking progress, while recognizing its potential as a *training environment* as an important direction for future research.
>
> We hope these clarifications address the reviewer’s concerns about evaluation consistency, human baselines, and the broader practical utility of V-MAGE.

---

> ### Comment · Reviewer_aqeE · 2025-11-25
> **My takeaways to the authors' response**
>
> I appreciate the authors' clarifications. Please find my comments below regarding each respective point.
>
> > **Q1 - The fundamental cross-game incomparability of the ELO ranking metric**
>
> I respectfully disagree with the authors' implication that ELO **normalizes heterogeneous raw metrics** across multiple games. Rather, the ELO metric only **normalizes and rescales the raw metric that is exclusive to a specific game**. In real-life scenarios, it only makes sense to compare the ELO rankings among all Chess players in one leaderboard, and the ELO rankings among all the Go/Weiqi players in a different leaderboard. However, a 2800-ELO-score player in Chess (like Magnus Carlsen) **CANNOT** be fairly compared with a 2800-ELO-score player in Go (like Shin Jin-seo) in any sort of equivalent capability, since the rules of the games are fundamentally different to begin with.
>
> The 5 games incorporated in the V-MAGE benchmark are similar to the aforementioned real-life examples - each game is supposed to own its exclusive scoreboard, and **only the entries within the same leaderboard are comparable to each other**. Although the level layouts can be deterministically controlled with fixed random seeds according to the authors' explanation, each game fundamentally requires a distinct set of strategies in order to achieve high scores. Therefore, unfortunately, as long as V-MAGE keeps this ELO-based metric as its evaluative basis, its benchmarking baselines would be filled with aggregated apple-to-orange comparisons across incomparable game scores.
>
> > **Q2 - On human and TAS baselines**
>
> I do agree that the TAS baselines are infeasible to obtain due to the time limit. Still, the authors do reaffirm my original worry that the current-stage human baselines in V-MAGE are indeed varying and un-robust performative upper bounds.
>
> > **Q3 - Downstream usage of V-MAGE**
>
> I appreciate that the authors propose many potential downstream use cases for V-MAGE. It is just not very convincing without seeing any concrete qualitative or quantitative evidence to confirm that V-MAGE *can* help in those use cases.

---

> > ### Author Response · Authors · 2025-11-25
> >
> > We thank the reviewer for taking the time to provide follow-up comments. We respond to each point below.
> >
> > > **Q1 – On cross-game comparability of the ELO metric**
> >
> > We fully agree that *directly* comparing raw scores across fundamentally different games (or sports) is not meaningful, and V‑MAGE does **NOT** do so at the scoring stage.
> >
> > In our framework, ELO is computed **within each individual level**, based only on pairwise comparisons of model scores on that **same** level. In other words, every level has its **own implicit leaderboard**: models are only “matched” against each other under identical rules and layouts.
> >
> > As shown in Table 2, for presentation we then average these per-level ELOs along the game dimension (and finally across games) to give a compact overall summary. This averaging is a *post-hoc aggregation for readability*, not how the ratings are originally computed.
> >
> > Using your analogy, what we are doing is closer to the following:
> >
> > - For each board game or variant (e.g., Go, chess, a house-rule chess where knights can also move diagonally, etc.), we maintain its **own** ELO leaderboard and rate players within that specific rule set.
> > - At the end, if we wish to talk about a player’s “overall board-game strength”, we might average their ratings across several such leaderboards as a rough integrated indicator, while still retaining the per-game breakdown.
> >
> > This is exactly how we use the cross-game average in V‑MAGE: as a coarse indicator of a model’s **aggregate** dynamic-vision competence, while all concrete judgments about behavior are made and analyzed at the per-level and per-game granularity.
> >
> > We also agree that different games require different surface strategies. Our intention, however, is to treat them as **emergent basic capabilities** from large-scale pretraining, rather than as specialized domains like professional chess or Go that require years of focused human training. From this perspective, averaging over several “micro-environments” that all exercise similar foundational skills is still informative, as long as the per-game breakdown remains available, which we do provide.
> >
> > We will clarify this computation–aggregation separation more explicitly in the paper to avoid the impression that heterogeneous raw scores are directly mixed during rating.
> >
> > > **Q2 – On human and TAS baselines**
> >
> > TAS is not only an issue due to the time limit, but also fundamentally constrained by randomness: it is impossible for us to record a TAS run for every random seed. In addition, as we have already mentioned, the human baselines on these relatively simple games show almost no variance, and in most cases even a complete novice human player can quickly reach the upper bound we set. Therefore, we believe the current human baselines are sufficient to reflect model capability.
> >
> > We have extensively expanded our benchmark to include the latest state-of-the-art models, including **GPT‑5**, **GPT‑5.1 (basic)**, **Gemini 2.5 Pro Thinking**, **Gemini 2.5 Flash**, and **Qwen3‑VL**. As can be seen from the results below, even the strongest systems so far only reach **43.4%** of the human baseline, which is still far from the regime where we would need to worry about the small variance in human performance.
> >
> > |Model|Avg ELO Rating|Performance Ratio (%)|
> > |:-|:-:|:-:|
> > |**GPT‑5‑2025‑08‑07**|**1709.6**|**43.4**|
> > |**Gemini‑2.5‑Pro**|**1604.0**|**36.3**|
> > |**Claude‑3.7‑Sonnet**|**1542.9**|**30.8**|
> > |**Gemini‑2.5‑Flash**|**1528.3**|**23.8**|
> > |GPT‑4o|1526.4|26.6|
> > |Gemini‑2.0‑Flash‑Thinking|1515.5|22.6|
> > |**Qwen3‑VL‑235B‑A22B‑Instruct**|**1515.4**|**24.3**|
> > |Qwen2.5‑VL‑72B‑Instruct|1515.0|22.8|
> > |InternVL2.5‑78B|1492.2|19.2|
> > |**GPT‑5.1‑2025‑11‑13**|**1486.3**|**20.1**|
> > |Gemini‑2.0‑Flash|1484.2|16.7|
> > |Qwen2‑VL‑72B‑Instruct|1473.2|16.5|
> > |InternVL2.5‑8B|1441.2|12.9|
> > |Qwen2.5‑VL‑7B‑Instruct|1439.3|12.1|
> > |Qwen2‑VL‑7B‑Instruct|1437.5|11.4|
> > |Keye‑VL‑8B‑Preview|1434.2|13.1|
> > |Phi‑4‑multimodal‑instruct|1429.2|13.7|
> > |Random|1425.6|11.0|
> >
> > Given this large gap, we believe the current human baselines are sufficiently robust to reflect model capability, and that their small residual variance does not materially affect our claims about the limitations of today’s MLLMs on V‑MAGE.
> >
> > > **Q3 – On downstream usage of V‑MAGE**
> >
> > We would like to reiterate that our primary goal is to provide a **diagnostic benchmark** for evaluating models’ visual reasoning abilities. We will seriously consider your suggestions, but running sufficiently robust experiments along these lines is indeed difficult to complete within the current time frame. At this stage, we still prefer to position V‑MAGE mainly as a **test environment**. We again thank you for your thoughtful ideas and suggestions.

---

> ### Comment · Reviewer_aqeE · 2025-11-26
> **The fundamental flaws in the design**
>
> I appreciate the authors' follow-up to address my original concerns. However, upon examining the authors' exchanges with other reviewers, I am more than convinced that the framework of V-MAGE, at its current state, contains many fundamental flaws in its design, as I come to agreement with a lot of Reviewer 4tME's findings.
>
> > **The built-in flaws of using ELO ranking as a diagnostic metric for generalization, and the problematic benchmark design choices in general.**
>
> I understand that each game-wise ELO ranking is computed over separate intensified levels - I have no problem with this particular design at all. However, the true problem is that **the ELO ranking fundamentally is domain/game-specific** - ELO is a heavily biased measure that naturally incorporates large amounts of 'in-domain familiarity' (as Reviewer 4tME puts it). I acknowledge the authors may argue that the 5 games included in V-MAGE all constitute the very same set of **generalizable atomic capabilities**, i.e. common input commands and visual interpretation skills, etc. However, the actual presentation shows quite the contrary - according to Appendix C, every game technically must leverage a unique set of visual perceptions (Observation Space) and must employ a distinct set of input commands (Action Space). Unfortunately, the authors haven't shown enough theoretical or empirical evidence to convince me why the ELO scores by those examined MLLMs in Table 2 are **true and faithful reflections of game-invariant emergent basic capabilities**. Personally speaking from a speedrunner's perspective, beating the hardest levels with the highest points in a horizontal platformer like Super Mario can hardly be translated to surviving the longest time in Flappy Bird or Tempest Run. The 5 games are already too genre-specific to be generalizable under a universal ELO-based scale.
>
> As a closing suggestion, to strengthen the argument of V-MAGE's generalization purpose, I would recommend the authors incorporate additional game instances (environments) that are in the same game genres with the current 5 in V-MAGE. For example, I highly recommend they check out the toolkits of *Gymnasium*  and *Atari* for constructing controllable environments that resemble classic arcade or NES games.
>
> With all being said, I decide to keep my original rating of 4.
>
> **References**
> - Gymnasium (formerly OpenAI Gym): https://gymnasium.farama.org/
> - Atari: https://ale.farama.org/

---

> > ### Author Response · Authors · 2025-12-03
> >
> > We thank the reviewer for sharing this perspective and for the concrete suggestions.
> >
> > On the role of ELO, we agree that it is inherently domain-specific and should not be read as a universal scale across all possible games. In V‑MAGE we use ELO in a narrower way: it is computed **within each level** to capture relative robustness between models under the same rules and layout, and then averaged to summarize performance **within this particular family of dynamic visual tasks**. Our intent is **not** to claim that an ELO score on Super Mario is **directly “equivalent”** to an ELO score on Flappy Bird in a game-agnostic sense, but to provide a compact indicator of how well a model handles this small, shared set of vision-only, multi-frame control problems. The detailed per-game and per-level breakdowns are what we rely on for diagnosis; the averages are mainly for summarizing trends.
> > Regarding the benchmark design, we see the main contribution of V‑MAGE as providing a **small, controlled core** of visually irreducible, continuous-state environments, with a unified interface and analysis pipeline (unit tests, error taxonomy, anchoring-bias and CoT/ICL ablations). This is meant to make it easy to plug in new models or agents and immediately see where they fail in multi-frame perception and action, rather than to cover the full space of possible game genres.
> >
> > We agree that adding more environments within the same genres (e.g., Gymnasium/Atari-style variants of side-scrollers, runners, or paddle games) would strengthen the generalization story and reduce dependence on any single iconic title. That is very much in line with how we intend V‑MAGE to evolve: the current paper focuses on establishing and analyzing a compact core; extending the suite with additional, genre-consistent games is a natural next step that our framework is explicitly designed to support.

---

### Official Review · Reviewer_4GB9 · 2025-11-06

**Soundness:** 3
**Presentation:** 3
**Contribution:** 2
**Rating:** 4
**Confidence:** 3

**Summary:**

This paper introduces V-MAGE, a game-based evaluation framework designed to assess the vision-centric reasoning and interaction capabilities of multimodal large language models (MLLMs). The benchmark consists of five visually rich video games with over 30 dynamic levels, requiring models to perform perception, temporal reasoning, and decision-making based solely on visual inputs. V-MAGE employs a dynamic ELO-based ranking system for consistent model comparison. Experimental results across leading MLLMs (e.g., GPT-4o, Gemini, Qwen-VL) and human baselines reveal a clear gap between models and humans in dynamic, vision-grounded tasks, highlighting current MLLMs’ limitations in temporal reasoning and interactive planning.

**Strengths:**

- The paper presents a well-designed evaluation framework. In particular, the introduction of a dynamic ELO-based ranking system provides a unified and interpretable metric across heterogeneous game environments, effectively mitigating the bias caused by varying score scales and task difficulty levels.

- The study offers comprehensive experimental details, including environment configurations, prompt templates, and implementation settings in the appendix. This high level of transparency ensures that the results are fully reproducible and facilitates fair benchmarking and future extensions.

**Weaknesses:**

- **Overemphasis on Visual-Only Modality**: The evaluation framework focuses exclusively on visual input, omitting textual or memory-based information that is often present in real-world multimodal scenarios. This design choice, while emphasizing visual reasoning, may cause a distributional mismatch with practical multimodal applications that rely on both linguistic and contextual cues.

- **Simplified Game Environments**: Although the selected games (e.g., FlappyBird, Pong) introduce dynamic visual elements, they remain relatively low in semantic complexity. As a result, the benchmark may not fully capture the semantic understanding and strategic planning challenges encountered in more realistic interactive environments.

**Questions:**

Please refer to the weakness part.

---

> ### Author Response · Authors · 2025-11-21
>
> We thank the reviewer for the constructive feedback and for recognizing both the **soundness of our ELO-based ranking system** and the **transparency and reproducibility** of our framework. We also appreciate your thoughtful concerns about the modality scope and the semantic complexity of the environments. Below, we clarify our design rationale and positioning.
>
> > **Q1. "Visual-only" modality and real-world alignment**
>
> We agree that many real-world tasks interleave visual, textual, and memory-based information. Our choice to restrict the *state* channel in V-MAGE to pixels (while still providing textual instructions and rules) is deliberate, and aims to address a critical blind spot in current evaluations.
>
> **Decoupling visual perception from textual shortcuts.**
> Recent work has shown that many MLLMs exhibit strong “text bias” [1, 2], often solving multimodal benchmarks by exploiting textual cues (e.g., OCR, captions, overlays) rather than engaging in genuine visual reasoning. By keeping the game state purely visual, V-MAGE acts as a **controlled stress test** of the **visual reasoning pathway**: strong performance cannot be achieved by reading text off the screen or relying on pre-existing textual descriptions of states. This helps isolate whether models can truly perceive and track objects, motions, and layouts from pixels.
>
> **Relevance to “visual-first” applications.**
> While text is ubiquitous, many important applications are inherently visual-first. For example, in **GUI agents**, a model often must detect subtle but critical visual changes (e.g., a loading spinner stopping, a disabled button becoming active) without any accompanying textual signal. In **robotics**, sensors provide raw visual or depth data rather than semantic descriptions. V-MAGE targets this foundational layer of perception-driven decision-making, which higher-level multimodal reasoning ultimately depends on.
>
> We view V-MAGE as complementary to existing text-vision benchmarks: it deliberately stresses the visual channel in isolation so that future extensions (e.g., adding richer textual context or memory modules) can be evaluated on top of a well-understood perceptual core.
>
> > **Q2. Simplified environments and strategic planning**
>
> We agree that games like *Pong* and *FlappyBird* are semantically simpler than complex strategy or story-driven games. This simplification is intentional and aimed at achieving **diagnostic precision**.
>
> **“Deceptively simple” but dynamically challenging.**
> Our results show that even these semantically simple tasks remain very challenging for current SOTA models. For example, in **Pong Level 1** (Table 6), GPT-4o achieves an average score of about **0.51 per round**, compared to **10.0** for human players. Despite the low semantic load, models struggle with **dynamic tracking, velocity estimation, and temporally consistent control**. This suggests that fundamental dynamic visual skills are not yet solved, even before introducing rich semantics or long-horizon narratives.
>
> **Isolating the core bottleneck.**
> More complex environments inevitably introduce additional confounders (game lore, long-term objectives, domain knowledge, etc.). By using controlled, visually irreducible yet semantically simple settings, we can more confidently attribute failures to specific components of the vision-reasoning-action loop (for example, tracking errors, misestimation of motion, or anchoring bias in multi-frame reasoning) rather than to confusion over complex rules or storylines.
>
> In this sense, the capabilities tested in V-MAGE serve as **“visual primitives”**. If a model cannot reliably track and react to a moving object in a constrained 2D setting, it is unlikely to succeed in semantically rich 3D environments that require long-horizon planning and interaction. Therefore, we position V-MAGE not as a replacement for more complex interactive benchmarks, but as a **rigorous diagnostic tool** for the dynamic, vision-centric layer that underlies them. Extending V-MAGE with richer semantics and additional modalities is a natural next step, once these core deficits are better understood and addressed.
>
> ---
>
> **References:**
>
> [1] *Words or Vision: Do Vision-Language Models Have Blind Faith in Text?*
> [2] *Unveiling Intrinsic Text Bias in Multimodal Large Language Models through Attention Key-Space Analysis*

---

### Official Review · Reviewer_J2KG · 2025-11-12

**Soundness:** 2
**Presentation:** 2
**Contribution:** 2
**Rating:** 4
**Confidence:** 3

**Summary:**

## Summary
This paper proposes V-MAGE, a game-based, vision-centric evaluation suite for Multimodal Large Language Models (MLLMs). The framework spans 5 video games and over 30 levels, featuring a minimal agent wrapper, multi-frame inputs, and an ELO-style comparative ranking system for evaluating models across heterogeneous tasks. Experiments cover a range of state-of-the-art models, report notable human–model performance gaps, and include unit tests for core visual abilities (positioning, tracking, visual grounding, and timing). Additionally, the paper analyzes anchoring bias in model inference and presents a hand-annotated error taxonomy. Sections on ethics and reproducibility are included, and the appendices provide details on implementation procedures and prompt designs.


## Strengths
- **Clear Problem Framing**: The paper compellingly argues that grid-based or text-reducible games fail to adequately stress vision-centric competencies of MLLMs, while free-form video games serve as an appropriate stress test for such abilities.
- **Comprehensive Decomposition and Coverage**: The framework includes 5 distinct games, a multi-level design, and unit tests that map to interpretable sub-skills—with weaknesses in tracking and timing being particularly insightful.
- **Cross-Task Comparability**: The ELO-based ranking system delivers a unified, scale-robust metric for comparing models across diverse tasks, while still retaining per-level diagnostic capabilities to identify fine-grained performance differences.
- **Robust Error and Bias Analysis**: The quantification of anchoring bias and qualitative breakdown of model errors go beyond mere leaderboard reporting, providing actionable insights to guide the community in addressing specific model limitations.
- **Human Baselines and Ablation Studies**: Incorporating human gameplay baselines and a “perception bypass” ablation (via textual descriptions of game states) strengthens the claim that both visual perception and downstream reasoning jointly constrain model performance.


## Weaknesses
1. **Insufficient Root Cause Analysis**: While the paper effectively documents the existence of model limitations (e.g., poor tracking ability), it lacks a deeper exploration of their underlying causes. For instance, are these deficits attributable to architectural constraints (e.g., inadequate temporal modeling in vision encoders), gaps in training data (e.g., limited exposure to dynamic visual sequences), or inference inefficiencies (e.g., suboptimal frame sampling or context window limitations)?
2. **Limited Connection to Real-World Model Capabilities**: The paper only evaluates model performance through game-specific scores, with insufficient analysis of how these scores correlate with the models’ abilities in real-world vision-centric scenarios


## Questions
Q1: Could you supplement the ablation analysis to disentangle whether model limitations stem from the agent framework (e.g., agent-side processing bottlenecks) versus inherent deficiencies in the MLLMs themselves? (Addressed to Weakness 1)
Q2: Could you demonstrate the connection between models’ performance on V-MAGE and their capabilities in real-world tasks to further validate the benchmark’s effectiveness? (Addressed to Weakness 2)

**Strengths:**

See Summary

**Weaknesses:**

See Summary

**Questions:**

See Summary

---

> ### Author Response · Authors · 2025-11-21
>
> We thank the reviewer for the constructive and detailed feedback, and we appreciate the recognition of our problem formulation, the decomposition of visual sub-skills, and the strengths of our ELO-based comparative framework and error/bias analyses. Below, we address the reviewer’s two main questions with additional experimental evidence and clarifications.
>
> > **Q1: Distinguishing agent-side effects from inherent model limitations**
>
> We agree that separating agent-side bottlenecks from intrinsic model deficiencies is critical. Our ablation studies in **Appendix E.1** and **F.3.2** were designed precisely with this goal in mind. Overall, the results strongly suggest that the observed limitations stem primarily from **model-side factors (architecture and training data)** rather than from constraints imposed by our minimal agent.
>
> **1. Evidence that the agent is not the primary bottleneck**
>
> If the agent-side information pipeline were the main limiting factor, enriching it (longer history or denser sampling) should bring substantial improvements. However, both interventions only produced marginal changes:
>
> **History length.** As shown in **Table 29 (Appx. E.1)**, extending the history window from 3 to 8 steps resulted in negligible performance changes for Qwen2.5-VL-7B (12.68% $\rightarrow$ 12.14%).
>
> | Model | 3 steps (Default) | 5 steps | 8 steps |
> | :--- | :---: | :---: | :---: |
> | **Qwen2.5-VL-7B** | 12.68 | 11.78 | 12.14 |
> | **Qwen2.5-VL-72B** | 21.74 | 20.20 | 23.14 |
>
> *(Scores represent % performance relative to the human baseline across games.)*
>
> **Sampling frequency.** In **Table 36/37 (Appx. F.3.2)**, increasing sampling from 8 frames to 1 frame (providing the model with near-real-time visual updates) only modestly improved the 72B model’s average score (14.65 $\rightarrow$ 18.03):
>
> | Sampling strategy | 8 frames | 5 frames | 3 frames (Default) | **1 frame (Dense)** |
> | :--- | :---: | :---: | :---: | :---: |
> | **Avg. score (72B)** | 14.65 | 16.40 | 16.65 | **18.03** |
>
> We observe qualitatively similar patterns across both model sizes, suggesting that simply providing richer or denser inputs to the agent does not meaningfully improve performance. This makes it unlikely that the minimal agent design itself is the main bottleneck.
>
> **2. Root cause analysis (model-side factors)**
>
> Our results instead point to two dominant model-side causes:
>
> **Lack of temporal modeling capacity (architectural).**
> The failure patterns in our Tracking and Timing unit tests (Fig. 5) show that models treat frames largely independently and often update their reasoning only rarely (anchoring bias). This aligns with our anchoring-bias analysis in **Appendix F.3** and is consistent with recent work [1, 2] arguing that current MLLMs lack explicit temporal modeling in their vision encoders.
>
> **Training data limitations (static-heavy pretraining).**
> Most evaluated models are primarily trained on static image-text pairs with limited exposure to dynamic visual sequences. This matches the empirical pattern we observe: relatively strong single-frame grounding but weak multi-frame reasoning and trajectory understanding.
>
> We also tested **inference-related factors** (e.g., sampling strategy, input resolution) in **Appendix E.1/E.2**. While these settings do have some effect, their impact is modest compared to the deficits revealed in our unit tests and anchoring-bias analysis, further supporting the view that the core limitations are architectural/data-related rather than agent-side.
>
> ---
>
> **References:**
>
> [1] STORM: Token-Efficient Long Video Understanding for Multimodal LLMs.
> [2] Can Multimodal LLMs do Visual Temporal Understanding and Reasoning? The answer is No!

---

> ### Author Response · Authors · 2025-11-21
>
> > **Q2: Connection to real-world capabilities**
>
> Although our evaluation uses controlled game environments, the underlying abilities required in V-MAGE directly correspond to those needed in real-world settings.
>
> The sub-skills tested in V-MAGE are prerequisites for real-world embodied or interactive tasks, including:
>
> - **Object tracking** (e.g., Tracking in Pong),
> - **Spatial planning and navigation** (e.g., pathfinding and acceleration reasoning in Race),
> - **Reaction timing** (e.g., Timing in FlappyBird),
> - **Long-horizon visual memory and strategy adaptation** (e.g., multi-step history in SuperMario and TempestRun).
>
> A model that frequently hallucinates object positions or directions even in a simplified 2D setting (perception errors account for roughly half of GPT-4o’s errors in Fig. 7) is unlikely to be reliable in more complex 3D navigation or GUI control scenarios. These are foundational perceptual-cognitive skills for robotics, virtual assistants, and OS/GUI agents.
>
> Moreover, the stronger closed-source models (GPT-4o, Gemini) and larger open-source models (Qwen2.5-VL-72B, InternVL2.5-78B) that perform best on V-MAGE are also among the top-ranked models on widely used multimodal benchmarks such as **MMBench** [3] and **MMMU** [4]. This qualitative alignment suggests that V-MAGE is consistent with general multimodal competence, while placing a much stronger emphasis on **temporal, vision-grounded reasoning** than existing static-image benchmarks.
>
> We hope these clarifications demonstrate that V-MAGE is not just a game suite, but a diagnostic tool specifically designed to isolate and quantify the dynamic, vision-centric reasoning gap in current primarily statically trained MLLMs.
>
> ---
>
> **References:**
>
> [3] MMBench: Is Your Multi-modal Model an All-around Player?
> [4] MMMU: A Massive Multi-discipline Multimodal Understanding and Reasoning Benchmark for Expert AGI

---

### Note · Authors · 2025-12-25

I have read and agree with the venue's withdrawal policy on behalf of myself and my co-authors.